# PRRX1 is a master transcription factor of stromal fibroblasts for myofibroblastic lineage progression

Keun-Woo Lee [1,9], So-Young Yeo [1,9], Jeong-Ryeol Gong[2,9], Ok-Jae Koo[3], Insuk Sohn[4], Woo Yong Lee[5], Hee Cheol Kim[5], Seong Hyeon Yun[5], Yong Beom Cho[5], Mi-Ae Choi[1], Sugyun An [2], Juhee Kim [2], Chang Ohk Sung [6✉], Kwang-Hyun Cho [2✉] & Seok-Hyung Kim[1,7,8✉]

Although stromal fibroblasts play a critical role in cancer progression, their identities remain unclear as they exhibit high heterogeneity and plasticity. Here, a master transcription factor (mTF) constructing core-regulatory circuitry, *PRRX1*, which determines the fibroblast lineage with a myofibroblastic phenotype, is identified for the fibroblast subgroup. *PRRX1* orchestrates the functional drift of fibroblasts into myofibroblastic phenotype via TGF-β signaling by remodeling a super-enhancer landscape. Such reprogrammed fibroblasts have myofibroblastic functions resulting in markedly enhanced tumorigenicity and aggressiveness of cancer. PRRX1 expression in cancer-associated fibroblast (CAF) has an unfavorable prognosis in multiple cancer types. Fibroblast-specific *PRRX1* depletion induces long-term and sustained complete remission of chemotherapy-resistant cancer in genetically engineered mice models. This study reveals CAF subpopulations based on super-enhancer profiles including *PRRX1*. Therefore, mTFs, including *PRRX1*, provide another opportunity for establishing a hierarchical classification system of fibroblasts and cancer treatment by targeting fibroblasts.

[1] Department of Health Science and Technology, Samsung Advanced Institute for Health Science and Technology, Sungkyunkwan University, Seoul 06351, Republic of Korea. [2] Department of Bio and Brain Engineering, Korea Advanced Institute of Science and Technology, Daejeon 34141, Republic of Korea. [3] ToolGen Inc., Gasan digital 1-ro, Geumcheon, Seoul 08594, Republic of Korea. [4] Arontier Inc., Gangnam-daero, Seoul, Republic of Korea. [5] Department of Surgery, Samsung Medical Center, Sungkyunkwan University School of Medicine, Seoul, Republic of Korea. [6] Department of Pathology, Asan Medical Center, University of Ulsan College of Medicine, Seoul, Republic of Korea. [7] Department of Pathology, Samsung Medical Center, Sungkyunkwan University School of Medicine, Seoul, Republic of Korea. [8] Single Cell Network Research Center, Sungkyunkwan University School of Medicine, Suwon, Gyeonggi-do 16419, Republic of Korea. [9]These authors contributed equally: Keun-Woo Lee, So-Young Yeo, Jeong-Ryeol Gong. ✉email: co.sung@amc.seoul.kr; ckh@kaist.ac.kr; platoshkim@gmail.com

Cancer-associated fibroblasts (CAFs) are poorly understood mesenchymal cells that are abundant in activated tumor stroma. Growing evidence indicates that CAFs are not just passive bystanders but key players promoting cancer progression and resistance to treatment[1–4]. Such tumor-promoting activity of CAFs is mediated through various mechanisms, including synthesis and remodeling of extracellular matrix (ECM), secretion of a wide repertoire of soluble factors, extensive interactions with cancer cells, and regulation of angiogenesis and immunity[5–7].

Although CAF treatment and inhibition is a promising new cancer treatment strategy[8–11], the clinical application of CAF inhibitors is lacking due to several challenges. One such challenge is the lack of fundamental knowledge on the heterogeneity of CAF populations. CAFs differ in several aspects, including their origin and ability to promote tumors[12]. In terms of their ability to promote tumors, there exists a broad spectrum of cell types ranging from highly pro-tumorigenic subsets to less tumorigenic subsets as well as cancer-repressing subsets of CAFs (rCAFs)[13]. To selectively target only the highly pro-tumorigenic CAFs, without interfering with the rCAFs, the CAF subset that strongly promotes cancer should be identified.

Another challenge is the absence of a suitable innovative in vivo mouse model that recapitulates desmoplasia, a critical feature of the tumor microenvironment (TME) observed in human cancer tissues[14]. Desmoplasia, characterized by excessive fibroblast pro-liferation and extensive ECM deposition, is frequently observed in various cancers, including colorectal cancer. Recently, desmoplasia has been reported to be strongly associated with resistance to immuno- and chemotherapy[15]. Several studies have attempted to address these challenges, including the use of single-cell RNA sequencing (scRNA-seq)[16–18], but have been unsuccessful for several reasons, such as CAF complexity, alterations in CAFs caused by in vitro culture, and lack of specific surface markers.

A powerful but untested solution is to identify master tran-scription factors (mTFs) for each CAF subset. mTFs are at the top of the gene regulation hierarchy and determine cell identity based on cell fate and differentiation. Specifically, a group of mTFs forms a core-regulatory circuitry (CRC) that controls the core transcriptional program governing cell identity. Canonical examples of mTFs include *NANOG*, *OCT4*, and *SOX2*, which maintain pluripotency in stem cells[19]. However, no mTF has yet been identified in CAFs. Therefore, identifying subset-specific mTFs in CAFs may enable accurate classification of the various subsets of CAFs with higher specificity than other known mar-kers, including FAP, SMA, and FSP1[20]. Additionally, mTFs represent ideal therapeutic targets that would enable selective targeting of CAF subsets due to their high degree of cell-type specificity. Finally, identifying a CAF mTF would allow the development of a mouse model of cancer capable of recapitu-lating desmoplasia. Using genetically engineered mice, desmo-plasia can be recapitulated by reprogramming fibroblasts into highly aggressive CAFs in vivo through forced expression of the CAF-specific mTF in fibroblasts. The discovery of CAF mTFs can help overcome current challenges in the study of CAF biology.

In this work, through extensive in vivo and in vitro studies and system biology approaches, including large-scale chromatin immu-noprecipitation sequencing (ChIP-seq), RNA sequencing (RNA-seq), and scRNA-seq of fibroblasts, we identify the mTF, *PRRX1* (paired related homeobox 1), which may help determine the lineage of a fibroblast subgroup exhibiting myofibroblastic phenotype.

## Results

### *PRRX1* is a CAF lineage-specific TF that strongly correlates with unfavorable clinical outcomes. To identify CAF-specific and functionally important TFs, large-scale scRNA-seq data[21–27]

were analyzed with annotation using the CellAssign probabilistic model[28] (Supplementary Fig. 1A). *PRRX1* was more highly expressed in CAFs than in fibroblasts of normal tissues, and *PRRX1* expression was strongly correlated with fibroblast-activation score in all six datasets from six organs, including cancer and normal tissues (lung, colon, pancreas, head and neck, ovary, and stomach) (Fig. 1A, B, Supplementary Fig. 1B).

Further analysis of these scRNA-seq data revealed that *PRRX1* is expressed in 30–90% of the total CAFs (Fig. 1C, Supplementary Fig. 1C) and is highly specific to CAFs (Fig. 1D, Supplementary Fig. 1D). Using copy number analysis (anueploidy) of the scRNA-seq and cell-type annotation, we confirmed that most *PRRX1*-expressing cells in the scRNA-seq data were not cancer cells (Supplementary Fig. 1E).

Next, PRRX1 expression was examined in various types of cancer, including colon, stomach, lung, esophageal, and breast cancers, using immunohistochemistry. PRRX1 protein was highly expressed in CAFs across the assessed tissues but not in nontumor fibroblasts and other cells (Fig. 1E, Supplementary Fig. 1F). In the colon, stomach, and esophageal cancers, patients with PRRX1 protein expression in CAFs showed unfavorable prognosis (Fig. 1F, Supplementary Data 1). The Cancer Genome Atlas (TCGA) and other independent public datasets were used to confirm these findings. We found a significant positive correlation between *PRRX1* expression and poor survival (Supplementary Fig. 2A, B). These data show that *PRRX1* is highly expressed mainly in CAFs and strongly correlates with poor survival across several cancer types.

### Prrx1 is essential for CAFs to facilitate tumorigenicity and metastasis of cografted tumors. The clinical implications of CAF-specific *PRRX1* expression prompted the investigation into the role of Prrx1 in the tumor-promoting ability of CAFs in vivo. Tumor cells were cografted with fibroblasts into immunodeficient mice, and the effect of *Prrx1* expression on fibroblast-driven tumor progression was evaluated. Due to the complexity and heterogeneity of tumor cells and fibroblasts, several cotrans-plantation experiments with various cancer cells and fibroblasts were repeatedly performed to avoid cell-dependent bias (Sup-plementary Figs. 3–5).

First, 168FARN breast cancer cells were cografted with MMTV-CAFs into NOD/SCID mice. MMTV-CAFs were established from MMTV-PyMT breast cancer mouse models[21]. As expected, MMTV-CAFs significantly enhanced tumor growth compared with tumor cells grafted alone. However, *Prrx1*-deleted MMTV-CAFs failed to completely enhance the growth of cografted tumors, indicating that Prrx1 is essential for the tumor-promoting ability of MMTV-CAFs (Supplementary Fig. 3A). Additionally, the 168FARN cancer cells became histologically poorly differentiated when cografted with MMTV-CAFs; however, this effect was not observed when cografted with *Prrx1*-deleted MMTV-CAFs (Supplementary Fig. 3A bottom).

Then, the systemic effect of MMTV-CAFs on tumor cells was explored by injecting MMTV-CAFs into the tail vein of NOD/SCID mice. In contrast, 168FARN cancer cells were implanted into a mammary fat pad (Supplementary Fig. 3B). Massive metastasis of 168FARN cancer cells to the lungs was observed, causing the death of four of five mice. These findings suggested that MMTV-CAFs in lungs moved from the tail vein were likely to induce metastasis of 168FARN cancer cells present in the fat pad through systemic effect. This interesting phenomenon was not observed in mice injected with Prrx1-deleted MMTV-CAFs. Immunostaining for cytokeratin and αSMA confirmed the metastasis of 168FARN cells to the lung (Supplementary Fig. 3C).

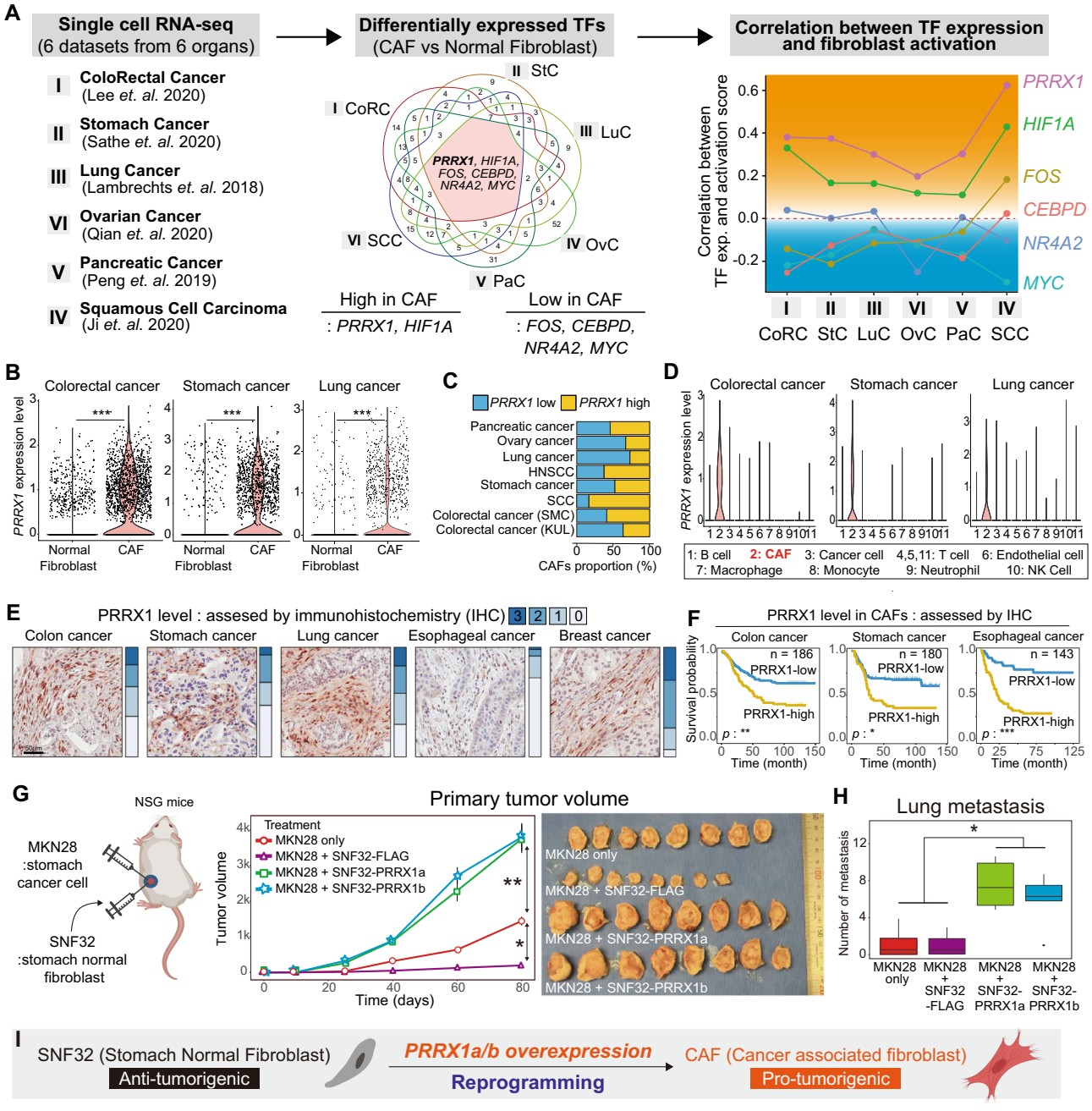

**Fig. 1 Discovery of PRRX1 as a highly CAF-specific TF with significant correlation with poor prognosis in diverse cancer types. A** Schematic diagram showing the method used to identify activated CAF-specific TFs in six cancer tissues compared with that in normal fibroblasts using large-scale single-cell RNA sequencing (scRNA-seq) datasets. PRRX1 was identified as a CAF-specific TF and was highly correlated with the CAF activation score. **B** Level of PRRX1 expression in normal fibroblasts and CAFs in colorectal, stomach, and lung cancers (two-sided Wilcoxon rank-sum test, ***$p < 2e-16$, Colorectal cancer $n = 2047$, Stomach cancer $n = 1821$, Lung cancer $n = 2002$). Others are shown in Supplementary Fig. 1B. **C** Bar plots of the proportion of CAFs with high-PRRX1 expression in the eight scRNA-seq datasets. The criteria used for categorizing high (+)- and low (−)-PRRX1 CAFs are shown in Supplementary Fig. 1C. **D** Analysis of eight scRNA-seq datasets revealed that PRRX1 expression is highly restricted to CAFs across 11 cell types (two-sided Wilcoxon rank-sum test, ***$p < 2e-16$). Others are shown in Supplementary Fig. 2E. **E** Immunohistochemical analyses revealed that PRRX1 protein is specifically detected in CAFs in all five cancer types. Representative Images from five FFPE tumor microarray sections (TMA). **F** Kaplan–Meier (KM) survival analyses using immunohistochemistry revealed that a high level of CAF-specific PRRX1 expression is significantly associated with unfavorable overall survival in the colon, stomach, and esophageal cancers. $p$ value was calculated using log-rank test: *$p < 0.01$, **$p < 0.001$, ***$p < 0.0001$. **G**, **H** (i) Human stomach cancer cells (MKN28) alone, (ii) mixed with SNF32 (human stomach normal fibroblasts)-FLAG (control), or either (iii) Prrx1a- or (iv) Prrx1b-overexpressing SNF32 were co-injected subcutaneously into NSG mice. Measurement of primary tumor volume and H&E images of tumors in each group. **G** Data are presented as the mean ± SEM, $n = 8$ tumors per group, two tailed $t$ test: *$p < 0.005$, **$p < 0.0001$. **H** Box plot showing number of lung metastasis($n = 6$ mice/group). $p$ value was calculated using two tailed $t$ test: *$p < 0.05$. Data are presented as box-and-whisker plots indicating median (middle line) and minimum and maximum values (whiskers). **I** Schematic summary of the results. Source data and exact p values are provided as a Source Data file.

Similar experiments were repeated by cografting human colon cancer cells (HT29 and HCT116) and stomach cancer cells (MKN28) with MEF (mouse embryofibroblast). Then, identical results, which are presented in Supplementary Fig. 4 were obtained.

Next, we aimed to determine whether Prrx1 alone could reprogram normal fibroblasts into CAFs in vivo. Stomach normal fibroblast 32 (SNF32) was cografted with stomach cancer cells (MKN28) into NSG mice. SNF32 inhibited the tumor growth of MKN28 cells, confirming that normal fibroblasts, unlike CAFs, suppress tumor progression. In contrast, SNF32-Prrx1a and SNF32-Prrx1b, which express the Prrx1a and Prrx1b isoforms, respectively, significantly increased the growth of MKN28 cografted tumors (Fig. 1G, Supplementary Fig. 6A, B). Moreover, SNF32-Prrx1a and SNF32-Prrx1b promoted the formation of a micropapillary pattern characterized by a mixture of primitive stromal cells and stem cell-like cancer cells (Supplementary Fig. 5). Further, SNF32-Prrx1a and SNF32-Prrx1b significantly enhanced MKN28 lung metastasis (Fig. 1H). Notably, the metastasized cancer cells were surrounded by fibroblasts, mimicking desmoplasia (Supplementary Fig. 5).

These results and additional supporting findings (Supplementary Fig. 4) indicate that Prrx1 is essential for the tumorigenicity and metastasis-inducing capacity of CAFs and that Prrx1 alone can reprogram tumor-suppressive normal fibroblasts into CAFs (Fig. 1I).

**Fibroblast-specific *Prrx1* expression alone is sufficient to increase tumor recurrence after surgical removal in immunocompetent mice.** The importance of Prrx1 in the pro-tumorigenic activity of CAFs was confirmed by cotransplanting cancer cells and CAFs into immunodeficient mice. To better understand *Prrx1* role in vivo, we generated genetically engineered mouse models that satisfied the following conditions: the mice should be immunocompetent, and Prrx1 expression could be turned on or off in indigenous fibroblasts within tumor-bearing mice.

Initially, a mouse model in which *Prrx1* was conditionally deleted only in the fibroblasts was generated by combining fibroblast-specific CAS9-expressing mice (FSP1;cre CAS9EGFP) with CRISPR-CAS9 technology. Then, the effect of fibroblast-specific deletion of *Prrx1* on tumorigenicity in this mouse model was examined. *Prrx1* in fibroblasts was deleted by injecting single-guide RNA (sgRNA) against *Prrx1* into the FSP1;creCAS9EGFP mice, and LLC1-Luc-GFP cancer cells were subcutaneously transplanted. The LLC1 tumorigenicity in FSP1;creCAS9EGFP mice decreased from 6/6 in the control (treated by sgNS) to 2/10 in the fibroblast-specific *Prrx1*-deleted group (treated by sgPrrx1), indicating that *Prrx1* expression in fibroblasts is indispensable for tumorigenesis (Fig. 2A).

Next, we confirmed whether fibroblast-specific induction of *PRRX1* expression promotes cancer progression. A doxycycline (Dox)-inducible mouse model, Col1a2;rtTATetO7-Prrx1Luc, was developed by crossing Col1a2-rtTA mice with TetO7-Prrx1-luciferase mice. In this mouse model, *Prrx1* expression was induced specifically in fibroblasts by Dox. LLC1-Luc-GFP cancer cells were transplanted into Col1a2;rtTATetO7-Prrx1Luc mice, and tumor growth and lung and liver metastases were found to be significantly enhanced in the fibroblast-specific *Prrx1*-expressing group [Dox (+)] (Fig. 2B, Supplementary Fig. 7A) compared with Dox (−) control group. These results suggest that the induction of *Prrx1* expression in fibroblasts alone could promote tumor progression and metastasis in vivo.

In addition to metastasis, tumor recurrence after surgical resection is also clinically important. An in vivo model of tumor recurrence was established using Col1a2;rtTATetO7-Prrx1Luc mice, and the effect of fibroblast-specific *Prrx1* expression on tumor recurrence was evaluated.

Specifically, colon cancer cells (MC38) were implanted subcutaneously in Col1a2;rtTATetO7-Prrx1Luc mice; once the tumor grew, it was surgically excised. These mice were treated with Dox to induce Prrx1 expression in fibroblasts activated by this surgical procedure. Notably, when Dox was administered, cancer recurrence was observed in the mediastinum around the heart 3 weeks after surgical resection of the tumor. However, no such recurrence was observed in the control group that was not treated with Dox (Fig. 2C). These results indicated that the induction of *Prrx1* expression in fibroblasts alone induces recurrence of cancer following the surgical resection of the primary tumor.

Next, the effect of fibroblast-specific *Prrx1* expression on de novo carcinogenesis in vivo was investigated. Col1a2;rtTATetO7-Prrx1Luc mice were treated with AOM/DSS to initiate de novo carcinogenesis in the colon. In the Dox-untreated control group of these mice, only one of eight mice developed carcinoma (early stage), whereas in the Dox-treated group, invasive carcinoma developed in all 13 mice, including two mice showing liver metastasis (Fig. 2D, Supplementary Fig. 7B, C). Increased fibroblast proliferation with stromal fibrosis around tumor cells, a characteristic of desmoplasia, was observed in almost all Dox-treated mice. Desmoplasia is an important feature of human colorectal cancer and causes resistance to immunotherapy with immune checkpoint inhibitors. These data suggest that fibroblast-specific *Prrx1* expression induces desmoplasia and promotes cancer progression.

**Prrx1 activates a wide spectrum of myofibroblast-like functions in CAFs.** The clinical and in vivo significance of Prrx1 in CAFs shown above prompted the investigation into how Prrx1 induces the tumor-promoting activity of CAF. First, to gain a broad insight into the Prrx1 role in CAF in overall human tumors, the scRNA-seq data of seven cancer types were analyzed. Myofibroblast-related gene sets were remarkably enriched in the Prrx1-high CAF subpopulation across all seven cancer types (Fig. 3A left panel). To gain a deeper understanding of the function of Prrx1 in CAFs, the effect of Prrx1 knockdown in representative mouse fibroblasts, such as MEF, MMTV-CAF (CAF derived from MMTV-PyMT mouse cancer), and wound-healing fibroblasts (fibroblast derived from mouse wound-healing tissue), were investigated. Similarly, Prrx1 knockdown induced a significant decrease in the expression of a set of myofibroblast-associated genes common in these cells (Fig. 3A right panel).

Based on the mRNA expression data, we validated the potential functions of Prrx1 in CAFs. First, we found that Prrx1 expression is indispensable for cellular proliferation and resistance to apoptosis consistently across the types of CAFs (Fig. 3B, Supplementary Fig. 8A, B). For example, the Prrx1-high subpopulation, sorted from SCAFs containing Prrx1 promoter-GFP reporter, showed lower apoptosis and stronger resistance to serum starvation-induced apoptosis than the Prrx1-low subpopulation did (Fig. 3B, Supplementary Fig. 6C, D).

Then, the effect of Prrx1 on the ECM-remodeling capability of fibroblasts, a key function of myofibroblasts, was examined. Prrx1-high subpopulation, sorted from SCAFs, showed higher ECM contraction than the Prrx1-low subpopulation did and lost their ECM contracting capability in response to shRNA-mediated Prrx1 silencing (Fig. 3C, left panel). Additionally, shRNA-mediated knockdown of Prrx1 in MMTV-CAFs resulted in a significant loss of ECM contractile activity (Fig. 3C, right panel). The significance of Prrx1 in ECM remodeling was repeatedly

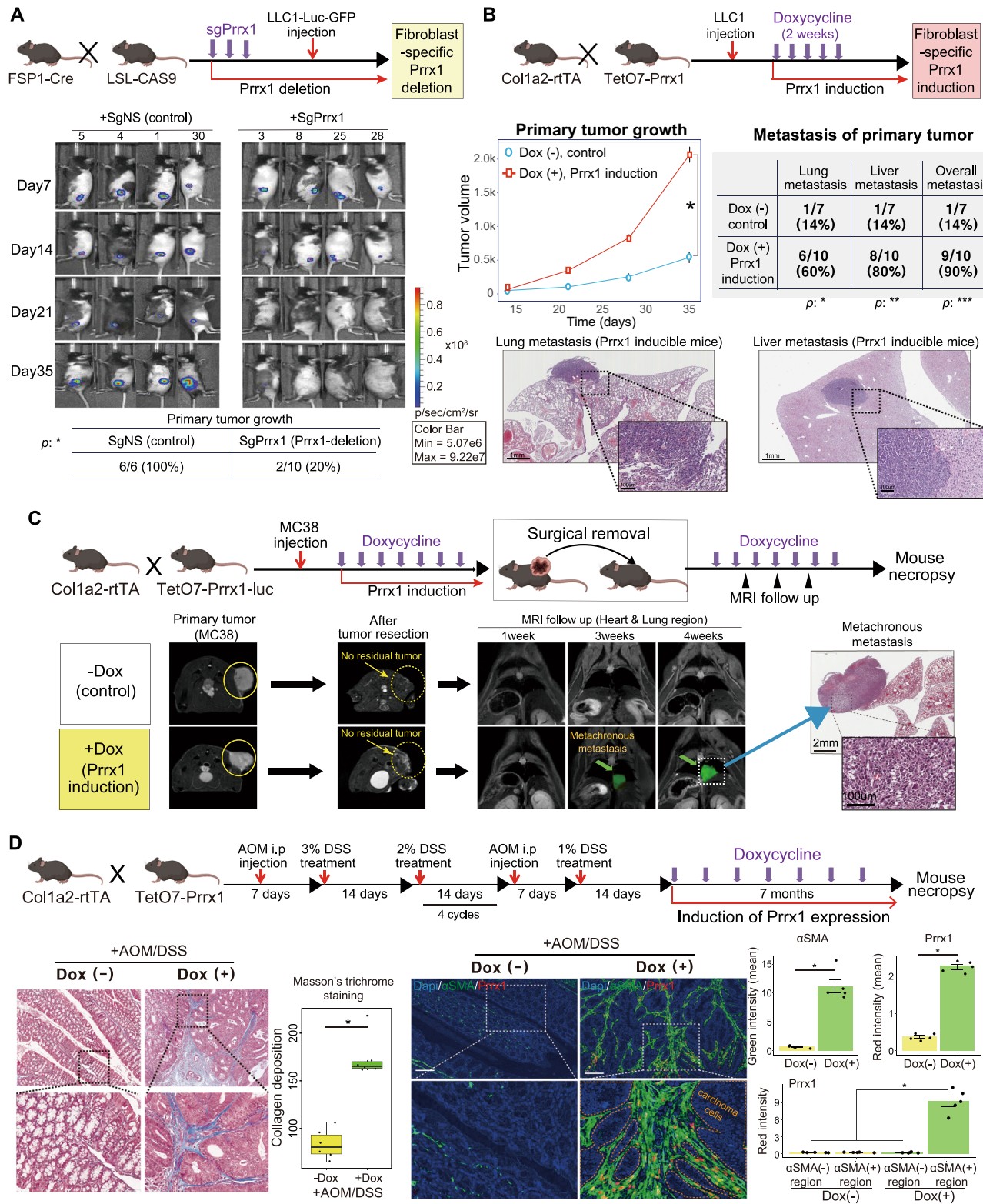

observed in other types of fibroblasts and CAFs (Supplementary Fig. 8C).

The direct effect of CAFs on cancer cells was investigated using a coculture model (Fig. 3D). When tumor cells (168FARN-Luc-GFP) were cocultured with CAFs (MMTV-CAFs) in the hanging drop culture system, the size of the resulting spheroids was significantly larger than that of spheroids formed by cancer cells alone. Notably, *Prrx1*-deleted CAFs [MMTV-CAFs (sgPrrx1)] did

not increase the spheroid formation when they were cocultured with tumor cells (168FARN-Luc-GFP), suggesting that Prrx1 is vital for the tumor-promoting activity of CAFs (Fig. 3E). These findings were confirmed again in similar and additional experiments employing other cancer cells and CAF as shown in Supplementary Figs. 8D–H and 9A, B.

To better understand the changes in cancer cells during coculture with CAFs, 168FARN-Luc-GFP and LLC cancer cells

**Fig. 2 Prrx1 expression in fibroblasts is essential for in vivo tumorigenicity and metastasis in genetically engineered immunocompetent mice. A** Mouse model in which fibroblast-specific Prrx1 was depleted using sgRNA against Prrx1 (sgPrrx1) before LLC1 transplantation. Bioluminescence imaging of representative mice in each group by date. sgNS was used as the control (n = 6 mice: sgNS group, n = 10 mice: sgPrrx1 group, *p < 0.005 by chi-squared test). **B** Transgenic mice (Col1a2;[rtTA]TetO7-Prrx1[Luc]) capable of fibroblast-specific Prrx1 induction at a desired time point using the Tet-on system. Tumor growth curves; Data are presented as the mean ± SEM; n = 7 mice: Dox- group, n = 10: Dox+ group (two tailed t test: *p < 0.005). Table comparing the incidence of secondary metastases. H&E staining of representative lung and liver metastasis in Prrx1-induced mice. *p < 0.05, **p < 0.01, ***p < 0.005 by chi-squared test. **C** Prrx1 induction in fibroblasts after cutaneous transplantation of mouse colon cancer cells MC38-Luc-GFP (tumor formation, skin incision, and tumor removal, followed by MRI for 4 weeks). Representative MRI (yellow arrow indicates metastatic growth near the lung) and H&E stained images of metachronous metastasis from Prrx1-induced mice (n = 10 mice: Dox− group, n = 15 mice: Dox+ group). **D** Timeline for in vivo model of AOM/DSS-induced colorectal cancer. (n = 8 mice: AOM/DSS+ Dox- group, n = 13 mice: AOM/DSS+ Dox− group) Masson's trichrome staining for assessing desmoplasia in mice and quantification of collagen deposition. Box plots showing quantification of collagen deposition(n = 6 measurements, *p < 0.001 by two tailed t test). Data are presented as box-and-whisker plots indicating median (middle line) and minimum and maximum values (whiskers). Immunohistochemical staining was conducted using AOM/DSS treatment in WT and fibroblast-specific Prrx1-inducible (Col1a2;[rtTA]TetO7-Prrx1[Luc]) mouse tissues. Visualization using ImageJ Fiji (DAPI [hematoxylin]-blue, αSMA-green, Prrx1-red) and quantification of staining images. Scale bar, 100 μm. Bar plot showing quantification of aSMA and Prrx1 intensity. Data are presented as the mean ± SEM; n = 5 measurements (two tailed t test: *p < 0.0001). Source data and exact p values are provided as a Source Data file.

within the spheroids were isolated from GFP-expressing fibroblasts within spheroids and analyzed using RNA-seq (Fig. 3F, Supplementary Fig. 10). *Prrx1*-expressing CAFs induced significant and consistent gene expressional changes in cancer cells; they increased gene expressions related to poor prognoses, such as EMT and metastasis, cancer stem cell phenotype, chemotherapy resistance, tumor cell proliferation, invasion, and glycolysis. Next, we investigated whether these gene expression changes in cancer cells induced by coculture with CAFs resulted in changes in functions of cancer cells. Indeed, the migration, invasion, and stemness of 168FARN cancer cells were significantly increased when cocultured with expressing MMTV-CAFs compared with cancer cells cocultured with *Prrx1*-deleted MMTV-CAFs (Fig. 3G). Additionally, the soluble factors secreted only by Prrx1-expressing CAFs were identified using a cytokine array. Prrx1-induced soluble factors included IL6, CSF, CCL5, and CXCL12, which promote both cancer progression and wound healing (Supplementary Fig. 9C, D). Overall, these results indicate that Prrx1 controls a spectrum of myofibroblastic functions, ranging from ECM remodeling to secretion of wound healing-related soluble factors.

**Prrx1 is a master Transcription Factor (mTF) orchestrating myofibroblastic programs in murine fibroblasts.** As shown above, Prrx1 is critically involved in various myofibroblast-related core functions. Therefore, Prrx1 could be a high-level regulator of myofibroblast-related genes in a coherent gene regulatory program. Recently, Prrx1 was proposed as an mTF candidate in mesenchymal cells in embryos and certain cancer types[22,23]. Therefore, we investigated whether Prrx1 is the mTF that drives myofibroblastic programs in fibroblasts and CAFs using an integrated systems biology approach.

We identified candidate mTFs in the three representative murine fibroblasts, namely, MEFs, skin wound-healing fibroblasts, and MMTV-CAFs, based on genome-wide super-enhancer mapping. For super-enhancer mapping, H3K27ac ChIP-seq was conducted on these fibroblasts; the resulting ChIP-seq data were processed using the bowtie-MACS-ROSE pipeline[19,24–27] to identify super-enhancers (Supplementary Data 2). The mTF candidates were initially screened out based on these super-enhancer mapping data and then further narrowed down by selecting those specifically enriched in activated fibroblasts of murine wound tissue (Fig. 4A, Supplementary Fig. 11A, B).

As expected, Prrx1 was identified as a common mTF candidate in all three types of fibroblasts along with seven other candidates (Fig. 4B). The potential target genes for these eight common TFs, including Prrx1, were mainly involved in ECM stiffness, TGF-β

pathway, angiogenesis, and chromatin remodeling pathway (Fig. 4C). Among the eight common mTFs, Prrx1 was the most highly expressed gene in the activated fibroblasts of murine wound tissue (Fig. 4D, Supplementary Fig. 11C). To confirm that Prrx1 is an mTF, additional ChIP-seq for Prrx1 (Supplementary Fig. 12A–D) in these fibroblasts was performed. We found that Prrx1 is preferentially bound to super-enhancers over typical enhancers like other known mTFs[19,29,30] (Fig. 4E). Prrx1 was found to directly bind to its super-enhancers, forming an auto-regulatory loop (Fig. 4F left). Moreover, it is directly bound to super-enhancers of the other seven mTF. Finally, the binding motifs of the other candidates were abundant in their super-enhancers and in others, forming an interconnected feed-forward transcriptional loop. Therefore, Prrx1 satisfied all criteria proposed by Young *et al.* for being an mTF[29,30]. Consistently, Prrx1 bound to the super-enhancer of Col1a1, which was invariably upregulated in activated fibroblasts (Fig. 4F right).

mTFs can remodel the super-enhancer landscape[19]. To determine whether Prrx1 can remodel super-enhancer landscapes, the super-enhancer profiles were compared before and after Prrx1 deletion in the three types of fibroblasts. Prrx1 deletion was induced in these three fibroblast types, and H3K27ac ChIP-seq was performed on these cells. Results showed significant changes in the super-enhancer landscape in all fibroblast types (Supplementary Fig. 13A, B). Additionally, the genes associated with Prrx1-dependent super-enhancers that disappeared following Prrx1 deletion were significantly enriched in the myofibroblastic signatures (Supplementary Fig. 13C).

As broad H3K4me3 peaks have recently been reported to be epigenetic markers of cell-specific identity[31,32], we performed genome-scale mapping of broad H3K4me3 peaks in these three fibroblast types. The resulting ChIP-seq data were analyzed using the MACS2 algorithm (Supplementary Fig. 14A, B, Supplementary Fig. 15). All mTF candidates, including Prrx1, were associated with broad H3K4me3 peaks in their promoter regions, thereby validating mTFs identified on the basis of super-enhancer mapping through an independent assay (Supplementary Fig. 13). Moreover, the genes associated with Prrx1-specific broad H3K4me3 peaks were significantly enriched in myofibroblastic signatures (Supplementary Fig. 14C, D).

Thus, Prrx1 is the mTF that orchestrates myofibroblast programs in the three representative murine fibroblasts.

***PRRX1* forms a complex with Smad3 for TGF-β signaling.** To gain further insights into the cellular programs regulated by Prrx1, 515 genes directly occupied by Prrx1 as revealed by ChIP-seq were examined for Prrx1. These 515 genes were highly

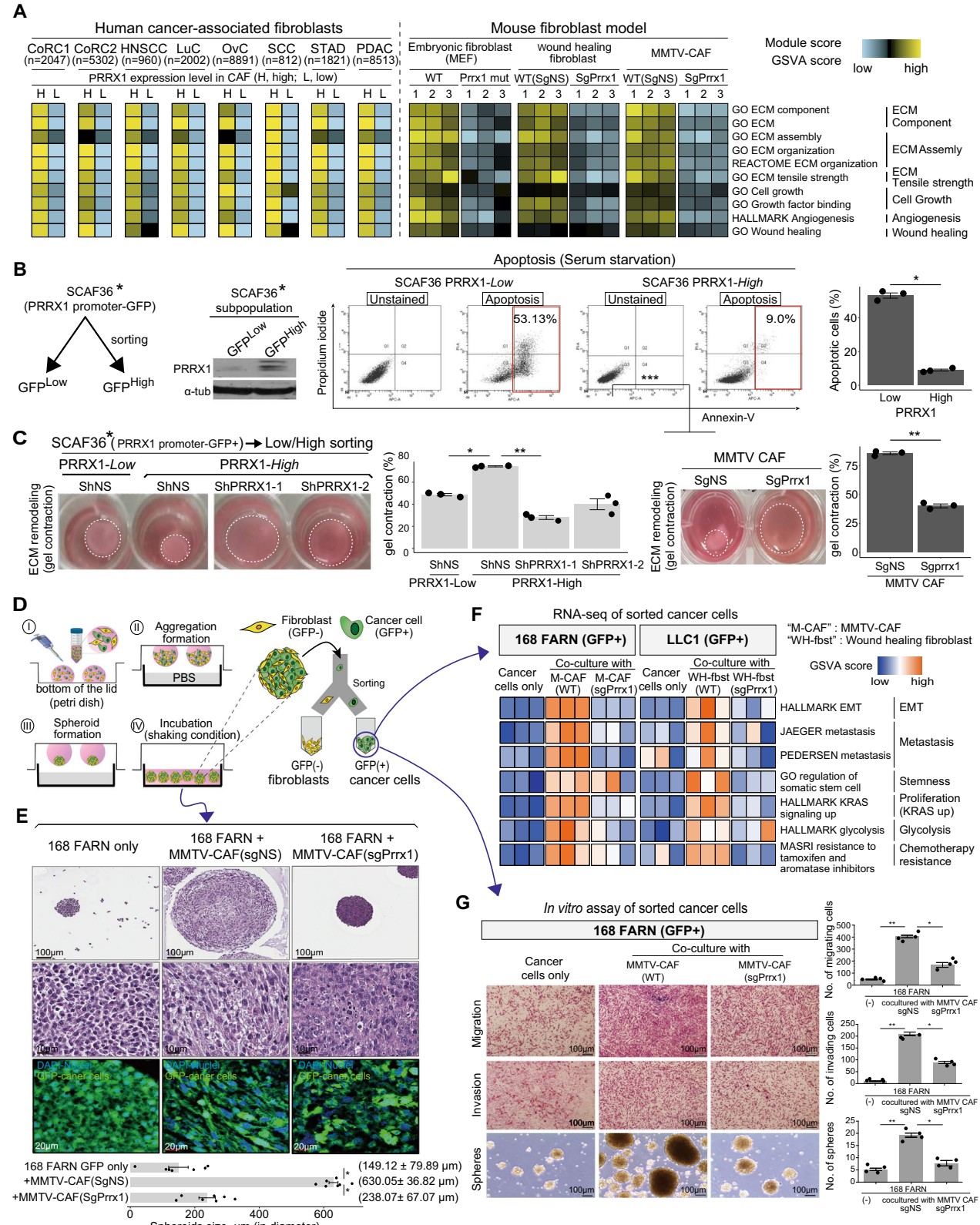

enriched in myofibroblastic programs (Fig. 5A); this observation is consistent with that shown in Fig. 3. Among those enriched myofibroblastic programs, the TGF-β signaling was included, implying that Prrx1 executes myofibroblastic programs by TGF-β signaling. An assay for three-dimensional (3D) ECM formation was performed to validate this hypothesis. The assay revealed that *Prrx1*-expressing MEFs enhanced fibronectin deposition and

cell polarity alignment in response to TGF-β, whereas this effect was not observed in *Prrx1*-knockout MEFs. These results suggest that Prrx1 is essential for TGF-β-mediated induction of myofibroblastic phenotypes (Fig. 5B).

To explore the link between Prrx1 and TGF-β, binding motifs analysis was performed, and we found that the binding motifs of TGF-β signaling-related TFs (Smad2, Smad3, Smad4, and

**Fig. 3 Prrx1 orchestrates myofibroblast-like functions in CAFs across diverse cancer types, enhancing EMT and stemness of tumor cells. A** In public scRNA-seq data, human CAFs were virtually sorted into high (+)- and low (−)-PRRX1 CAFs. Both high- and low-Prrx1 CAFs were compared using module score analysis. Ex vivo fibroblast cultures of either Prrx1 WT or Prrx1 KO for each type of mouse fibroblast were established. Then, the two groups were compared for each type of fibroblast using GSVA analysis. Heatmap shows the module score from scRNA-seq of human CAFs and GSVA score from bulk RNA-seq of mouse fibroblast models. **B** After establishing PRRX1 promoter-driven model of SCAF36, apoptosis analysis was performed at each group (PRRX1 low and PRRX1 high). Experiments were conducted in triplicate. Bar plot showing percentage of cell apoptosis(%). Data are presented as the mean ± SEM; $n = 3$ independent experiments (two tailed $t$ test: $*p < 0.0001$). **C** Gel contraction assay, depending on the expression of Prrx1 in fibroblasts. Representative images are shown. Data are presented as the mean ± SEM; $n = 3$ independent experiments (two tailed $t$ test: $*p < 0.005$, $**p < 0.0001$). **D** Establishment of hanging drop direct 3D spheroid coculture system. Breast cancer cells 168FARN-Luc-GFP were cocultured with either MMTV-CAFs sgNS or sgPrrx1, and lung cancer cells LLC1-Luc-GFP were also cocultured with either wound-healing fibroblasts sgNS or sgPrrx1, using 3D spheroid systems. **E** H&E and immunohistochemical staining of representative spheroids in each group (DAPI: nuclei, GFP: Luc-GFP-tagged cancer cells). Bar plot showing quantification of spheroid size. Data are presented as the mean ± SEM; $n = 7$ independent experiments (two tailed $t$ test: $*p < 0.001$). **F** GFP$^+$ cancer cells were isolated from spheroids using FACS, and RNA-seq of these cancer cells was performed. **G** 168FARN-Luc-GFP$^+$ cancer cells were isolated from spheroids using FACS, and then invasion, migration, and sphere forming assays were performed on isolated cancer cells. Data are presented as the mean ± SEM; $n = 4$ independent experiments (two tailed $t$ test: $*p < 0.005$, $**p < 0.0001$). Source data and exact p values are provided as a Source Data file.

---

Ap1)[33,34] were frequently found close to the Prrx1-binding site (Fig. 5C, D). The complex of Smad3 and Prrx1 predicted by motif analysis was confirmed by a co-immunoprecipitation experiment (Fig. 5E). Prrx1 and Smad3 were simultaneously co-expressed in the nucleus of Prrx1-WT MEF but not in Prrx1-mutant MEF under the stimulation of TGF-β (Fig. 5F, G). On the basis of the above findings, a simple model for how TGF-β induced myofibroblast program is activated via Prrx1-Smad3 complex is proposed (Fig. 5H).

**Super-enhancer mapping of ex vivo-cultured human colorectal CAFs identified a myofibroblastic subpopulation with *PRRX1* as an mTF.** As Prrx1 was shown to function as an mTF in murine myofibroblastic fibroblasts, *PRRX1* is probably an mTF in human CAFs. Indeed, Prrx1 induction alone was sufficient to reprogram normal fibroblasts into CAFs in vivo (Fig. 1G, Supplementary Fig. 3). These data further support the possibility of *PRRX1* being an mTF in at least certain subsets of human CAFs.

To confirm this hypothesis, we established nine ex vivo CAFs cultured from patients with advanced colorectal cancer and identified mTF candidates through super-enhancer mapping for each ex vivo-cultured CAFs. Specifically, H3K27ac ChIP-seq was performed for each of the nine CAFs; subsequently, ChIP-seq data were analyzed using the bowtie-MACS-ROSE pipeline[19,24,26,27] to identify enhancer and super-enhancer profiles for CAFs (Fig. 6A, Supplementary Fig. 16A–C, Supplementary Data 2). These enhancer and super-enhancer profiles for CAFs were compared with those of 171 cells obtained from ENCODE and dbSE databases (Fig. 6A).

Then, using super-enhancer profiles of the CAFs and ENCODE database, the nine ex vivo culture CAFs were characterized using principal component analysis (PCA). Notably, the nine CAFs were grouped into two distinct groups, one of which was near primitive mesenchyme-derived cells, including mesenchymal stem cells, and both groups were located far from embryonal stem cells or immune cells (Fig. 6B). The two clusters of nine CAFs based on super-enhancers were repeated using independent ATAC-seq (Assay for Transposase-Accesible Chromatin sequencing), which indicated that the two groups are epigenetically different and super-enhancer-based classification can be robust (Fig. 6C, Supplementary Data 3). Next, the super-enhancer-associated genes in these two groups (Cluster-A and Cluster-B groups) were examined using gene set enrichment analysis (GSEA) to improve our understanding of these groups. The genes associated with super-enhancers specific to the Cluster-A group were consistently enriched in gene sets related to

myofibroblasts, such as ECM remodeling, generation of high contraction force, high levels of TGF-β and VEGF signaling, and angiogenesis[12], whereas the Cluster-B group was characterized by senescence signature (Fig. 6D, E). Cluster-A showed significantly increased contraction (Fig. 6F), migration, and invasion abilities (Fig. 6G), consistent with GSEA results of the two CAF clusters. Using the 463 target genes of super-enhancers for the Cluster-A group, GSVA enrichment analysis of bulk RNA-seq expression data of TCGA colorectal cancer was performed to obtain Cluster-A signature, which showed a poor prognosis in patients with colorectal cancer with high Cluster-A signature (Supplementary Fig. 16D), which is consistent with invasion and migration assays.

Then, the mTF candidate in the nine CAFs was identified. Particularly, a core-regulatory circuit (CRC) set (Supplementary Data 4) composed of mTF candidates using the Coltron algorithm[25] was constructed, which integrated the super-enhancer profiles of each CAF with the MEME TF-binding motif database[35]. Using this approach, 23 and 29 mTF candidates (CRC TFs) were profiled for Cluster-A and Cluster-B, respectively. Using the RNA sequencing data of these CAFs, we confirmed that nine and six mTF candidates for Cluster-A and Cluster-B, respectively (expressed CRC TF), were highly and exclusively expressed between the two clusters (Fig. 7A). All lists of CRC TFs identified are shown in Fig. 7B. PRRX1 belonged specifically to the Cluster-A group as expressed CRC TF (Fig. 7C, Supplementary Fig. 12E–H).

Using the CRC sets constructed for each of the nine CAFs, the internal network structures of CRC in these CAFs were compared, and their similarities were calculated[36]. The nine CAFs were again subdivided into two identical subgroups corresponding to Cluster-A and Cluster-B groups (Fig. 7D, Supplementary Fig. 17) according to the enhancer pattern, suggesting that CRC sets of these CAFs account for such a subgrouping of these CAFs. Remarkably, *PRRX1* was identified as an mTF candidate for all Cluster-A CAFs exhibiting a myofibroblastic phenotype but not for Cluster-B CAFs (Fig. 7D right). Further, the potential target genes of common CRC of Cluster-A CAFs were enriched in functional pathways of ECM stiffness, TGF-β signaling, angiogenesis, and Wnt pathway, indicating that CRC of Cluster-A CAFs controlled myofibroblastic program (Fig. 7E).

To confirm *PRRX1* as an mTF of Cluster-A CAF, ChIP-seq for *PRRX1* was performed in CAF4, and the subsequent analysis showed a *PRRX1* binding of more than four times to the super-enhancer of Cluster-A compared with Cluster-B (Fig. 7F). Additionally, *PRRX1* Chip-seq also showed that the direct target genes of *PRRX1* were enriched in gene sets related to key

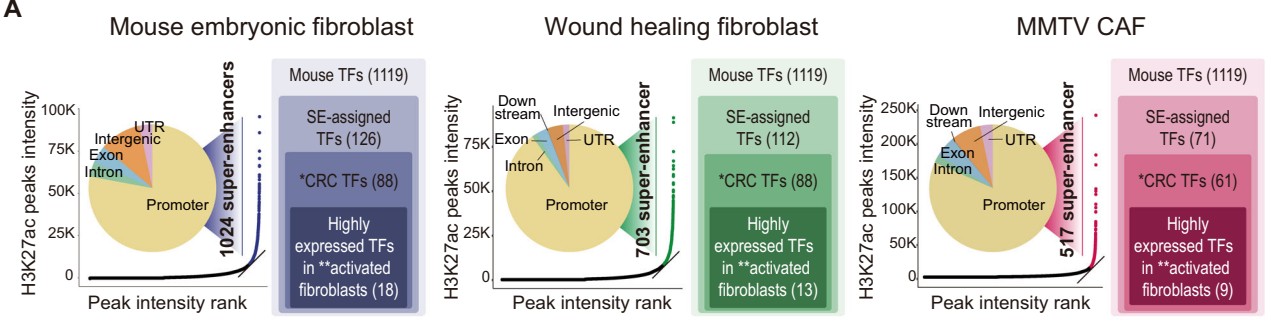

* Core regulatory circuitry (CRC) TFs : Self-loop and binding super-enhancers of other potential TFs
** Activated fibroblasts : clustered from single cell RNA-seq of mouse skin wound tissue (day 10) (Fig. 4D)

functions of myofibroblasts, such as wound healing, ECM organization, stress fiber assembly TGF-β signaling, etc. (Fig. 7G). To confirm the biological significance of *PRRX1*, shRNA-mediated knockdown of *PRRX1* was performed in these nine *ex vivo*-cultured CAFs. In response to *PRRX1* knockdown, four CAFs belonging to Cluster-A exhibited significantly increased apoptosis, whereas the remaining five were unaffected. Thus, four

Cluster-A CAFs showed dependence on *PRRX1* for their survival (Fig. 7H, Supplementary Fig. 18). Given that such a transcriptional addiction mostly occurs in response to an mTF[37,38], these results corroborate our finding that *PRRX1* is the mTF of Cluster-A CAFs.

Therefore, the nine *ex-vivo-cultured* CAFs were divided into two subsets: Cluster-A and Cluster-B. To determine whether such

**Fig. 4 Prrx1 of master transcription factor in activated murine fibroblast. A** Identification of candidate mTFs in three different fibroblast types. Total 18 highly expressed mTFs in MEF, 13 in wound-healing fibroblasts, and nine in MMTV-CAF were identified. **B** Among the mTFs, eight common mTFs in the three fibroblast types were detected. **C** Prrx1 was one of the eight common mTFs, highly expressed in activated fibroblasts. Gene ontology using target genes with three or more binding motifs for the eight common TFs showed enriched ECM stiffness, TGF-β pathway, angiogenesis, and chromatin remodeling pathway. **D** Clustering of single cells from mouse skin wound tissue (day 10 after post wound) revealed two clusters of the activated fibroblast group and inactivated fibroblast group, respectively (left). Prrx1 was the most significant highly expressed gene in the activated fibroblast group compared with the inactivated fibroblast group (right) among the candidate mTFs. The detail analysis results of these scRNA-seq were described in supplementary figure 10. **E** Prrx1 binds to SE with a higher rate than to TE in all three types of fibroblasts. **F** Super-enhancer peaks in a representative gene such as Col1a1 that are involved in ECM and contraction as well as Prrx1 in all three types of fibroblasts. There were other ChIP-seq profiles for Nfix, Klf13, Tead1, Fosl2 in Supplementary Fig. 11A–D. Source data are provided as a Source Data file.

clustering of CAFs is recapitulated in large-scale colorectal CAF populations, two colorectal cancer scRNA-seq datasets were integrated using the CCA algorithm[39,40] after which we performed a pseudo-time trajectory CAF analysis[41]. Indeed, CAFs presented two trajectories corresponding to Cluster-A and Cluster-B (Fig. 7I). *PRRX1* and myofibroblast-related signatures were highly enriched in the Cluster-A-like trajectory. Similar findings have also been reported by Li et al.[42]. In contrast, Terminal #2 (Cluster-B-like) was a fibroblast group in non-myofibroblastic fibroblasts. This trajectory pattern in colorectal cancer was similar to the other six tumor types (Supplementary Fig. 19).

The data strongly suggest that *PRRX1* governs a certain subpopulation of human colon CAFs that exhibit a myofibroblastic phenotype as an mTF.

**Prrx1 is a master regulator of myofibroblast activation during wound-healing in vivo**. Through a systems biology approach[43–45], including bioinformatics analysis and several in vitro experiments, Prrx1 was identified as a key mTF of fibroblasts and CAFs with myofibroblastic phenotype. We further confirmed whether Prrx1 is the master regulator of myofibroblasts in vivo. Myofibroblasts were first discovered in wound-healing tissues and were well defined in the wound-healing process. Hence, wound healing represents an optimal in vivo testing platform to evaluate the effect of Prrx1 on myofibroblasts. For this purpose, full-thickness excisional skin wounds were created in conditional *Prrx1* knockout and *Prrx1*-expressing mice using skin punch biopsy, and the Prrx1 effect on the wound-healing process was investigated.

First, fibroblast-specific *Prrx1* knockout (FS-Prrx1-KO) mice were generated by injecting sgRNA against Prrx1 into FSP1;cre-CAS9[EGFP] mice, and cutaneous wounds were created using skin excision (Fig. 8A, Supplementary Figs. 20, 21). Severe deficiency in myofibroblastic fibroblasts was observed in the FS-*Prrx1*-KO mice during wound healing. Consequently, the wound contraction was significantly reduced, and wound closure was severely delayed in the FS-Prrx1 KO mice (Fig. 8B). Microscopic examination of the wounded tissue in the FS-Prrx1-KO mice at 12 days postexcision revealed the thickness of both epithelial and dermal layers to be significantly decreased, accompanied by a significant decrease in the total number of fibroblasts compared with that in the control mice (Fig. 8C upper left). Particularly, the number of fibroblasts with myofibroblastic phenotype, SMA+/TNC+/PDGFRa+, was significantly decreased in the FS-Prrx1-KO mice (Fig. 8C bottom right); the number of PCNA+ fibroblasts exhibiting cell proliferation/survival was also markedly decreased. These results indicate that *Prrx1* deletion in fibroblasts reduces the myofibroblast subpopulations, which are key to the wound-healing process in vivo.

The effect of *Prrx1* overexpression in fibroblasts on the wound-healing process was further investigated in vivo. Fibroblast-specific Prrx1 overexpressing (FS-Prrx1-OE) mice were generated by

administering Dox to fibroblast-specific inducible Prrx1-expressing mice (Col1a2;rtTATetO7-Prrx1[Luc]) 5 days after skin injury (Fig. 8D). Wound contraction and healing rates were significantly higher in FS-Prrx1-OE mice than in control mice (Fig. 8E). These results suggest that wound healing is accelerated in FS-Prrx1-OE mice. Moreover, the population of proliferative fibroblasts was maintained in the FS-Prrx1-OE mice even after the completion of wound healing. In contrast, the population of fibroblasts normally disappears due to extensive apoptosis during the later stages of wound healing. Microscopic examination of wounded tissues 21 days after skin injury revealed that both epithelial and fibroblast-filled dermal layers remained abnormally thick in the FS-Prrx1-OE mice (Fig. 8F). Moreover, the number of fibroblasts that were positive for aSMA, TNC, PDGFRa, and PCNA was significantly increased in these wounded tissues compared with that in the control. Thus, *Prrx1* overexpression in fibroblasts accelerates the wound-healing process and induces fibrosis due to the persistent activation of fibroblasts.

In conclusion, it was confirmed that the presence or absence of *Prrx1* expression in fibroblasts significantly affects the myofibroblast subpopulation during wound healing, strongly suggesting that Prrx1 is the key mTF that determines the activation of fibroblasts with myofibroblast-like phenotype.

**Induced deletion of Prrx1 gene in fibroblasts eradicated established cancers in vivo**. The previous experiments highlighted Prrx1 as a potential therapeutic target for designing anticancer drugs that selectively inhibit the myofibroblastic CAF subpopulation. Hence, we investigated whether Prrx1 inhibition in fibroblasts exerts an anticancer effect in vivo.

Tumor cells (LLC1) were injected into fibroblast-specific CAS9-expressing (FS-CAS9) mice, and tumor growth was monitored using MRI scanning. When the tumor volume reached 30 mm³, Prrx1 deletion in fibroblasts was initiated by administering sgRNA against Prrx1 (sgPrrx1) to the FS-CAS9 mice (Fig. 9A). In this sgPrrx1-administered group, complete remission (CR) was induced in eight of ten mice and partial remission in the remaining two. In the control group (sgNS-treated), persistent tumor progression was observed in all ten mice. Although cisplatin has been reported to be partially effective against LLC1[46], it was only partially effective in two of ten mice (Fig. 9B). Notably, CR was induced in all ten mice in the combined sgPrrx1 and cisplatin therapy group. Microscopic examination of the tumor site in the CR-induced mice revealed no residual tumor cells (Fig. 9C right). Additionally, once CR was induced, it was maintained for at least 6 months (Fig. 9D). Thus, fibroblast-specific Prrx1 depletion induces long-term sustained CR in established cancers.

To understand the mechanisms that induce cell death in cancer cells after fibroblast-specific Prrx1 deletion, the frequency of apoptosis in spheroid cocultures composed of cancer cells and fibroblasts used in the above mouse experiments was evaluated. Consequently, apoptosis was observed much more frequently

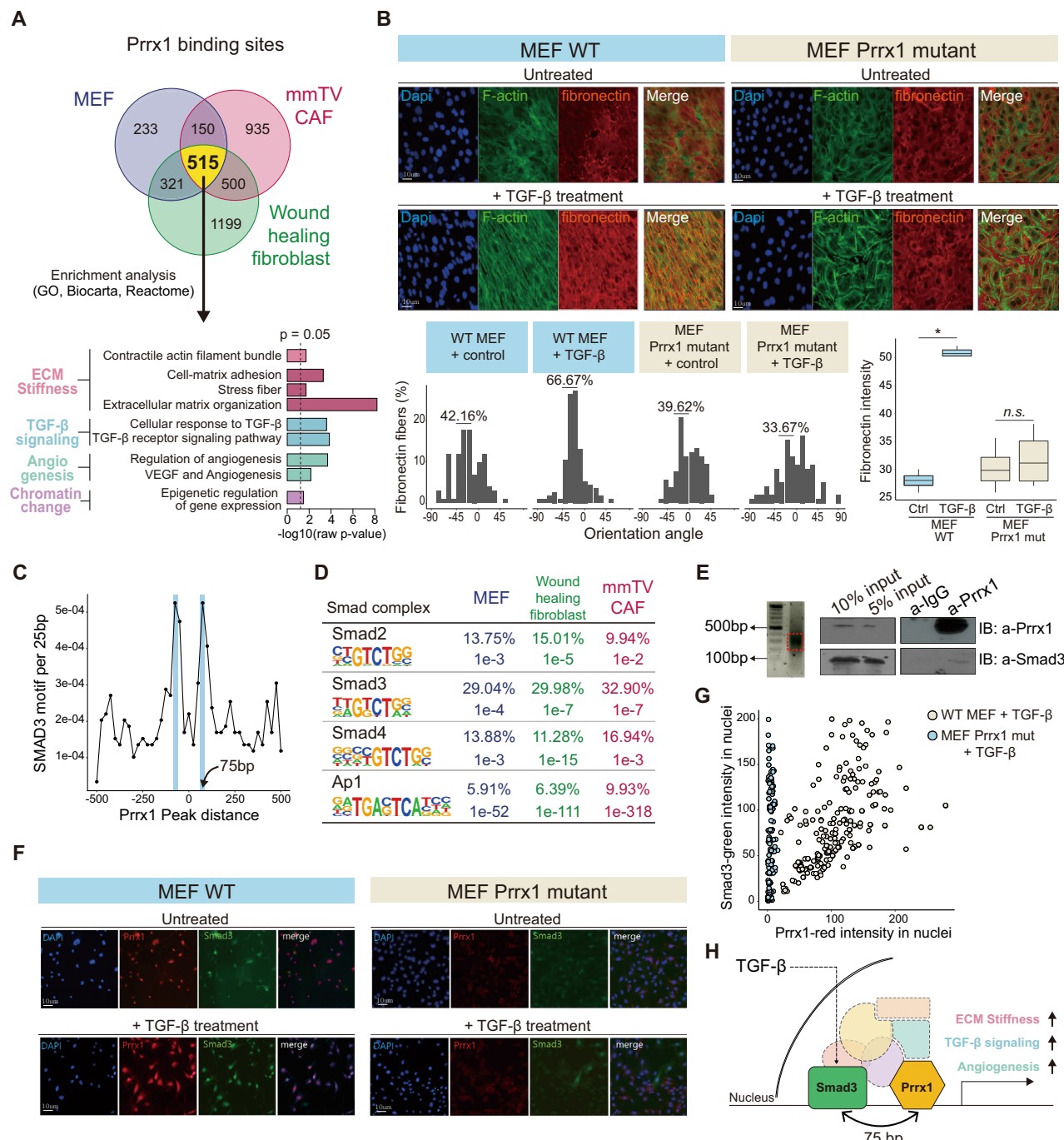

**Fig. 5 Prrx1/Smad3 complex executes a myofibroblastic program induced by TGF-β in fibroblasts. A** Total 515 common binding sites of Prrx1 in three different fibroblasts by Prrx1 Chip-seq. The genes with TSS within 1 kb from the 515 binding sites were enriched in various pathways such as ECM remodeling, TGF-β, and angiogenesis pathways, which were features of activated cancer-associated fibroblasts. **B** 3D matrix remodeling capacity of MEFs depending on the presence of Prrx1 was assessed by the alignment of the cytoskeleton (F-actin) and ECM deposition (fibronectin). TGF-β mediated Prrx1-induced remodeling of. No exhibitions affected Prrx1-induced ECM remodeling in TGF-β untreated groups (Red: fibronectin, Green: F-actin, Blue: DAPI). Box plot showing quantification of fibronection intensity (n = 6 independent experiments, *p < 0.0001 by two tailed t test). Data are presented as box-and-whisker plots indicating median (middle line) and minimum and maximum values (whiskers). **C** Predicted binding of Prrx1 and Smad3 at 75-bp interval between them using motif analysis. **D** Tables based on Prrx1 ChIP-seq motif analysis show that Prrx1 cobinds with Smad complexes. **E** Precipitation of Prrx1 after sonication of MEF genome 100 bp in length, confirmed that Smad3 was located near Prrx1. **F, G** Prrx1 and Smad3 were simultaneously expressed in the nucleus in the MEF WT group (Prrx1 normal expression) when TGF-β was treated with the MEFs, but not in the MEF Prrx1-mutant group (Impaired Prrx1 expression due to mutation). For (**E, F**): this experiment was independently repeated three times with similar results. **H** Schematic model of activated fibroblast induced angiogenesis, ECM stiffness, and TGF-β signaling via Prrx1-Smad3 complex. Source data and exact p values are provided as a Source Data file.

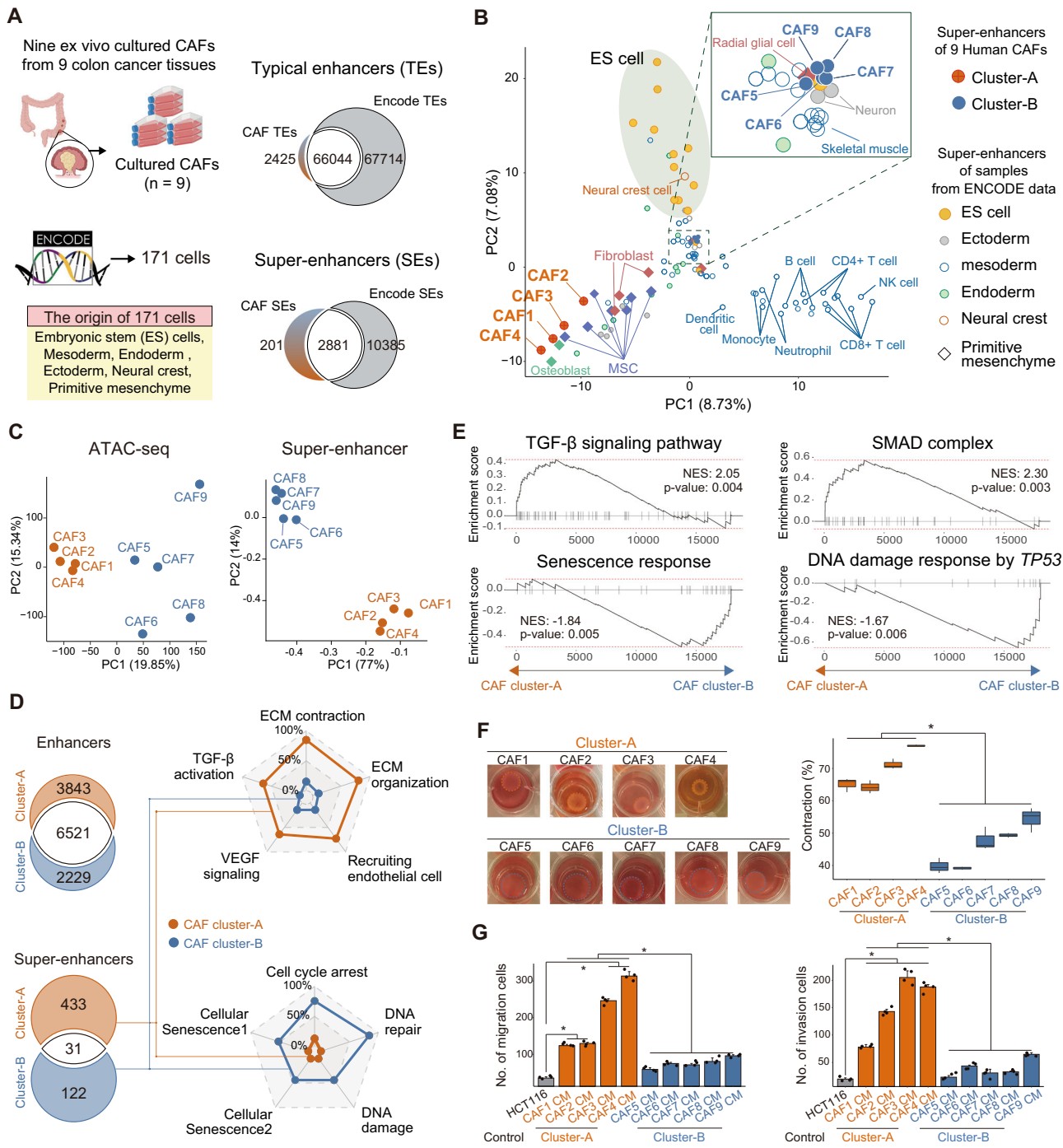

in the sgPrrx1-treated group than in the sgNS-treated group, and the effect was more significant when cisplatin was co-administered (Supplementary Fig. 22).

## Discussion

In this study, based on epigenetic analysis, including genome-wide super-enhancer mapping, we found that Prrx1 acts as a lineage-specific mTF of a subset of CAFs exhibiting a myofibro-blastic phenotype.

The contribution of Prrx1+ CAF to tumor behavior has been well documented in various studies[47–49]. Recently, Feldman et al.[49] reported that Prrx1 is critical for tuning CAF activation and has a significant effect on pancreas adenocarcinoma biology

and therapeutic resistance. These findings are similar to our findings.

This study showed that conditional Prrx1 depletion in fibroblasts successfully induced CR in established chemoresistant cancers in vivo. One reason for the potential application of Prrx1 as an effective therapeutic target for anti-CAF drugs may be the key role of Prrx1-modulated CAF subset in cancer progression. Additionally, as Prrx1 is an inherently lineage-specific mTF, targeting Prrx1 may not interfere with other tumor-suppressive subsets of CAFs. Another reason is the absolute dependence on Prrx1 of the Prrx1-modulated CAF subset for its survival (Fig. 7H). Such a transcriptional addiction, similar to oncogenic addiction[26,37,38,50,51], may be caused by disrupting the core-regulatory network in cells in response to an mTF loss[26]. Notably, targeting Prrx1 is least likely to produce side effects since its

**Fig. 6 Super-enhancer mapping of human colorectal CAF identifies distinct subpopulations. A** Typical enhancers (TEs) and super-enhancers (SEs) based on H3K27ac ChIP-seq were identified from 9 CAFs cultured from nine colon cancer tissues and 171 cells from ENCODE database. Many of them from the nine CAFs were overlapped with TEs and SEs from ENCODE data. **B** Principal component analysis (PCA) using SEs for the nine CAFs and 171 cells from ENCODE revealed two distinct subgroups of the nine CAFs into the 'Cluster-A' and 'Cluster-B'. **C** ATAC-seq was performed for all nine CAFs and PCA using the peak intensities of ATAC-seq also revealed two subgroups, which corresponded to the two CAF clusters identified using SEs from ChIP-seq. **D** GO analysis of SE-associated genes (genes containing TSS within 1 kb from the SE) showed that each of the two CAF clusters is related to distinct molecular functions. For the Cluster-A group, the enriched gene ontologies are ECM remodeling, VEGF and TGF-β activation, which are representative myofibroblastic functions, whereas cellular senescence signatures were increased in Cluster-B group. **E** RNA-seq was performed in the nine CAFs, then RNA-seq data were analyzed by gene set enrichment analysis (GSEA) for key cellular functions. The GSEA analysis of the mRNA expression dataset of these nine CAFs confirmed the result of GO analysis of SE-associated gene shown in (**D**). **F** Gel contraction assay revealed increased contraction abilities of CAFs in Cluster-A compared with Cluster-B. Box plots showing quantification of gel contraction ($n = 3$ independent experiments, $*p < 0.001$ by two tailed $t$ test). Data are presented as box-and-whisker plots indicating median (middle line) and minimum and maximum values (whiskers). **G** The invasion and migration of HCT-116 cancer cells were significantly enhanced by conditioned medium (CM) of CAFs in the Cluster-A group compared with Cluster-B ($*p < 0.001$ by two tailed $t$ test). Data are presented as the mean ± SEM; $n = 4$ independent experiments. HCT116 derived CM without CAF was used as a control. Source data and exact $p$ values are provided as a Source Data file. CAF, cancer-associated fibroblast; TSS, transcription start site; SE, super-enhancer.

---

expression is restricted to embryonic mesenchymal cells[52,53], which are rarely expressed in normal adult tissues, except for a small number of mesenchymal stem cells. We treated mice for more than 5 months with sgRNA against Prrx1 and observed no serious side effects. However, full evaluation using a systemic Prrx1 inhibition model would be needed to evaluate the true therapeutic index for Prrx1 targeting treatment.

This study expands our knowledge of CAF biology by contributing an innovative cancer mouse model that recapitulates desmoplasia by inducing Prrx1 expression in fibroblasts (Supplementary Fig. 6A). Additionally, a specific mouse model of postsurgical tumor recurrence was generated using fibroblast-specific Prrx1-inducible mice (Fig. 2D). Of note, both mouse models are immunocompetent; hence, they can be used as effective platforms for testing cancer immunotherapy. The success of both models is suspected to be because of the reprogramming of fibroblasts into CAFs in vivo by the mTF, Prrx1. These results indicate that it is plausible to create innovative tumor mouse models that replicate the TME of human cancers more accurately by manipulating genes encoding CAF-specific mTFs using genetically engineered mice. The overall results of the study are summarized in Fig. 10.

According to Van Groningen et al. study[23], Prrx1 is associated with tumor-intrinsic mesenchymal cell state change and resistance to therapy in neuroblastoma. It is plausible that Prrx1 may induce common mesenchymal programs in cancer cells, however, this requires further study.

In conclusion, this study identified Prrx1 as an mTF of a lineage-dependent CAF subpopulation that shows myofibroblast-like phenotype and plays a critical role in promoting tumorigenesis, metastasis, and cancer recurrence. The CAF subpopulation with Prrx1 as an mTF shows absolute dependence on Prrx1 expression, and conditional depletion of Prrx1 in fibroblasts in vivo induced and maintained CR of chemoresistant cancer. The systems biological approach used in this study to identify mTFs is an effective alternative method to resolve some of the current challenges in CAF research, especially those pertaining to CAF heterogeneity[54,55]. This study provided data to support the notion that cancers can be completely inhibited by suppressing CAFs using a genetically engineered mouse model. Finally, Prrx1 was reported as the mTF identified in CAFs that can serve as a potential therapeutic target to achieve CR of chemoresistant cancer.

## Methods
**Ethical regulations**. This study complies with all ethical regulations.

**Human research participants**. Tissue samples were prepared from patients with five cancer tissues including colorectal cancer ($n = 185$), stomach cancer ($n = 178$), lung cancer ($n = 80$), esophageal cancer ($n = 168$), and breast cancer ($n = 80$) confirmed as pathologic diagnosis. For the isolation of cancer-associated fibroblasts and normal fibroblasts from fresh colon cancer patient ($n = 9$) undergoing surgery at Samsung Medical Center, Seoul, South Korea. The samples were collected as standard of care. The samples were selected as randomly to minimize selection biases. The study protocol was approved by the institutional review board (SMC 2021-02-048) of Samsung Medical Center. These samples were collected from who provided informed consent for collection, storage, and distribution of samples and data for use in this study.

**FSP1;creCAS9EGFP transgenic mice**. FSP1-Cre-expressing mice (Stock No: 012641)[56], Rosa26-LSL-Cas9 knockin (Stock No: 024857) were obtained from The Jackson Laboratory. Mice were crossed to a C57BL/6 background. Mice used in this study were of both genders and within an age range of 1–4 months, except where noted for aging experiments. Summarized key resources are described in Supplementary Table 1. All studies and procedures involving animal subjects were approved by the institutional Animal Care and Use Committee (IACUC) of Laboratory Animal Research Center at Samsung Biomedical Research Institute (Protocol #20150424001, 20190607001, 2021012004). Animal Facility was provided by the Association for Assessment and Accreditation of Laboratory Animal Care (AAALAC).

**Col1a2;rtTA TetO7-Prrx1Luc transgenic mice**. The Prrx1 cDNA was PCR cloned into the bidirectional tetO7 vector S2f-IMCg at EcoRI and NotI sites, replacing the eGFP ORF. The resultant construct, Prrx1-tetO7-luc, was sequence confirmed, digested with KpnI and XmnI to release the bidirectional transgene and then used for injection of C57BL/6 pronuclei by the Macrogen Transgenic Facility. We ultimately obtained three founders from 25 pups after screening by tail genotyping using PCR as described below. These three founders were mated to Collagen Type 1 alpha-rtTA mice to screen for functional Prrx1-tetO7-luc founders. One founder failed to pass the transgene germline and one founder did not report inducible Prrx1 or luciferase expression. The remaining founder was used for all the experiments in this study. Prrx1 expression was activated in the bioluminescence imaging by administering doxycycline (Sigma) to the drinking water weekly [2 mg/mL] starting at the age of 3–5 weeks. The conditional Col1a2;rtTA TetO7-Prrx1Luc were generated by backcross. All procedures were performed in accordance with AAALAC guidelines and animals were housed in a pathogen-free environment. All studies and procedures involving animal subjects were approved by the institutional Animal Care and Use Committee (IACUC) of Laboratory Animal Research Center at Samsung Biomedical Research Institute (Protocol #20191017002).

**Animal ethics**. All animal procedures were IACUC approved and performed in Laboratory Animal Research Center at Samsung Biomedical Research Institute. C57B/6 and Balb/C mice were purchased from Jackson Laboratories. Mice were housed in the ventilated cage (max 5 mice/cage) supplied with food and water in a 12-h light/12-h dark cycle at 22 °C and 41% humidity. Maximal tumor burden permitted is 2000 mm³. In some cases, this limit has been exceeded the last day of measurement and the mice were immediately euthanized. The actual tumor size (even if larger than 2000 mm³) has been recorded and presented in the Article and Source data file. Mice were withdrawn from the study and sacrificed when the IACUC stipulated humane endpoint was reached: hunched appearance, more than 15% body weight loss, or tumor size in any dimension reaching 1.5 cm.

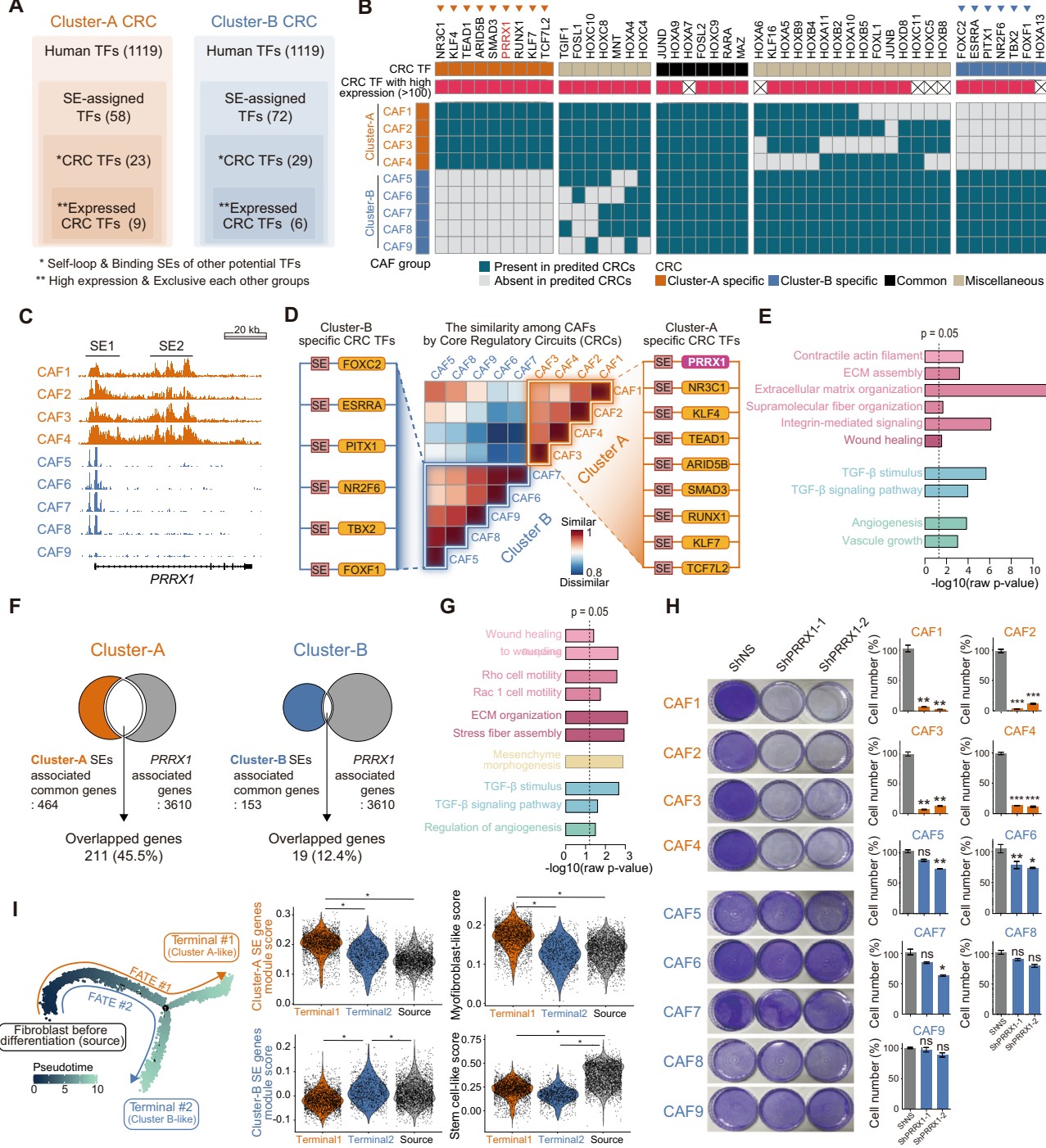

**PCR genotyping.** For genotyping PCR analysis, 1–2 mm sections of tail tip were dissolved in 0.5 ml of 50 mM Tris (pH 8.0), 100 mM EDTA, 0.5% SDS, and 0.5 mg/ml proteinase K (Roche) solution at 55 °C for at least one hour with vigorous shaking. The DNA was purified by Isopropanol followed by ethanol precipitation and then dissolved in 0.5 ml of Ultrapure water. The Collagen Type 1 alpha-rtTA and Prrx1-tetO7-luc transgenic lines were screened as described previously. The Prrx1-tetO7-luc line was detected with the following primers: mPrrx1-Luc3-F 5′-GGCCAACAGCATTGCC-3′ and mPrrx1-Luc3-R 5′- GAGCCTGGACCACTG-3′.

The Collagen Type 1 alpha-rtTA line was detected with the following primers: SV40_F1 5′- GGTTACAAATAAAGCAATAGCATCAC-3′ Col1a2_R2 5′- CAGCAGGAGGTTTCGACTAAG-3′. DNA was amplified using PCR and PCR products were resolved on a 2% agarose gel.

**Co-injection model.** 168FARN-Luc-GFP mouse cancer cells 1 × 10⁵ were injected subcutaneously in the left inguinal region of each NOD/SCID mouse. Where indicated, 168FARN 1 × 10⁵ cells were mixed with 3 × 10⁵ WT (SgNS) or

Prrx1-deficient MMTV-CAFs (SgPrrx1) prior to injection. Tumor growth was monitored by caliper. MKN28, HT29 or HCT116 5 × 10⁵ human cancer cells were injected subcutaneously in the left inguinal region of each NOD/SCID mouse. Where indicated, MKN28, HT29 or HCT116 cancer cells were mixed with 1.5 × 10⁶ WT or Prrx1-mutant MEFs prior to injection. Prrx1-mutant MEF cells were isolated from Prx-1lacZ mutant mice (Prrx1tm1Jfm/Mmmh, MMRRC_000347-MU, backcrossed to C57BL6), in which the *LacZ* gene was inserted in a frame within the homeodomain of Prrx1 exon 2. Tumor growth was monitored by caliper. MKN28 1 × 10⁶ human stomach cancer cells were injected subcutaneously in both flanks of each NSG(NOD/SCID/IL-2Rγnull) mouse. Where indicated, MKN28 1 × 10⁶ cancer cells were mixed with 3 × 10⁶ SNF32 (primary human stomach normal fibroblast) overexpressing -FLAG, -PRRX1a or -PRRX1b prior to injection. Tumor growth was monitored by caliper. Tumor volume was calculated using the formula: tumor volume = length × width × width × 1/2. All studies and procedures involving animal subjects were approved by the institutional Animal Care and Use Committee (IACUC) of Laboratory Animal Research Center at Samsung Biomedical Research Institute (Protocol #20150424001, 20210317001).

**Fig. 7 Prrx1 appeared as a master TF for the Cluster-A subpopulation of human colorectal CAF. A** Identification of master transcription factor (mTF) candidates constructing core-regulatory circuitry (CRC). Initially, mTF candidates were screened based on a super-enhancer profile derived from H3K27ac ChIP-seq. Then, mTF candidates with high mRNA expression were selected by RNA-seq of these nine CAFs. **B** Each TF in CRC between the Cluster-A group and Cluster-B group. Arrowheads indicate "expressed CRC TF" (CRC mTF with high mRNA expression and specificity for each group). Prrx1 belonged to the expressed CRC TF of the Cluster-A group. **C** Super-enhancer peaks in *PRRX1* gene loci were shown in the CAFs of Cluster-A. Contrastly, typical enhancer peaks were shown in similar *PRRX1* gene loci in the CAFs of Cluster-B. **D** The network similarities of each of the CRCs of the nine CAFs were calculated using a mathematical model based on graph theory. In detail, the components, the link profile, and degree distribution (the internal structure of links) of each of the CRCs were compared, and their similarities were calculated. **E** To identify phenotypes regulated by Cluster-A-specific CRC, target genes using motif analysis were inferred. These target genes had binding motifs of three or more TFs constituting Cluster-A-specific CRC. Then, gene ontology analysis revealed that these target genes were enriched in myofibroblastic cellular functions. **F** *PRRX1* showed more than four times binding to SE than TE by PRRX1 ChIP-seq in CAF4 cells of Cluster-A. **G** Enrichment analysis performed using the genes near the peak in PRRX1 ChIP-seq revealed that PRRX1 directly bound genes were significantly enriched in myofibroblastic phenotype-associated cellular functions. **H** Remarkable decrease of viable cell number in all Cluster-A CAFs in response to shRNA-mediated PRRX1 knockdown, with no effect in Cluster-B CAFs. Bar plot showing percentage of cell proliefartion. Data are presented as the mean ± SEM; $n = 12$ independent experiments (two tailed $t$ test: p *<0.05, **<0.01, ***<0.001). (**I**) Pseudo-time trajectory of 6,083 CAFs from the scRNA-seq of colorectal cancer tissue showed two differentiated fate (two tailed $t$ test, *$p < 2e{-}16$, $n = 6083$). Source data and exact p values are provided as a Source Data file.

**Orthotopic injection model**. Immunodeficient NOD/SCID mice were inoculated with $1 \times 10^4$ 168FARN cancer cells in the mammary fat pad. Next day, PBS (control), $5 \times 10^4$ MMTV-CAFs WT (SgNS) or Prrx1-deficient MMTV-CAFs (SgPrrx1) were injected into tail vein of each group of mice. After two weeks, sudden death began to be observed. All studies and procedures involving animal subjects were approved by the institutional Animal Care and Use Committee (IACUC) of Laboratory Animal Research Center at Samsung Biomedical Research Institute (Protocol #20150424001).

**LLC1 Subcutaneous injection into FSP1;creCAS9EGFP transgenic mice**. Prior to injection of cancer cells, sgRNA against Prrx1 was used to deplete prrx1 in fibroblasts. SgNS was used as a control. $5 \times 10^4$ LLC1-Luc-GFP cancer cells were injected subcutaneously into a flank of each mouse. LLC1 primary tumor growth was monitored by bioluminescence imaging in each group of mice on 7, 14, 21 and 35 days after LLC1 Luc-GFP injection. Mice were injected with 75 mg/kg D-luciferin (potassium salt, Biosynth L-8820) for in vivo bioluminescence imaging.

**LLC1 Subcutaneous injection into Col1a2;rtTA TetO7-Prrx1Luc transgenic mice**. LLC1 $5 \times 10^4$ cancer cells were injected subcutaneously into a flank of each mouse and then fibroblast-specific Prrx1 was induced by doxycycline for 2 weeks. The size of the tumor was measured by caliper for about a month. tumor volume = length × width × width × 1/2.

**Measurement of Col1a2 promoter activity in Col1a2;rtTA TetO7-Prrx1Luc mice**. Col1a2 promoter activity was monitored by bioluminescence imaging in each mouse after 12-mm biopsy punch skin wound. Skin wound was performed on the side of ventral of mouse. and then fibroblast-specific Prrx1 was induced by doxycycline [2 mg/mL]. After 3 days, bioluminescence Imaging was performed using a cooled charge-coupled device camera system (IVIS Imaging System 100; Xenogen/Caliper Life Sciences, Alameda, CA). Mice were I.P. injected with 75 mg/kg D-luciferin (potassium salt, Biosynth L-8820) for in vivo bioluminescence imaging.

**MC38 cell subcutaneous injection and Surgical removal of primary tumor in Col1a2;rtTA TetO7-Prrx1Luc transgenic mice**. Mc38-Luciferase cells $2.5 \times 10^5$ injection into right flank region of 8-week-old Col1a2-rtTA;TetO7-Prrx1 luciferase mice which were housed for several days duration. After tumor formation, when the tumor volume reached approximately 600 mm³ skin incisions and primary tumor removal surgery were performed under isoflurane anesthesia. The tumor is exposed to the incision through the opposite skin incision made through the tumor. The surgical incision was sealed with a suture. After 7 days, Animals that experienced primary tumor regrowth due to incomplete surgical tumor resection were not used for subsequent studies. Doxycycline induction [2 mg/mL] is performed after 7 days and followed up lung region by MRI imaging for 10 weeks. After the end of the experiment, MRI imaging and histological staining confirmed that the primary tumor was completely removed, precluding the possibility of remaining early transplanted cancer cells. Primary tumor growth was monitored with calipers and tumor volumes were calculated from length and width measurements using the formula: Tumor Volume (mm³)=0.5 × length × width².

**AOM/DSS induced colorectal cancer in WT mice**. 6–8-week-old wild-type mice (C57BL6 background) were injected intraperitoneally (i.p.) with 10.5 mg/kg Azoxymethane (AOM). After five days 2.5% DSS (MP Biomedicals, MW 36–50 kDa) was given in the drinking water over five days, followed by 14 days of regular water. This cycle was repeated four times (five days of 3% DSS and four days of 2% DSS) and Mice were sacrificed periodically after the last cycle. Colons

were removed, flushed with PBS, fixed as in 10% normal buffered formalin at 4 °C overnight and paraffin-embedded. Sections (5 μm) were cut stepwise (200 μm) through the complete block and stained with H&E.

**AOM/DSS induced colorectal cancer in Col1a2;rtTA TetO7-Prrx1Luc transgenic mice**. 6–8-week-old mice (Col1a2;rtTA TetO7-Prrx1Luc and Col1a2rtTA littermates on a mixed C57BL6 background) were injected intraperitoneally (i.p.) with 10.5 mg/kg Azoxymethane (AOM). After five days 2.5% DSS (MP Biomedicals, MW 36–50 kDa) was given in the drinking water over five days, followed by 14 days of regular water. This cycle was repeated four times (five days of 3% DSS and four days of 2% DSS) and Mice were injected intraperitoneally (i.p.) with 10.5 mg/kg Azoxymethane (AOM). After 5 days 1% DSS (MP Biomedicals, MW 36–50 kDa) was given in the drinking water over five days, followed by 14 days of regular water. Feeding mice with Doxycycline in food (cat, TD.01306) and mice were sacrificed after the last cycle. Colons were removed, flushed with PBS, fixed as "Swiss-rolls" in 4% paraformaldehyde at 4 °C overnight and paraffin-embedded. Sections (5 μm) were cut stepwise (200 μm) through the complete block and stained with H&E.

**Cutaneous wound healing in FSP1;creCAS9EGFP transgenic mice**. Adult FSP1;creCAS9EGFP are used for these studies. The wound-healing studies were performed in accordance with a protocol approved by the institutional Animal Care and Use Committee of Laboratory Animal Research Center at Samsung Biomedical Research Institute. Four 12-mm biopsy punch skin wound were placed on the side of dorsal of mice. The wound was covered with a transparent semi-occlusive dressing (Tegaderm, 3 M, Saint Paul, MN, USA) to prevent desiccation. Wounds were observed until closure or animals were killed. The excisional wounds were captured by camera on days 0, 3, 8, 10, 12, and 18. The wound areas were measured by using histological analysis and was described in Wound healing (%) and contraction (%) analysis methods. We confirmed that the sample size is statistically different based on previously published experiments[57,58].

**Cutaneous Wound healing in Col1a2;rtTA TetO7-Prrx1Luc transgenic mice**. Adult Col1a2;rtTA TetO7-Prrx1Luc are used for these studies. The wound-healing studies were performed in accordance with a protocol approved by the institutional Animal Care and Use Committee of Laboratory Animal Research Center at Samsung Biomedical Research Institute. Four 12-mm biopsy punch skin wound were placed on the side of dorsal of mice. The wound was covered with a transparent semi-occlusive dressing (Tegaderm, 3 M, Saint Paul, MN, USA) to prevent desiccation. Fibroblast-Specific Prrx1 expression was induced by the addition doxycycline in food after wound-healing initiation date and Wounds were observed until closure or animals were killed. The excisional wounds were captured by camera on days 0, 5, 10, 15. The wound areas were measured by using histological analysis and was described in Wound healing (%) and contraction (%) analysis methods. Investigators conducting functional measurements were blinded to the individual group during data collection. We confirmed that the sample size is statistically different based on previously published experiments[57,58].

**In vivo experimental strategy for therapeutic evaluation**. $1 \times 10^6$ LLC1-Luc-GFP cells were implanted into the 8-week-old FSP1;creCAS9EGFP mice. Cisplatin administration was started at a dose of 10.5 mg/kg i.p. twice per week when the tumors reached approximately 30 mm³ in volume. sgPrrx1 is administrated was started at a dose of MOI of 0.01 twice per week for 32 weeks duration, the mice were harvested, fixed in 10% normal buffered formalin and embedded in paraffin for subsequent analysis. Tumor size was measured every week using MRI imaging, and then the volume was calculated using using Paravision software version

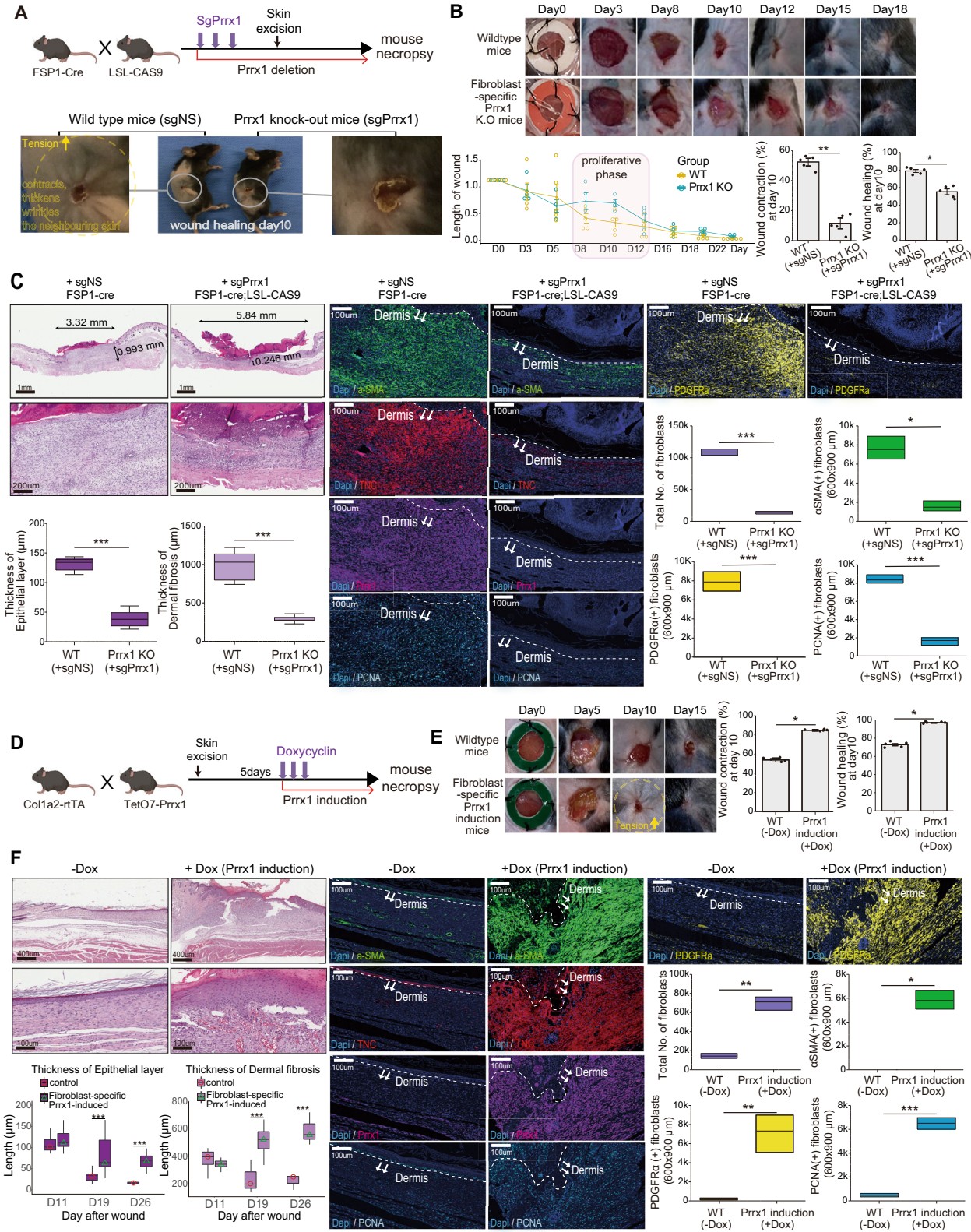

6.0 (Bruker-Biospin). All mice used in this project were maintained under defined conditions at the Animal Experiment Center of Samsung Biomedical Research Institute, and all animal experiments were approved by the Animal Care and Use Committee of Samsung Biomedical Research Institute.

**MRI hardware**. All in vivo MR imaging were carried on a 7 T/20 MR System (Bruker-Biospin, Ettlingen, Germany) equipped with a 20 cm gradient set capable of supplying up to 400 mT/m in 100 μs rise-time.

**MRI imaging**. A quadrature volume coil (35 mm i.d.) was used for excitation and receiving the signal. MR images were obtained from each mouse using a fast spin-echo T2-weighted MRI sequence (repetition time (TR)/echo time (TE) = 2500/45 ms; number of experiment (NEX) = 3; echo train length = 6; 130 × 130 μm² in plane resolution; slice thickness of 1 mm) with respiratory gating.

**MRI data analysis**. To quantify tumor volume from MR images, the area of tumor on each slice of the images were delineated and regions of interest (ROIs) were

**Fig. 8 Fibroblast-specific Prrx1 expression is critical in the expansion and maintenance of myofibroblast subpopulations during wound healing.**
**A** Illustration of the mouse model in which fibroblast-specific Prrx1 depletion. **B** Representative images of wound area from day 0 to day 18 post-operative wound in the fibroblast-specific Prrx1-knockout (sgPrrx1, $n = 6$) and control (sgNS, $n = 9$) mice, respectively. Delayed wound healing (%) and reduced wound contraction (%) due to Prrx1 deficiency on day 10 post-wound were shown by bar plot (bottom right). Data are presented as the mean ± SEM; $n = 5$ measurements (two tailed $t$ test, $*p < 0.0005$, $**p < 0.0001$). **C** Decreased myofibroblastic subpopulations during the wound-healing process due to a deficiency of Prrx1. H&E and immunohistochemical stainings of wound area on day 10 post-wound demonstrated the reduced epithelial and dermal layers due to Prrx1 deficiency (bottom left). Box plots showed the number of total fibroblast and fibroblasts expressing α-SMA, PDGFR α, and PCNA between WT and Prrx1 KO mice (bottom right). ($n = 3$, two tailed $t$ test: $*p < 0.005$, $**p < 0.0005$, $***p < 0.0001$). **D, E** Induction of Dox-mediated fibroblast-specific Prrx1 expression in mice 5 days after skin punching. Representative images of wound area from day 0 to day 15 after wounding in fibroblast-specific Prrx1-induced (+Dox, $n = 12$) and control (−Dox, $n = 11$) mice were shown. Bar plots indicated an increased wound-healing rate (%) and wound contraction (%) due to Prrx1 overexpression on day 10 post-wound. Data are presented as the mean ± SEM; $n = 5$ measurements (two tailed $t$ test: $*p < 0.0001$). **F** H&E and immunohistochemical images of representative wound area on day 30 post-wound were visualized using ImageJ Fiji (DAPI [hematoxylin]-blue, aSMA-green, TNC-red, PDGFRa-yellow, Prrx1-magenta, PCNA-cyan) (−Dox, $n = 11$ mice, +Dox, $n = 12$ mice). The thickness of both epithelial and dermal layers was increased in fibroblast-specific Prrx1 overexpressing mice (bottom left). Box plots showed the number of total fibroblast and fibroblasts expressing α-SMA, PDGFR-α, and PCNA in the wounded tissue (bottom right) ($n = 3$ measurements, two tailed $t$ test: $*p < 0.005$, $**p < 0.0005$, $***p < 0.0001$). Source data and exact $p$ values are provided as a Source Data file.

---

drawn around the tumor volumes using Paravision software version 6.0 (Bruker-Biospin).

**Immunohistochemistry and tissue samples.** Tissue samples were prepared from patients with five cancer tissues such as colorectal cancer ($n = 185$), stomach cancer ($n = 178$), lung cancer ($n = 80$), esophageal cancer ($n = 168$), and breast cancer ($n = 80$) after ethical approval by the institutional review board (SMC 2021-02-048) of Samsung Medical Center (Seoul, Korea). Immunostaining for the PRRX1 protein was performed using an anti-PRRX1 monoclonal antibody (LS-bio LS-C336798). There are Tissue microarrays (TMA) samples with clinical data including age, sex, tumor size, depth of invasion (T), nodal status (N), metastasis (M), overall survival (OS) and disease-free survival (DFS). Staging based on TNM classification was applied according to guidelines from the 2010 American Joint Committee on Cancer staging manual. Immunohistochemical staining was estimated according to our previous studies[4]. Sections were deparaffinized with xylene, then rehydrated with an alcohol, and finally hydrogen peroxide in methanol. The sections were then treated with TE buffer (Tris 10 mM and EDTA 1 mM, pH9.0) for antigen retrieval. To block non-specific staining, each of sections were treated with 4% skim milk in PBS with 0.1% Tween 20 (PBST) for 30 min and each of sections were incubated with primary primary antibodies: anti-Prrx1 (Origene TA803116, 1:400), anti-aSMA (DAKO M0851, 1:200), anti-PCNA (Abcam ab29, 1:1000), Anti-Tenascin C (Abcam ab108930, 1:100), anti-Prrx1 (LS-bio LS-C336798, 1:500), anti-PDGFRa (Cell signaling 3164, 1:100) antibody in TBST containing 4% skim milk for 60 min at room temperature. Following three washes with PBS buffer, sections were then incubated with an anti-mouse/rabbit-specific protein kit (Envision Plus, Dako, Carpinteria, CA, USA) at room temperature. The chromogen was used 3-amino-9-ethylcarbazole (AEC, SK-4205, Vector, Burlingame, CA, USA). We counterstained sections with Meyer's hematoxylin and generated the virtual slide images using Aperio® AT2 virtual slide scanner (Leica, Wetzlar, Germany). We measured Immunohistochemical scores semi-quantitatively[4]. In summary, we measured the intensity and the proportion of staining positive cells as follows: [Score 1], weak staining in <50% or moderate staining in <20% of stromal cells; [Score 2], weak staining in ≥50%, moderate staining in 20–50% or strong staining in <20%; [Score 3], moderate staining in ≥50% or strong staining in ≥20%.

**Image visualization.** Image visualization processing step and detailed protocol were described in a previous report[56]. Visualization was performed by converting the coregistered images into pseudo-color images using an Image J Fiji software. In Image J Fiji, the images were processed with Plugin, color deconvolution for separating of DAB/AEC and hematoxylin staining signal.

**Wound healing (%) and contraction (%) analysis.** To estimate the wound healing and contraction rate (%), wound images were acquired on the day of the wound (day 0) and the corresponding day (Fig. 7). Wound contraction and healing rates were quantified following a previous study[57,58]. The rate of wound healing was quantified as follows: (1 -(current wound area/original wound area)) x 100%, wound contraction rate(%) was calculated with following formulas, respectively: (1 - (current wound area epithelialized area)/original wound area) x 100%.

**Hanging drop spheroid coculture system.** Remove the lid of a 60 mm petri dish and place 4 ml pf PBS in the bottom of dish (act as a hydration chamber). Cancer cells and fibroblasts were detached and resuspend with complete medium. Count the cells using a hemacytometer. Invet the lid of a 60 mm petri dish and use a auto-pipette to deposit 10 μl drop onto the bottom of the lid of dish. $1 \times 10^3$ GFP tagged cancer cells ta and $3 \times 10^3$ fibroblasts (1:3 ratio) were in each 10 μl drop, at least 20–25 drops per dish. Invert the lid onto prepared PBS-bottom of dish and incubate at 37 °C, 5% $CO_2$ humidity culture incubator. after a day (~24 h) monitor the drops and check cell aggregates have formed. The aggregated spheroids were collected using 5 ml of complete medium, transferred to 100 mm petri dish, incubated for a week on a shaker in 37 °C, 5% $CO_2$ culture incubator. For H&E and IHC staining, the spheroids were transferred into 15 ml falcon tube, washed twice with PBS, fixed with 4% paraformaldehyde(PFA) for 12 h at 4 °C. For isolation of GFP tagged cancer cells, the spheroids were dissociated by incubating and resuspending in 1 ml of 0.25% trypsin. GFP tagged cancer cells were sorted by FACS Aria III.

**PRRX1 promoter cloning.** Genome DNA was isolated from human Stomach CAFs[4] using a DNA extraction kit (QIAGEN) and dissolved in water. Based upon our findings regarding the location of the transcriptional start site, a 1282 bp fragment from −1240 to +42 in the 5′-flanking region of the Prrx1 gene was generated by PCR. The PCR products were gel-purified, digested with MluI and ClaI, and subcloned into the pLVeGFP.Basic vector. All the sequences of the cloned promoter region were confirmed by DNA sequencing.

**GFP Low / High in CAF FACS sorting.** SCAF cells were infected lentivirus containing human-PRRX1 promoter specific GFP reporter. FACS sorting was performed to isolate FITC population. GFP high (PRRX1 high) and GFP Low (PRRX1 low) cells were sorted using by FACS Aria III.

**Plasmid construction for CRISPR/Cas9 delivery.** For efficient CRISPR/Cas9 system delivery, we screened several guide RNA(gRNA)s targeting the first exon of mPrrx1 gene and found the most efficient gRNA. For this study, we used that gRNA targeting 5′- GAGCGGCAACCGGCGCTGGG-3′ in Prrx1 gene unless otherwise described. We generated lentiCRISPRV2_tdTomato plasmid by inserting P2A(ribosomal skipping self-cleaving peptide)-tdTomato construct into BamHI restriction sites of lentiCRISPRV2 plasmid (Addgene, #52961) to label transduced cells. Then, we replaced the stuffer sequence in lentiCRISPRV2_tdTomato with sgRNA sequence for Prrx1 targeting to generate lentiCRISPRV2_tdTomato_sgPrrx1. Also, we generated pLKO.1_Puro_sgPrrx1 plasmid by inserting Prrx1 sgRNA sequence into pLKO.1_puro plasmid (Addgene, #8453) after digestion with AgeI and EcoRI. For labeling of transduced cells, we also added tdTomato followed by P2A sequence in frame with puromycin resistant gene using Gibson assembly.

**SgRNA formulation for delivery in vivo system.** sgNS and sgPrrx1 were produced by co-transfection of viral production plasmids in 10 cm culture plates containing HEK-293T cells per plate. DNA mixture for each plate was prepared as follows: 1 mL of plain Dulbecco's modified Eagle's medium (DMEM) with pLKO.1_Puro_NS or pLKO.1_Puro_sgPrrx1 expression vector, 7.5 μg of packaging plasmid, PSPAX2, and 2.5 μg of enveloping plasmid. DNA mixture was added to 500 μL of plain DMEM containing 40 μL of lipofectamine (lipo; Invitrogen, catalog 11668027) and incubated at room temperature in the dark for 20 min. The DNA/lipo mixture for each sample was further added to each plate containing HEK-293T cells and incubated for 16 h at 37 °C. The medium was replaced with 10 mL of fresh complete DMEM, and the plates were incubated for 24 h. Viral supernatant was harvested and filtered through a 0.45 μm MF-Millipore™ Membrane Filter (Millipore; catalog HAWP04700) and concentrated either by ultracentrifugation at 10,000 $g$ for 1–4 h or with Lenti-X Concentrator reagent for 30 min to a final concentration of 10× or 20× as per the instruction manual (CloneTech; catalog #PT4421-2). The supernatant was resuspended in phosphate-buffered saline (PBS). To test the feasibility of gene editing with lentiviral sgRNA delivery

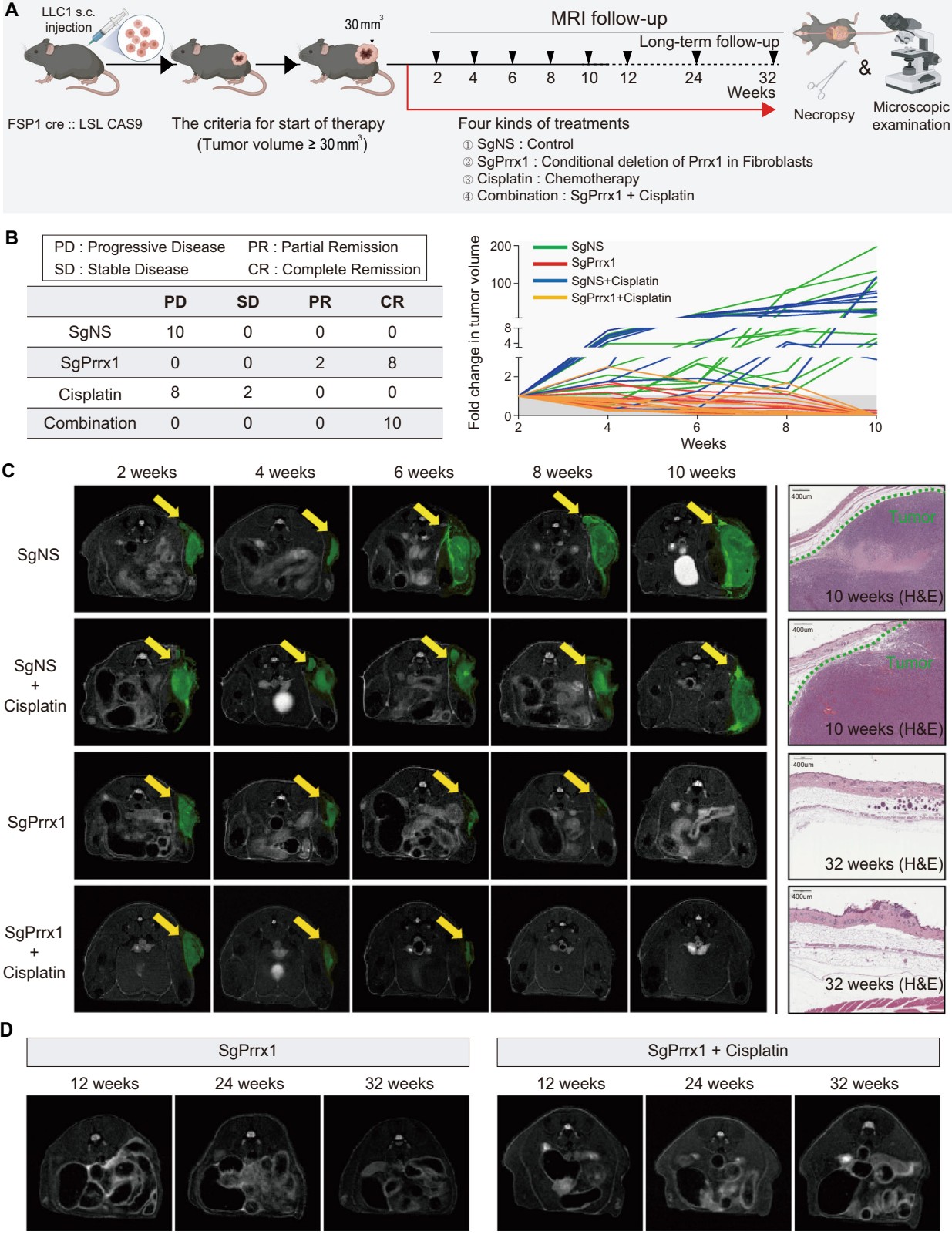

**Fig. 9 Induced depletion of Prrx1 in fibroblasts alone induced long-term complete remission of cisplatin-resistant cancers in immunocompetent mice.**
**A** LLC1-Luc-GFP cancer cells were transplanted subcutaneously into transgenic mice (FSP1;creCAS9EGFP). Four kinds of treatments were administered (n = 10 for each of four groups; sgNS, sgPrrx1, cisplatin, and sgPrrx1+cisplatin) after tumor formation (tumor size > 30 mm3). **B** Comparison of tumor response in each treatment. Tumor growth curves; n = 10 mice/group. **C** Tumor volume was monitored using MRI for 10 weeks. Representative images of tumor growth monitored using MRI. The yellow arrows indicate tumor (green). Representative H&E images of primary tumor site in each group. **D** Additional MRI follow-up was performed to confirm long-term complete remission of tumors in the two groups (sgPrrx1 and sgPrrx1+cisplatin). Source data are provided as a Source Data file.

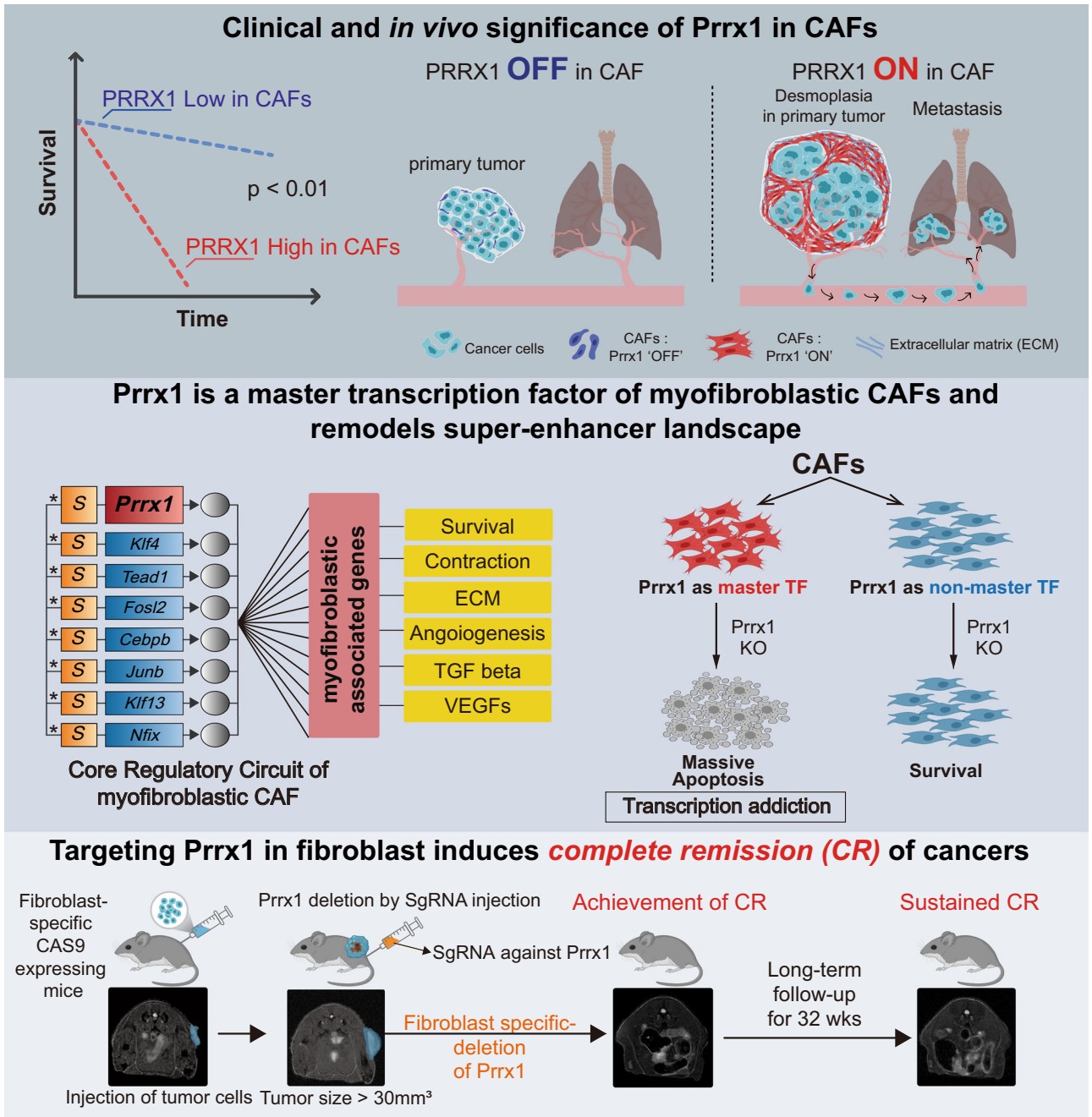

**Fig. 10 Summary of the overall findings.** *PRRX1* is the master transcription factor (mTF) that determines the lineage of highly pro-tumorigenic cancer-associated fibroblasts (CAFs) with a myofibroblastic phenotype through remodeling a super-enhancer landscape. Targeting the mTF, *PRRX1* belonging to the CAF subgroup with a myofibroblastic phenotype may be a promising cancer treatment strategy.

(Supplementary Fig. 20H), we designed seven unique sgRNAs targeting the gene *Prrx1*. After determining the most potent sgRNA in an in vitro assay in the mouse-derived wound-healing fibroblast cells (Supplementary Fig. 20F), titering of viral particles was done with qPCR and western blot analysis.

**Virus delivery into the skin**. The formulation was subsequently administered to Cas9-expressing mice. Two-month-old mice were anesthetized using isoflurane and set up in a biosafety cabinet. Mice were injected intradermally around the wound area with 100 μL concentrated sgRNA once every 3 days from the wounding start. For effective delivery, the intraperitoneal injection (IP) was performed simultaneously. In our study, we observed no significant change in the gross measure of health in mice.

**Detection of nuclease-induced genomic mutations**. Genomic DNA was extracted using DNeasy Blood and Tissue extraction kit (Qiagen) following the manufacturer's instructions. The target locus was amplified by PCR using High

Fidelity DNA Polymerase (NEB). Oligos are listed in the Supplementary Information. To evaluate indels caused by cleavage of one gRNA, purified PCR products were Sanger sequenced and analyzed using the SYNTHEGO ICE software.

**Invasion, migration assay**. Migration and invasion assay were performed as previously described[4]. In vitro migration and invasion assays were performed using Costar Transwell chambers (pore size 8.0 um, Cat 354234, Coring, NY) Cancer cells were incubated with CM (conditioned medium) from appropriate fibroblasts for 2–3 days. Cells were washed twice with PBS, detached and resuspended in a complete medium. After fibronectin coating and solidification of Matrigel, cancer cells ($5 \times 10^4$) with 200 μl of complete medium were seeded into upper chamber. Lower compartment was filled with complete medium containing 0.1% BSA. After 17 h in the cell culture incubator, cells were fixed with methanol and stained with H&E solution.

**Sphere forming assay**. Cancer cells were incubated with CM from fibroblasts for 2–3 days. Cells were detached, washed twice PBS and seeded into ultra-low-attachment 6-well plates (Cat: CLS3471, Corning, NY, USA). These cells were cultured with DMEM/F-12 supplemented with B27 (Invitrogen, Carlsbad, CA, USA), 10 ng/mL of EGF, and bFGF (Invitrogen, Carlsbad, CA, USA). The sphere forminng medium containing CM from fibroblasts was changed every 2 days until sphere formation was observed(about 2 weeks). After 2 weeks, spheres were counted.

**Apoptosis assay**. Apoptosis assay was performed according to the manual of BD Annexin V-APC apoptosis detection Kit (Cat. 550474, BD Biosciences, Franklin Lakes, NJ USA). CAFs were detached, washed twice with PBS and resuspended in 1×Annexin binding Buffer. These CAFs were stained with annexin V conjugated with APC at room temperature in the dark for 15–20 min. After incubation, cells were washed twice with 1 × Annexin binding Buffer and resuspended in 1 × Annexin binding Buffer containing Propidium Iodide and stained cells(apoptotic cells) were detected by flow cytometry immediately.

**Gel contraction assay**. Gel contraction assay was performed using a cell contraction assay kit (Cat: CBA-201, Cell Biolabs, Inc, San Diego, CA) according to the manufacturer's instructions. CAFs were trypsinized, washed twice with PBS, resuspend with complete medium(10ul) mixed with collagen and 5xDMEM, total volume 500 ul. This 500 µl of cell-collagen mixture was plated into 24-well plates and incubated at 37 °C for 2 h (for collagen polymerization). Gel contraction was initiated by releasing the gel from the side of the well, and changes in the collagen gel size were observed. The complete medium was changed every 2 days.

**Cytokine array**. Control (ShNS) or PRRX1-knockdown (ShPRRX1) CAFs with the same number of cells were seeded on 60-mm dishes. After incubation for 24 h, cells were washed twice with PBS, changed serum-free medium and incubated for 2 days. After incubation for 24 h, cells were washed twice with PBS. Replaced a serum-free medium, and cells were cultured for 2 days. After 2days, This conditioned medium was collected. Cytokine assays were performed using the cytokine array kit with the collected CM (cat. no. ARY005B; R&D Systems, Inc.) according to the manufacturer's instructions. The resulting image data was obtained and analyzed by ImageQuant LAS500.

**3D invasion assay**. 3D invasion assay was performed using the Oris™3D Embedded Invasion Assay kit (cat, EIA1). In brief, GFP tagged cancer cells were suspended in a collagen layer surrounding a circular (2 mm) central cell-free collagen -detection zone. After 1 h, the stoppers were removed from the 96well plate and the cell-free detection zone was filled with the same composition of collagen matrix. After 1 h, complete medium was added to each well and incubated for 24 h. The CM (conditioned medium) from fibroblasts was changed every 2 days. Each time the CM was replaced, 0.5% serum was added. After 2 weeks incubation, invading cells moved into the detection zone.

**3D extracellular Matrices**. 3D ECMs were generated following a previous study[59]. For 60 mm cell culture dish, $2 \times 10^5$ MEFs were seeded on gelatin cross-linked glass coverslip. Cells were cultured with MEF medium supplemented with 50 µg/ml ascorbic acid and, where indicated,10 ng/ml TGFb or 150 ng/ml SHH. To raise ECMs deposition by the cultured MEFs, culture medium was replaced every 2 days for a week. After a week, cells were washed with 1X PBS and fixed with 4% PFA for immunofluorescence imaging.

**qPCR**. Total RNA was extracted using by RNeasy kit (#74104, Qiagen). cDNA was synthesized from 2 g of total RNA using High capacity cDNA reverse transcription kits (#4368813, Applied Biosystems, Foster City, CA, USA). For PCR reaction, SYBR green PCR master mix (Applied Biosystems) was used with each primer set and the reaction was performed using a ABI 7900 HT Fast Real Time PCR system (Applied Biosystems, Foster City, CA). GAPDH was used as an internal control. qPCR cycling conditions: 1 step for 10 min at 95 °C followed by 40 cycles at 95 °C for 15 s and at 60 °C for 1 min. Fold-change (FC) in each gene expression was analyzed using the delta delta Ct method. Each measurement was performed in triplicate. The primer sets are described in Supplementary Table 1.

**Cell lines**. Human stomach cancer cell line MKN28, SNU668, and the colon cancer cell line HT29, HCT116 were purchased from the Korean Cell Line Bank that performs cell line characterizations using DNA fingerprinting analysis and passaged in our laboratory for fewer than 6 months. Cells were maintained in DMEM medium supplemented with 10% FBS and 1% antibiotics. Mouse breast cancer cell line 168FARN was provided by Dr. Jing Yang[60] (University of California, San Diego), mouse colon cancer cell line MC38 was provided by Dr. Hyon E. Choy[61] (Chonnam National University), Lewis Lung carcinoma cell line LLC1 was purchased by ATCC. All cells were cultured in DMEM medium supplemented with 10% FBS and 1% antibiotics.

**Human samples for ChIP-seq**. Ex vivo culture was carried out as described previously[4]. Human stomach and colon tumor specimens were obtained from patients who undergoing surgery at Samsung Medical Center of SungKyunKwan University of Medicine (Seoul, Korea). An experienced pathologist grossly examined and obtained representative samples of the tumor tissues (CAF, cancer-associated fibroblasts) and distal normal tissues (NF, normal fibroblast). The numbers after# are the order of the primary cells isolation. In detail, fresh tissues obtained from two different areas were cut into small pieces and minced with scalpels in a culture dish. Samples were enzymatically dissociated in 20 ml of DMEM complete medium containing collagenase I in a 37 °C incubator for 12−15 h using an orbital shaker. After digestion, samples were centrifuged at 700 rpm for 5 min. Cells were resuspended with DMEM complete medium were collected from the supernatant by centrifugation at 800 rpm for 8 min, washed twice with 1XPBS, and cultured in DMEM media supplemented with 10% FBS and 1% antibiotics. H3K27ac ChIP-seq for 'Cluster A' CAFs twice and 'Cluster B' CAFs thrice were performed.

**Mouse samples for ChIP-seq**. Mouse wound-healing fibroblasts were isolated from fresh tissue of dorsal wound area at day 10 post-wound in mice. the tissues were cut into small pieces, dissociated with 20 ml of DMEM complete medium containing collagenase I in a 37 °C incubator for overnight using an orbital shaker. After digestion, samples were centrifuged at 700 rpm for 5 min. Cells were resuspended with DMEM complete medium were collected from the supernatant by centrifugation at 800 rpm for 8 min, washed twice with 1XPBS, and cultured in DMEM media supplemented with 10% FBS and 1% antibiotics. MMTV-CAFs isolation was carried out as described previously(Ref). MMTV-CAFs were isolated from transgenic mice MMTV-PyMT, a breast cancer mouse model. Briefly, mouse tumor tissues were washed twice in PBS containing antibiotics, chopped into small pieces and minced with scalpels. Minced samples were enzymatically dissociated at 37 °C in 20 ml of DMEM medium containing collagenase/Dispase (Roche). After digestion, dissociated samples were centrifuged, washed twice in PBS and cultured in DMEM media supplemented with 10% FBS and 1% antibiotics. MEFs (Mouse embryonic fibroblasts) were isolated from embryos harvested at E10–13 using the same method. Prrx1 Chip-seq was duplicated and IgG was used as a normal control.

**Library preparation and sequencing for Chip-seq**. Cells were chemically cross-linked for 10 min at room temperature by the addition of one-tenth of the volume of fresh 11% formaldehyde solution to the cell growth medium. The crosslinking was stopped by incubating with 0.125 M glycine for 10 min. Cells were rinsed twice with 1xPBS and then the supernatant was aspirated. Cells were harvested using scraper and flash frozen in liquid nitrogen. Frozen cell pellet was stored at −80 °C prior to use.

Cells were resuspended with hypotonic lysis buffer(10 mM HEPES-KOH, pH7.8, 10 mM KCl, 1.5 mM MgCl2, with fresh protease inibitors) and incubated on ice for 10 min. Cells were centrifuged at 9000 rpm for 1 min. The pellet was resuspended with nuclei lysis buffer (1% SDS, 50 mM Tris-HCl,pH8.0, 10 mM EDTA, with fresh protease inhibitors) and sonicated for 6 times for 20 sec (1.0 s on/ 0.5 s off). We used virsonic550 ultra sonicator and samples were kept on ice at all times. DNA size(100–300 bp) was checked using 10 µl of sonicated sample. Sonicated lysate were diluted with ChIP dilution buffer(20 mM Tris-HCl,pH8.0, 100 mM NaCl, 2 mM EDTA, 0.5% Triton X-100). The resulting whole cell lysate was incubated overnight at 4 °C with agarose A/G beads that had been pre-incubated with appropriate antibody. 50 µl of agarose A/G beads were blocked with 0.1% BSA(w/v) in PBS. Beads were bound with 5ug of the indicated antibody.

Beads were washed twice 1x with the sonication buffer(1x ChIP lysis buffer), twice with Wash buffer1 (1xTE,pH7.4, 0.1%SDS, 0.1% NaDeoxycholate, 1% Triton X-100), twice with Wash buffer2(Wash buffer1 with 0.5 M Nacl), twice with Wash buffer3(1xTE, 0.25 M Licl, 1% NP40, 1% NaDeoxycholate), one time Wash buffer4(1xTE, 0.1% Triton X-100), and one time with 1xTE.

Bound sample was eluted from the beads(1% SDS, 0.1 M NaHCO₃) by heating at 65 °C for 2 h with occasional vortexing. Crosslinking was reversed overnight at 65 °C. (also perform reverse crosslink for input DNA control). Next day, Rnase A was added to each sample and the samples were incubated at 37 °C for 2 h, then proteinase K was added to each samples and the samples were incubated at 55 °C for 2 h. DNA was purified with phenol chloroform extraction and ethanol precipitation.

The construction of library was performed using NEBNext® UltraTM DNA Library Prep Kit for Illumina (New England Biolabs, UK) according to the manufacturer's instructions. Briefly, the chipped DNA was ligated with adaptors. After purification, PCR reaction was done with adaptor-ligated DNA and index primer for multiplexing sequencing. Library was purified by using magnetic beads to remove all reaction components. The size of library was assessed by Agilent 2100 bioanalyzer (Agilent Technologies, Amstelveen, The Netherlands). High-throughput sequencing was performed as paired-end 100 sequencing using HiSeq 2500 (Illumina, Inc., USA).

**Histone modification markers and Prrx1 ChIP-seq**. All three types of mouse fibroblasts and nine human CAFs ChIP-seq data were trimmed using

Trimmomatic version 0.39 (Bolger et al., 2014) with the "PE -phred33 LLUMI-NACLI TruSeq3-PE.fa:2:30:10 LEADING:3 TRAILING:3 SLIDINGWINDOW:4:15 MINLEN:36". The trimmed reads were aligned to the mm9 reference genome of the mouse genome using only annotated chromosomes 1–19, chrX, chrY, and chrM or the hg19 of the human genome using only annotated chromosomes 1–22, chrX, chrY, and chrM using Bowtie[62]. The mapped reads were indexed and sorted by samtools version 1.7[63]. The duplicate reads were removed using Picard tools (version 2.25.0). We next performed peak calling with corresponding input control using MACS2 peak caller tool with parameter -p 1e-5 for sharp peaks calling and with parameter --broad --broad-cutoff 0.1 for broad peaks calling.

**Identification of Super-enhancer**. The sharp peaks of H3K27ac data were found to be enhancers. Then, H3K27 ChIP-seq super-enhancers were detected by using the ROSE2 (Rank Ordering of Super-Enhancers2) algorithm with the default setting[19]. The defined super-enhancer corresponds to the description of Pott et al. article[64]. For identifying the target genes of super-enhancers by using R/Bioconductor packages such as ChIPseeker, TxDb.Hsapiens.UCSC.hg19.knownGene, and TxDb.Mmusculus.UCSC.mm9.knownGene.

**Identification of core-regulatory circuits**. The interconnected circuitry was identified based on an established methodology, "Coltron", (https://pypi.org/project/coltron) scanning TF motifs inside nucleosome-free regions (NFRs) inside super-enhancer regions. Briefly, we first find out super-enhancer-assigned TFs in each of fibroblasts using ROSE2 algorithm. Next, we scanned TF motif with FIMO in super-enhancer regions extended 500 bp both upstream and downstream, identifying for auto-regulated TFs which are bind their own super-enhancer regions. Finally, binding motifs were searched for in all auto-enhancer regions of each of fibroblasts for potential additional TFs. We then constructed core-regulatory circuits (CRCs) based on fully interconnected auto-regulatory loops and these TFs were identified as master TFs candidates.

**Discovery of motif analysis using Homer**. Homer was specifically employed to locate binding motifs that was most concentrated in peaks of interest. HOMER's findMotifs.py (http://homer.salk.edu/homer/ngs/peakMotifs.html) function was executed with the default setting for motif search[65].

**Identification of significant phenotypes of fibroblasts using EnrichR**. To identify significant phenotypes of fibroblasts, we performed Gene ontology enrichment analysis using EnrichR database. super-enhancer associated genes, Gene ontology and pathway analysis was performed using Enrichr database (http://amp.pharm.mssm.edu/Enrichr/)[66].

**Measurement of network structural similarity**. To quantify structural similarity (S) among core-regulatory circuits composed of the master TF candidates, we used a network structural dissimilarity measure, D-measure (D), which compares networks based on three core network structural features (i.e., network's distance distributions, node's distances distributions, and alpha centrality) [1]. The D is defined as

$$D(G, G') = w_1 \sqrt{\frac{J(\mu_G, \mu_{G'})}{\log 2}} + w_2 \left| \sqrt{NND(G)} - \sqrt{NND(G')} \right| + w_3 \left( \sqrt{\frac{J(P_{\alpha G}, P_{\alpha G'})}{\log 2}} + \sqrt{\frac{J(P_{\alpha G^c}, P_{\alpha G'^c})}{\log 2}} \right) \quad (1)$$

where G indicates the network, Gc indicates the complement of G, J is the Jensen–Shannon divergence, $\mu_G$ is network's distance distribution of G, NND(G) is network node dispersion of G, and $P_{\alpha G}$ is α-centrality values of G[36]. The first term captures the global topological difference based on network's distance distributions. The second term captures the difference in the connectivity of each node based on node's distances distributions. The last term captures the difference in the way the connectivity arises based on α-centrality values. D does not take node identity into consider but is an efficient and precise measure for quantifying network structural dissimilarity. We used recommended default weight setting (w1 = w2 = 0.45, and w3 = 0.1) for D.

The S (similarity score between G (network A) and G' (network B)) is defined as

$$S(G, G') = 1 - D(G, G') \quad (2)$$

because D has a value greater than or equal to zero and less than one[36].

**Bulk ATAC-seq (assay for transposase-accessible chromatin using sequencing)**. Prepare 100,000 cultured human CAFs using LUNA-FL™ Automated Fluorescence Cell Counter (logos biosystems). Cells were lysed using cold lysis buffer. Determine the nuclei concentration using Countess II Automated Cell Counter (ThermoFisher) and examine the nuclei morphology using microscopy. Immediately after lysis, continue to transposition reaction. Resuspend nuclei(50,000 cells) in transposition reaction mix. Incubate the transposition

reaction for 30 min at 37 °C. Immediately following transposition, purify using a Qiagen MinElute PCR purification Kit. Amplify transposed DNA fragment using Nextera DNA Flex kit. To reduce GC and size bias in PCR, the appropriate number of cycles is determined using qPCR. Run a qPCR side reaction. Calculate the additional number of cycles needed, plot liner Rn versus cycle and determine the cycle number that corresponds to 1/4 of maximum fluorescent intensity. Run the remaining PCR reaction to the cycle number determined by qPCR. Purify amplified library. The purified libraries were quantified using qPCR according to the qPCR Quantification Protocol Guide(KAPA) and qualified using Bioanalyzer (Agilent technologies). And then the libraries were sequenced using HiSeq platform(Illumina).

After removing adaptors using TrimGalore, 100 bp paired-end ATAC-Seq reads were aligned to the hg19 using Bowtie2 with default parameters. We removed mitochondrial and Y chromosome DNA reads and included properly paired reads with high mapping quality using SAMtools (MAPQ score >30, qualified reads). Using Picard tools (MarkDuplicates function) Duplicate reads, we removed duplicated reads in aligned ATAC bam files. Next, Blacklisted regions were excluded from called peaks (https://www.encodeproject.org/annotations/ENCSR636HFF/).

**Single-cell RNA and Bulk RNA sequencing**. Detailed methods and materials are described in the Supplementary Methods.

**Statistics and reproducibility**. Wilcoxon rank sum test or $t$ test were used to evaluate differences between groups for continuous variables. Chi-sqaure test or Fisher exact test were used to evaluate the association between groups for categorical variables. Overall Survival (OS) was defined as time interval between dates of curative surgery and death from any cause. Disease Free Survival (DFS) was defined as time interval between dates of Curative surgery and tumor recurrence or death. OS and DFS rates were estimated using the Kaplan–Meier method and was compared survival distributions between two groups using the log-rank test. The Cox proportional hazards model were used for univariate and multivariate analysis. Clinicopathologic factors, which were statistically significant in univariated analysis, were included as covariables in multivariate analysis. Hazard ratios (HR) and 95% confidence intervals (CI) were assessed for each factor. All tests were Two-sided, and P-value of less than 0.05 was considered statistically significant. The statistical analysis was performed using SPSS statistical software (SPSS Inc, Chicago, IL, USA). Statistical analysis was performed with GraphPad Prism 7.0 (GraphPad Software). Differences were considered to be significant when *$P < 0.05$. The western blot experiment was repeated independently three times with similar results.

**Reporting summary**. Further information on research design is available in the Nature Research Reporting Summary linked to this article.

## Data availability
All original sequencing data are publicly available on the NCBI Gene Expression Omnibus (GEO) (https://www.ncbi.nlm.nih.gov/geo/). The accession number for ChIP, RNA, and single-cell RNA sequencing data reported in this paper are available on the GEO database under the accession codes such as GSE169725 (Super-series) GSE169703 (Three types of murine fibroblasts RNA-seq), GSE169723 (Three types of murine fibroblasts ChIP-seq), GSE169697 (RNA-seq data of Cocultures of cancer spheroid with fibroblasts), GSE169720 (Human cancer-associated fibroblasts RNA-seq), GSE169601 (Human cancer-associated fibroblasts ChIP-seq), GSE169704 (Single cell RNA-seq from murine wound tissue). The publicly available datasets used in this study are listed in Supplementary Table 1. The authors declare that all the remaining data supporting the results of this study are available within the Article, Supplementary Information or Source Data files.

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

## Acknowledgements

This work was supported by the National Research Foundation of Korea (NRF) grant funded by the Ministry of Science and ICT (MSIP) and Ministry of Education, Science, and Technology (MEST) (2016R1A5A2945889, SHK; 2018R1C1B6006428, KWL; 2019R1I1A1A01062460, SYY; 2019R1A2C2003900, SHK; 2020M3E5E2040308, SHK; 2020R1A2B5B03094920, KHC) and by the Electronics and Telecommunications Research Institute grant funded by the Korean government (22ZS1100, Core Technology Research for Self-Improving Integrated Artificial Intelligence System, KHC).

## Author contributions

K.W.L., S.Y.Y., J.R.G., and S.H.K. conducted all experiments and analyses; W.Y.L., H.C.K., S.H.Y., and Y.B.C. contributed to sample acquisition. I.S., M.A.C., O.J.K., S.A., J.K., and C.O.S. contributed to the experiments or analyses. S.H.K., K.W.L., S.Y.Y., J.R.G., and C.O.S. wrote the paper. K.H.C. and S.H.K. designed the experiments, supervised the study, and provide readership.

## Competing interests

The authors declare no competing interests.
