## [Peer Review File · Nature Communications]

Reviewers' Comments:

Reviewer #1:

Remarks to the Author:

In this work, the authors aim to annotate CAF subsets based on the expression and function of master transcription factors. This would provide an additional critical layer for resolving CAF heterogeneity, and potential novel targets for therapy. The authors utilize a broad range of in-vitro and in-vivo approaches, including multiple mouse models, bioinformatic analyses of previously published single cell data, in-vitro functional assays, and chip-seq experiments, which together identify PRRX1 as a master regulator that promotes a myofibroblastic CAF phenotype.

• The findings are very interesting, and supported by a substantial amount of results, and a vast array of methods. Three main issues are the novelty of the findings, the experimental controls, and the clarity of presentation. Feldmann et al recently published very similar findings on PRRX1 in pancreatic cancer (PMID 33007300). This work is not cited and significantly impacts the novelty of the current manuscript. Key and critical controls are missing from many experiments, and these are essential to support the authors conclusions, namely that fibroblast-specific expression of PRRX1 drives malignancy. Also, the figures are extremely dense with data, some of which is very shortly and partially described in the text, making this manuscript hard to follow and not clear. The novelty and these technical issues should be considered and must be addressed before the paper could be accepted for publication.

Major comments:

- 1) Most of the images in the manuscript are extremely small and of low resolution; it is difficult to understand and make conclusions regarding much of the imaging data. Furthermore, there are several instances of not referencing figures in the text and not labeling figures to make it clear which groups are being compared (please see below). This makes assessing the manuscript very challenging. Most importantly, quantification and statistics are lacking for the majority of the manuscript.
- 2) PRRX1 is well-known to drive EMT and metastases in a variety of cancers- what markers did the authors use to differentiate between CAFs and epithelial cells that underwent EMT? For example, in figure 1E the authors claim following IHC experiments that PRRX1 is unique to CAFs, yet no additional staining or markers are utilized for verification. In addition, the sentence in line 166 "PRRX1 is highly expressed only in CAFs" should be revisited given the large amount of evidence regarding the role of PRRX1 in cancer cells.
- 3) In the section titled: "Prrx1 is essential for CAFs to facilitate tumorigenicity and metastasis of co grafted tumors": Quantification of PRRX1a and PRXX1b levels in SNF32 fibroblasts is missing (Fig 1G)- the authors should include rt-qPCR or western blot analysis to demonstrate the expression levels in these different fibroblast populations. Also, insets from the control images are not provided for figure 1H-J. Furthermore, no statistics or grading by a pathologist are given for the IHC stains in H and J-how many slides were analyzed and what is the scoring criteria?
- 4) For figure 2A- it is not clear how specific deletion in fibroblasts was achieved- where were sgRNAs injected to generate complete loss of prrx1 in fibroblasts? Was it knocked out in all mouse tissues? No data is shown to demonstrate prrx1 deletion in fibroblasts. Moreover, FSP1-Cre mice are driven by the s100a4 promoter, which is active in additional cells other than fibroblasts, even in the TME (Cabezón et al, Int J cancer, 2007). Thus, no evidence presented demonstrates that the model is sufficient. For figure 2B there is little compelling evidence regarding specific induction of Prrx1 in fibroblasts alone. The authors only show tumor growth and IHC staining without quantification. The authors also do not explain what are "control" mice in figure 2. The controls should be Col1a2rtTA;TetO7-Prrx1Luc mice that are not treated with doxycycline- is this the case?
- 5) Figure 2E: there is no quantification of the data in the AOM/DSS models. Furthermore, no labeling of carcinoma cells was conducted in the imaging experiments. Once again, it is difficult to ascertain the expression of Prrx1 in the fibroblast compartment. As stated above, the resolution is low, and it is difficult to draw conclusions for the figure.
- 6) For figure 3B, since there is no evidence provided regarding aberration of Prrx1 in fibroblasts only (knockout or loss of function mutant), it is difficult to reconcile the RNA seq data. Figure 3C- how many biological replicates were conducted for the FACS-based assays? What markers were used to validate the identity of stomach CAFs? What is their source? For 3D, there is no quantification of gel contraction, and one cannot draw conclusions from the images provided. The labeling of the boundaries of the contracted gel makes it very difficult to assess this objectively - this labeling should be removed. Also, figure 3F lacks quantification.

- 7) For figure 4H, there is no quantification, and it is difficult to draw conclusions from the figure. As mentioned above, the authors did not describe the L.O.F mutant. 4I-no evidence of Prrx1 silencing is provided.
- 8) Figure 5 lacks normal fibroblast controls for the Chip-seq. The RNA expression levels of Prrx1 in the "Cluster A" subset is not provided. For 5G, no evidence of Prrx1 knockdown is provided.
- 9) For Figure 6, as above, no Prrx1 silencing specifically in fibroblasts is demonstrated, and no detail regarding the induction of CAS9 activity and how silencing was obtained is discussed by the authors.
- 10) For Figure 7, conclusions regarding the dual effect of cisplatin and prrx1 deletion in fibroblasts cannot be drawn since the authors did not establish fibroblast-specific knockout of prrx1 in the CAS9 model.

Supplementary figures:

- 11) The authors utilize co-injection of MMTV-derived CAFs with the 168FARN cell line to demonstrate the effect of PRRX1 on tumor progression (figure S4), however no data regarding the identity of the MMTV-CAF is provided. Which markers did the authors use to validate these were indeed CAFs? The authors must also show that Prrx1 is knocked out in MMTV CAFs- no data are provided.
- 12) Figure S4C lacks critical controls. The authors claim that "MMTV-CAF moved from the tail vein to the lung and induced massive metastasis of fat pad-residing 168 FARN cancer cells" however no markers for FARN cells or CAFs in the metastatic niche are provided. Given that no data are given regarding the identity of MMTV-CAF, these could likely be cancer cells that underwent EMT and established metastases in the lung. At least 1 component-168 FARN or CAFs must be labeled with a reporter or stained with a marker to validate the authors' claims. It is also curious that all cancer cells injected alone induce no observable tumor after more than 40 days inoculation. HT-29 cells and HCT116, for example, are highly aggressive and develop rapidly in immunodeficient animals.

The authors conclude the section with the sentence:

"These results and additional supporting findings (Supplementary Fig. S5) suggest that Prrx1 is essential for the tumorigenicity and metastasis-inducing capacity of CAFs and that Prrx1 alone can reprogram tumor-suppressive normal fibroblasts into CAFs".

Supplementary figure 5 contains a significant amount of data and should not be summarized with a single broad sentence. The authors must detail what was conducted in the figure, namely the murine experiments with co-injections of CAFs/MEFs, the fluorescent staining, and the conclusions that are drawn from these experiments. Also, the IHC images are also not quantified and since the resolution is low, it is difficult to ascertain what the authors' conclusions are. There is also no quantification or discussion regarding the IF staining in Fig. S5, and the images are of poor resolution. Finally, for the coinjection experiments with Prrx1 knockdown, the authors should show that the knockdown of Prrx1 does not affect the viability or proliferation of the fibroblasts themselves.

13) Fig. S6A-B: As stated for fig.2 above, there is no compelling evidence of induction of Prrx1 in fibroblasts uniquely. The IF staining is of low resolution and lacks quantification. Furthermore, S6A and B do not contain any staining for the carcinoma cells, making the specificity of staining for fibroblast difficult to resolve. S6D: no quantification of the data in the AOM/DSS models and no labeling of carcinoma cells was conducted in the imaging experiments.

14) Fig. S7C: no quantification of gel contraction is provided, and one cannot draw conclusions from the images provided. Fig. S7D-S7G contain large amounts of data, and the authors do not detail the results in the text - these should be discussed.

Minor comments:

- 1) Figure 1E lacks quantification of PRRX1 levels in the stroma compared to the cancer compartment.
- 2) There are some confusing inconsistencies with the table in figure S4A- under the "Figure" column, the title "supple 4A" is redundant since it is not describing a scheme- did the authors intend to call supple 4A supple 4B and supple 4B supple 4C? In addition, there is a typo in Supple 5A-it should read "Co-injection" not "Go-injection".
- 3) Many of the figures are too crowded and explanations in the text would suffice: one example is in figure 2D "summary box" is unnecessary, and the explanation of experiments conducted in 2E

can be moved to a separate supplementary table.

4) The migration, invasion, and sphere formation assays in S6C are not referenced in the text.

5) It is not stated what is the mutation in *Prrx1* in Fig S5A. How does this mutation prevent the function of *Prrx1*? Is it a canonical mutation?

Reviewer #2:

Remarks to the Author:

In Lee et al., the authors identify PRRX1, a transcription factor associated with fibroblast activation, mesenchymal identity and wound healing as a master transcription factor (mTF) of cancer associated fibroblasts (CAFs). PRRX1 is over expressed in CAFs, associated with tumor promoting fibroblast functions. In a series of elegant genetic experiments toggling PRRX1 specifically in fibroblasts, the authors show that PRRX1 is required for the pro-tumorigenic activity of CAFs. This is repeated in both immune compromised and immune competent mouse models and spans studies of recurrence, tumor growth/metastases, and tumor initiation. As expected PRRX1 also plays a critical role in wound healing suggesting that cancers co-opt and hyperactivate wound healing functions in fibroblasts to drive growth and metastases. Although this has long been suggested and implied, these data vigorously support this hypothesis.

These functional experiments are combined with rigorous mechanistic investigation into the role of PRRX1. The authors confirm that it is a master transcription factor by multiple approaches. Finally, the authors validate PRRX1-targeting in CAFs as a therapeutic target using fibroblast lineage-based depletion in concert with cisplatin treatment in a lewis lung carcinoma model.

Overall this is an extremely impressive study that combines extensive in vivo modeling of PRRX1 with a rigorous mechanistic examination of PRRX1 function and its molecular signatures in CAFs. I only have four minor comments for the authors.

1. It is possible I missed this, but the authors should confirm that loss of *Prrx1* and its associated super enhancers translates into loss of gene expression for the myofibroblastic signatures.

2. Are the cluster B CAFs that lack PRRX1 less tumorigenic?

3. The authors cite the Van Groningen paper (citation 31) which shows that PRRX1 is associated with tumor-intrinsic mesenchymal cell state change and resistance to therapy in neuroblastoma, but don't really comment on what PRRX1 might be doing when expressed in cancers. Can the authors comment more on this in the discussion?

4. This is likely beyond the scope of this study, but to truly validate the therapeutic index and potential of PRRX1 as a CAF target, the authors should consider a systematic model to down regulate it such as a dox inducible shRNA or degradation system. In addition to impaired wound healing, what are the likely consequences of systematic PRRX1 inhibition (even transiently) and would this support a therapeutic index?

Reviewer #4:

Remarks to the Author:

In this manuscript by Lee, Yeo, Gong and colleagues, the authors investigate putative master transcription factors that regulate cancer-associated fibroblast fate. They identify PRRX1 as a key regulator of the classically activated, myofibroblastic CAF subtype, consistent with its established role in non-cancer-associated mesenchymal lineages. The authors perform extensive gain- and loss-of-function experiments in diverse mouse models of cancer and wound healing, as well as robust functional genomics studies, computational analyses of publicly available datasets from several human cancer types, and homotypic and heterotypic culture experiments. This suite of experimental systems includes several sources of murine fibroblasts as well as human CAFs from several cancer types. This expansive analysis convincingly implicates PRRX1 as a critical regulator

of myofibroblastic CAF fate and function in several solid tumor types, and suggests a tumor-promoting role for stromal PRRX1. This gains significance in light of the recent flurry of studies suggesting transcriptional heterogeneity for CAFs across solid tumors, and begins to fill a crucial knowledge gap with respect to the regulators of these distinct CAF subtypes as well as molecular determinants of their tumor-supportive versus tumor-suppressive function. The present study is very impressive in breadth, and the data presented generally support the central claims of the paper. The manuscript is nicely written, experimental design is generally sound though exceptions are noted (detailed below), statistical analyses are appropriate throughout, and the study is likely to have a strong impact on the CAF field. A limited number of additional experiments, analyses, and/or edits will help to clarify the authors' findings.

Specific comments:

1. The authors have neglected to cite several key papers highly relevant to this study. These papers include recent reports of the key role of PRRX1 in fibroblasts in the context of wound healing and/or fate determination (Leavitt et al., *Cell Reports*, 2020, PMID: 33176144; Currie et al., *Biol Open*, 2019, PMID: 31278164) and another defining a tumor-promoting role for PRRX1 in pancreatic CAFs (Feldmann et al., *Gastroenterology*, 2021, PMID: 33007300). The lattermost does not meaningfully reduce the impact of the present study, but is important to acknowledge in the manuscript.
2. The relevance of intravenously injected CAFs for the results depicted in Fig S4C is unclear, and should be explained in the text.
3. The authors use sgRNA against *Prrx1* in mice as a basis for many loss-of-function experiments. Based on the results shown across diverse tumor models, it seems that the authors used this model to deplete *Prrx1* in fibroblasts systemically. However, important experimental details are lacking to explain how the sgRNAs were formulated and delivered to enable this systemic depletion; this is only briefly mentioned on line 65 of the supplement.
4. Validation should be provided for the fibroblast-specific knockout of *Prrx1* in vivo as depicted in Fig 2A and elsewhere in the manuscript.
5. Clarification is needed to explain why cancer cell outgrowth after surgical resection is only observed at the mediastinum around the heart in Fig 2D, S6C with systemic induction of *Prrx1* in fibroblasts. Based on the authors' central claims, it seems that *Prrx1* induction in fibroblasts systemically, as is the case in their model, should generate tumor-supportive niches across many anatomical sites.
6. Based on the results in Fig 3C and S7A, it seems that *Prrx1* knockdown or knockout is killing the CAFs. Is this the basis for the in vivo results involving *Prrx1* loss? Or is this due to CAF reprogramming? These conclusions have very different implications, especially because broad CAF depletion strategies seem undesirable as a therapeutic strategy based on the current literature. This also impacts interpretation of loss-of-function assays like those depicted in Fig 3D and S7C. Is the reduced contractility due to reprogramming and loss of transcriptional activity at genomic loci critical for myofibroblastic function? Or are the CAF/fibroblasts all dying?
7. The results in Fig 4H are interesting, but perhaps overstated in the text. Is there molecular evidence to suggest a functional interaction between TGF β and PRRX1? To this end, it would be helpful to treat control and PRRX1 knockdown/knockout CAFs with TGF β and do qPCR for fibrogenic genes, and/or analyze SMAD translocation under the same conditions.
8. The results in Figure 7 do not necessarily support a role for PRRX1 in cisplatin resistance. As genetic inhibition of *Prrx1* alone profoundly reduces tumor growth in this model, the authors cannot conclude that the CRs observed in the combination-treated group are due at all to cisplatin treatment (similar responses here +/- chemo). This role might be supported by histological analyses to show differences between *Prrx1*-depleted and *Prrx1*-depleted plus cisplatin, or by survival analysis. Otherwise, the conclusions should be tempered accordingly.

Reviewer #5:

Remarks to the Author:

The manuscript presented by Keun-Woo Lee, So-Youing Yeo, Jeong-Ryeol Gong and colleagues characterises the role of the transcription factor *Prrx1* in the specification of myofibroblasts and its impact on wound healing and tumorigenesis.

This manuscript presents a wealth of information including in silico analysis of public datasets, molecular analysis, in vitro and in vivo models for loss and gain of Prrx1 function, functional assays and experiments with patient-derived material.

Most of the conclusions claimed by the authors are supported by experimentation and experiments are generally well designed and executed. There are however key some concerns that need attention. Of particular importance, given the title of the manuscript, are the concerns associated to the epigenetic characterisation presented in Figures 4 and 5. As it stands, the title of the manuscript should be carefully reviewed.

Here is a summary of the concerns:

1. The manuscript includes extensive information and in some instances, this could be streamlined to focus on the main claims of the manuscript. I suggest evaluate redundancies and only keep the necessary information to support the claims and the main message of the manuscript.

2. Perhaps associated to the point above, there are some inconsistencies between the text and figures, to mention a few, in Figure 1 the text mentions 7 tumour types but in the figure, there are only 6 (head & neck seems to be missing). In Figure 2a, the text says 10/10 but the figure says 6/6. Figure 2E states 72.73% (2/13). S9 panels D and G have the same legend...etc. The manuscript should be carefully reviewed.

3. In Figure 1, what is the origin of the normal fibroblasts? What is their activation score as assessed in panel 1A? Are these comparable between cancer types?

4. What is the level of expression of Prrx1 after Cas9 deletion and upon Dox-mediated overexpression, and what is the baseline level of expression of Prrx1 in WT fibroblasts?. How do these levels compare to the different species of fibroblasts described in Fig 1C. There is abundant literature that reports transcriptional signatures to define at least 2 different types of CAFs in human and mice (PMID: 29988129 (Ref#23), PMID: 29198524, PMID: 30514914, PMID: 33852842, PMID: 32790115) what signatures overlap with high and low Prrx1 populations.

5. Results in Figure 2c are intriguing. The authors sought to explain the distant activation of Prrx1 after a wound-healing effect. It seems there is a distant hotspot of Prrx1 activation but it seems widespread activation to some extent in the whole animal. Can authors elaborate more on this point? From this part of the manuscript, it would be interesting to add an explicit narrative in the text highlighting similarities/differences of Prrx1 activation in the context of wound healing and cancer. This is important because the title specifies cancer context but rather the results seem to point to a general function of Prrx1 in myofibroblasts. In which case the title should be reworded accordingly as this is not CAF specific. Some of the wound-healing effects of Prrx1 enhancer has been already reported PMID: 31278164.

Was this hotspot always localised in the same area? What is the Luc expression of not wounded mice after Dox exposure? (it seems here "Control" is not really a Col1a2 reporter mouse). What is the luciferase signal after tumour implantation, does the hotspot dynamic (e.g. time span) change from sham and tumour implanted mice?

What is the basis for this specific site for hotspot activation? How do the authors rule out there is a higher concentration of Dox in the heart mediastinum? Did the tumour relapse at the incision site? These questions are important to evaluate if there is a specific metastatic tropism in Prrx1 overexpressed mice, or this is dependent on how much Prrx1 expression is achieved in each tissue. If metastasis correlates with high Prrx1 expression it might also appear in form of relapse in the primary hotspot around the wound produced by the tumour resection.

6. The desmoplasia phenotype is not convincing. How is fibroblast proliferation without tumour? What is the desmoplastic phenotype during wound healing instead AOM. Following my previous comment, is this a common phenotype driven by Prrx1 or is it cancer-specific?

7. What is the basis of the gating for figure 3C. A non-stained control should be shown. Representative images are shown in Figures 3C and D but no statistical analysis nor extent of replicates performed is shown.

8. The results shown in Figures 4 and 5 are intriguing and open a number of key questions.

8.1. The authors use a super-enhancer mapper to define super-enhancers based on H3K27ac and H3K4me3 profiles. The typical definition (and peak profile) of super-enhancers are large >12.5Kb clusters of H3K27ac, consistent with some of the results shown by the authors, for example in figure S9, where the mean length of SE range from ~19 to 46Kb in size, however, the results shown in figure 4 and particular 4C does not seem to be consistent with this definition, rather is consistent with a typical promoter definition. The authors should clearly define super-enhancers and apply appropriate visualisation in the figures to evaluate their presence and enable comparisons.

This SE definition is again repeated in figure 5, particularly concerning the SE definition in ADAMTSL.

8.2. It is also concerning the results shown for Prrx1 ChIP. What is the peak calling analysis for this ChIP, the data seems quite noisy, how many replicates have been performed of each ChIP? As a transcription factor-based ChIP it might need more replicates to ascertain the noise level. Additionally, a comparison of an IgG control is necessary for these ChIPs. What is the scale for the visualisation of 4C? Can authors specifically define what do they interpret as a SE in 4C?

8.3. Additionally, the visualisation of super-enhancers is not appropriate (for example in 4I or Fig S9 B-G and subsequent figures) should be performed with a wider scale (not just +/-1kb or 2kb of the TSS) as these regulatory regions are large in nature. The visualisation presented is just not informative for SE.

Figure S9 B -G the graphs should be shown in the same graph WT vs Prrx1 deletion to allow appropriate comparison. Attention should be given to the scale of these graphs.

8.4. Conceptually the results also open some key questions that need to be addressed. The data presented shows that Prrx1 KD results in a wide reprogramming of SE, however, the super-enhancer reorganisation is very different in the 3 different fibroblasts tested with only 6 SE in common. If Prrx1 is a master regulator of the myofibroblast fate, a more consistent reprogramming should be expected. Subsequently and given the low overlap, the functions presented although consistent with a myofibroblast functional fate are not consistent between the models, rather each model presents certain aspects of each of these functions. What are the specific functional modules underpinned by each of these models? This is also applicable to 4L. What is the overlap between the Prrx1 bound genes in the 3 fibroblast models? As a keeper of specific cell identity, these should extensively overlap.

Thus this contradicts one of the main claims of the manuscript which is assigning a master role to Prrx1.

8.5. Another question that might explain the disparate reprogramming is perhaps the initial state of these fibroblasts as they come from very different origins. What is the level of Prrx1 expression in all these lines before manipulation? How does this compare with the Prrx1 high/low definition? This is further shown in the heterogeneity found in the 9 CRC CAF cultures shown in figure 5. where authors find two very distinct SE organisations. How is the level of Prrx1 in these lines? Given the results in Figure 1, a more direct classification of the 9 CRC CAF explanted fibroblast could just rely on Prrx1 expression. Are these explanted cell lines pure clones in nature or present heterogeneity of expression of Prrx1?

9. In figure 5 authors find two main groups of fibroblasts in CRC patients. Clinically what is the survival data associated with these patients, is this consistent with the KM graphs shown in the in silico analysis?

Secondly, authors characterise cluster A extensively and basically keep cluster B as "not A"; apart from the obvious value of this analysis the results from cluster B can also validate the functional

definition of cluster A. What are the genes associated with cluster B? What are the differential SE overrepresented in cluster B over A, what are the functional modules associated with cluster B? What is the distribution of K27Me3Ac in cluster B and how does that correlate with the distribution of peaks in cluster A, from figure 5B it seems most of the identified peaks are flanking the TSS. The authors' interpretation of the SE distribution of the 9 CRC CAF lines is that they belong to two different groups based on Prrx1 biology, however, there is an alternative interpretation of this data irrespective of biology. Cluster B presents a lower number of loci, there are fewer SE identified and these are smaller (Fig S12), thus these data can be also interpreted as technical bias, due to potentially lower quality ChIPs for some of the cell lines analysed. In this case cell lines with high-quality ChIPs would be represented by cluster A and cell lines with low-quality ChIPs would form cluster B. Replicates will be necessary to address whether these 2 clusters are due to technical variation rather than biology. This hypothesis is further supported by the results in Figures 5E and 5F. As mentioned before, authors focus on Cluster A (high-quality ChIPs) but do the same analysis stands for cluster B, can authors identify direct correlations with the SE defined in cluster B and the gene expression profile? What is the level of transcriptional activity of Cluster A vs B, are these of the same magnitude? Authors should clearly control for quality biases and show this cluster A/B definition is not driven by quality.

Additionally, there is a third possibility that is not addressed in the experiments and it is the potential of senescence associated with cluster B, this is further supported by the results in panel 5I, where cluster B-like terminal shows overall less activation, and thus this could explain the overall reduced output of the K27me3ac ChIPs. Do Prrx1-high fibroblasts present canonical markers of CAF activation? For example, as defined in PMID: 20839032

To rule out technical biases and conclusively compare the K27me3Ac within these lines a spike in ChIP seq normalisation might be required.

10. The experiments shown in Figure 5G are not convincing, first apoptosis cannot be assessed by crystal violet, as cell numbers are mostly dependent on cell proliferation and only balanced by apoptosis levels; second, what is the level of expression of Prrx1 in the cells from cluster A and B? And what is the Prrx1 level after knockdown. Controls are missing for this experiment, therefore "transcriptional addiction to Prrx1 is not conclusively demonstrated".

11. The pseudotime analysis shows terminal #1 and Terminal #2, again assigning a myofibroblast identity to cluster A (terminal 1-like) but what is the identity of Cluster B terminal #2 remains unknown. Importantly what is the origin of these two terminals, what is the nature of the fibroblast source? As presented in Figures 5H and 5I there are 3 classes of Fibroblasts in the CAFs analysed, namely Source, Terminal 1 and Terminal 2. As such the violin plots in this analysis should contain the three cellular states/fates.

How does the module SE score for cluster A and B correlate with the murine fibroblasts of figure 4?→

12. The results presented in figure 7 are astonishing. What is the mechanism that induces cell death in cancer cells after Prrx1 deletion? How does this compare with the effects of SNF32? Did the cells migrate to metastatic sites? Understanding the effects in cancer cells is essential to propose Prrx1 depletion as a therapeutic avenue. For example, authors could assess the apoptosis of cancer cells in early time points. Are these experiments restricted to resistant cancer cells? Additionally, as a control, the levels of Prrx1 in the CAFs should be assessed. All 4 arms should be presented in the same graph (panel B) as they form part of the same experiment. The word "therapeutic" should be removed from "therapeutic deletion" or "therapeutic depletion", the result and the hypothesis can't be together in the same definition.

Reviewer #1

In this work, the authors aim to annotate CAF subsets based on the expression and function of master transcription factors. This would provide an additional critical layer for resolving CAF heterogeneity, and potential novel targets for therapy. The authors utilize a broad range of in-vitro and in-vivo approaches, including multiple mouse models, bioinformatic analyses of previously published single cell data, in-vitro functional assays, and chip-seq experiments, which together identify PRRX1 as a master regulator that promotes a myofibroblastic CAF phenotype.

- The findings are very interesting, and supported by a substantial amount of results, and a vast array of methods. Three main issues are the novelty of the findings, the experimental controls, and the clarity of presentation. Feldmann et al recently published very similar findings on PRRX1 in pancreatic cancer (PMID 33007300). This work is not cited and significantly impacts the novelty of the current manuscript. Key and critical controls are missing from many experiments, and these are essential to support the authors conclusions, namely that fibroblast-specific expression of PRRX1 drives malignancy. Also, the figures are extremely dense with data, some of which is very shortly and partially described in the text, making this manuscript hard to follow and not clear. The novelty and these technical issues should be considered and must be addressed before the paper could be accepted for publication.

Response:

We appreciate your interest in our research and constructive comments on our study. We believe that our study is novel since we elucidated the role of Prrx1 as a master transcription factor through extensive epigenetic studies including a large number of Chip-Seq/ATAC-SEQ experiments and *in vivo* studies using various genetically engineered mouse models and tumor graft models. We believe that the study by Feldmann *et al.* [PMID 33007300] mainly included tumor growth and progression by Prrx1 + CAF. However, our paper presented a considerable number of completely novel findings in the following aspects:

[1] This is the first study to reveal the epigenetic landscape of CAFs. In this study, the comprehensive profile of enhancers, super-enhancers, and promoters for 12 mice and human CAFs were revealed through H3K27ac/H3K4me3 ChIP-SEQ and ATAC-SEQ. For this study, 48 mouse fibroblasts ChIP-SEQ fastq files, 36 human CAF ChIP-SEQ fastq files, and 18 human CAF ATAC-SEQ files were newly generated, all of which reached 1.9 TB in size. Consequently, we reported 2,425 novel enhancers and 201 super-enhancers that have not been reported in the ENCODE database.

[2] This study discovered master transcription factors of CAFs for the first time. Since master TFs establish cellular identity, the identification of master TFs of CAF would contribute critically to the understanding of the most fundamental issues of CAFs, such as cellular heterogeneity.

[3] Our study is the first to reveal novel CAF subpopulations based on super-enhancer profiles of CAFs. Recently, numerous attempts to discover the CAF subpopulations based on single-cell RNA sequencing have been done. However, the classification of cell subpopulation based on super-enhancer profiles is suggested to be a more accurate and powerful method as reported in several studies including that by Gartlgruber *et al.* (Super-enhancers define regulatory subtypes and cell identity in neuroblastoma. *Nature Cancer* 2, 114–128 (2021)).

[4] This study is the first to develop a fibroblast-specific *Prrx1*-inducible mouse model. This mouse model was created by crossing two transgenic mice that were first established by us: A *col1a2*-rtTA and a *Rosa26*-TetO7-*Prrx1* mouse. Using this model, we first demonstrated that induced *Prrx1* expression in fibroblast can induce cancer metastasis even after the surgical removal of the primary tumor. Second, we induced a desmoplastic stromal reaction in this mice model. Last, we demonstrated the exaggerated wound healing, causing scar formation, in this fibroblast-specific *Prrx1* inducible mice model.

[5] This study is the first to demonstrate that the suppression of CAF alone can induce a long-lasting complete remission of established cancer. One of the biggest questions in CAF research is how much anti-cancer therapeutic effect can be achieved by targeting CAFs. Surprisingly, we observed that targeting CAF by fibroblast-specific *Prrx1* deletion can cure cancer completely in genetically engineered mice models.

[6] We used completely different genetically engineered mice from the study by Feldmann *et al.* for fibroblast-specific *Prrx1* deletion. We used *FSP1*-cre:LSL-CAS9 mice and induced fibroblast-specific *Prrx1* deletion by applying single guided RNA for *Prrx1* to these mice. In contrast, Feldmann *et al.* used *Sm22-Cre^{ERT}:Prrx1^{fl/fl}* mice for *Prrx1* deletion.

We have cited the reference "PMID 33007300" that we had missed earlier, and we have added a discussion for this reference with the above-mentioned points in the Discussion section. We have modified the figures for clarity in the revised manuscript. We performed additional experiments including control, chip-seq, ATAC-seq, and further functional experiments during the revision of the manuscript to obtain clearer findings.

Major comments:

1) Most of the images in the manuscript are extremely small and of low resolution; it is difficult to understand and make conclusions regarding much of the imaging data. Furthermore, there are several instances of not referencing figures in the text and not labeling figures to make it clear which groups are being compared (please see below). This makes assessing the manuscript very challenging. Most importantly, quantification and statistics are lacking for the majority of the manuscript.

Response:

Thank you for the detailed comment. As you suggested, we have modified the figures for high resolution. We have also carefully checked whether the figures are mentioned in the text. Additionally, we have presented the quantitative and statistical values for measurable image results in the revised manuscript (revised Fig. 2D, Fig. 3B-3C, Fig. 3G, Fig. 5B, Fig. 6F-G).

2) PRRX1 is well-known to drive EMT and metastases in a variety of cancers- what markers did the authors use to differentiate between CAFs and epithelial cells that underwent EMT? For example, in figure 1E the authors claim following IHC experiments that PRRX1 is unique to CAFs, yet no additional staining or markers are utilized for verification. In addition, the sentence in line 166 "PRRX1 is highly expressed only in CAFs" should be revisited given the large amount of evidence regarding the role of PRRX1 in cancer cells.

Response:

Initially, we, as professional pathologists, used distinctive histomorphological features of fibroblasts different from cancer cells presenting round shape with hyperchromatic coarse atypical nuclear morphology. However, as per your comment, we further performed pan-cytokeratin for epithelial cancer cells and vimentin for mesenchymal cells and Prrx1. As observed in a figure presented below (Supplementary Fig. 1F), Prrx1+/vimentin+ fibroblast with spindle shape in tumor stroma was not associated with cytokeratin + epithelial cancer cells. We have described these additional experiments in the revised manuscript.

As additional evidence, we assessed the Prrx1 expression in cells exhibiting high copy number alteration since it is more likely to occur in cancer cells. Consequently, using single-cell RNA sequencing (scRNA-seq) data, we further confirmed that fibroblasts with high Prrx1 expression revealed less copy number alteration whereas cancer cells with high copy number alteration revealed very rare Prrx1 expression. To identify aneuploid and diploid cells, we used the CopyKAT algorithm to calculate DNA copy number alteration from scRNA-seq data (Gao, Ruli, *et al.* "Delineating copy number and clonal substructure in human tumors from single-cell transcriptomes." *Nature biotechnology* 39.5, 599–608, 2021). From these results, we observed that aneuploidy is a highly specific and reliable quality for cancer cells since most of the aneuploid cells were cancer cells (below Fig. A). Among the CAF cells, only 11 out of 5508 cells were aneuploid, indicating that CAFs derived from cancer cells were extremely rare. These findings have been added in Supplementary Fig. 1E.

Furthermore, we have previously studied the mutation profile of human stomach cancer cells and patient-matched *ex vivo* culture of cancer-associated fibroblasts (Lee *et al.* Twist1 is a key regulator of cancer-associated fibroblasts. *Cancer Res* 75(1), 73–85, 2015). In that paper, we reported that multiple lines of mutations were observed in cancer cells, whereas no pathogenic mutations were observed in CAFs as presented in the figure given below. This data also strongly suggested that most of the CAFs are not derived from cancer cells.

To resolve the error in the sentence in line 166, we have revised the sentence "Prrx1 is highly expressed only in CAFs" to "Prrx1 is highly expressed mainly in CAFs~" to not contradict the various evidence regarding the role of Prrx1 in cancer cells.

3) In the section titled: "Prrx1 is essential for CAFs to facilitate tumorigenicity and metastasis of co grafted tumors": Quantification of PRRX1a and PRXX1b levels in SNF32 fibroblasts is missing (Fig 1G)- the authors should include rt-qPCR or western blot analysis to demonstrate the expression levels in these different fibroblast populations.

Response:

Thanks for your detailed instructions. We tried to present many results, however, we apologize for missing the most fundamental data. As you suggested, we have added the western blot data of Prrx1a and Prrx1b overexpression in SNF32 to the revised Supplementary Fig. 6.

Also, insets from the control images are not provided for figure 1H-J. Furthermore, no statistics or grading by a pathologist are given for the IHC stains in H and J-how many slides were analyzed and what is the scoring criteria?

Response:

According to your comments, we have provided inserts for Fig. 1H-J. We have also performed additional statistical analysis in Figs. H and J. All these results are presented in Supplementary Fig. 5A-B. The number of tissue slides was more than 30, and scoring was performed based on semiquantitative criteria including the percentage of positive cells and staining intensity by two experienced pathologists (COS & SHK).

4) For figure 2A- it is not clear how specific deletion in fibroblasts was achieved- where were sgRNAs injected to generate complete loss of prrx1 in fibroblasts? Was it knocked out in all mouse tissues? No data is shown to demonstrate prrx1 deletion in fibroblasts.

Response:

In this study, we induced the deletion of the *Prrx1* gene using the CRISPR/Cas9 genome-editing tool. To this end, single-guided RNA for *Prrx1* (sgPrrx1) was periodically introduced into FSP1^{cre}-CAS9^{EGFP} mice by intraperitoneal injection (information is described in detail in the Supplementary information). *Prrx1* deletion in the target organ was confirmed by western blotting of *Prrx1* in the target organ, i.e., wound healing tissue.

Additionally, we directly confirmed genome-editing by CRISPR-cas9 using T7E1 (T7 endonuclease 1) assay, which is used to determine the accuracy for indels produced by CRISPR-Cas9. If CRISPR-CAS9 mediated DNA cleavage occurs, T7E1 recognizes cleavage-associated DNA alteration and additionally cleaves DNA fragments. We collected wound healing tissues of FSP1^{cre}-CAS9^{EGFP} mice and performed T7E1 assay as follows: In sgPrrx1 injected mouse samples, T7E1-mediated DNA cleavages were observed, whereas in sgNS injected control mouse samples, no such cleavages were

observed.

The validation protocol of CAS9-mediated cleavage using T7E1 (T7 endonuclease I) assay

Finally, we confirmed CAS9-dependent *Prrx1* deletion in fibroblasts in wound healing tissue using multiplex immunohistochemistry as presented below. We injected single guided RNA for *Prrx1* to FSP1-cre:ROSA26-LSL-CAS9 mice and further created a skin wound and examined the expression of *Prrx1* and CAS9 in this wound healing tissue. An abundance of CAS9 expressions was observed in stromal fibroblasts, and remarkably, *Prrx1* expressions were rarely observed compared with those in control mice. Interestingly, the stromal cells expressing CAS9 were consistently negative for *Prrx1* and vice versa (Supplementary Fig. 21A).

In the above figure, a substantial proportion of stromal fibroblast in wound healing tissue was strongly positive for CAS9 whereas *Prrx1*-expressing fibroblast (arrow) was rare.

All these data presented above clearly indicated that Prrx1 was extensively and specifically deleted in fibroblasts by injection of sgPrrx1 to FSP1-cre:ROSA26-LSL-CAS9 mice.

Moreover, FSP1-Cre mice are driven by the s100a4 promoter, which is active in additional cells other than fibroblasts, even in the TME (Cabezón et al, Int J cancer, 2007). Thus, no evidence presented demonstrates that the model is sufficient. For figure 2B there is little compelling evidence regarding specific induction of Prrx1 in fibroblasts alone. The authors only show tumor growth and IHC staining without quantification. The authors also do not explain what are “control” mice in figure 2. The controls should be Col1a2rtTA;TetO7-Prrx1Luc mice that are not treated with doxycycline- is this the case?

Response:

① According to Cabezón *et al.*'s study, S100A4(FSP1) is expressed in various cell types such as macrophages, fibroblasts, activated lymphocytes, and even cancer cells. In this study, S100A4 expression was studied using antibodies for S100A4 by immunohistochemistry (IHC). However, the activity of the s100a4 promoter does not always match the level of S100A4 protein detected by IHC. The main reason for this is that the s100a4 promoter sequence inserted into the genome of FSP1-cre mice does not include the entire gene-regulatory elements for S100A4. Although the gene-regulatory elements are composed of enhancers, super-enhancer, and promoters, the s100a4 promoter sequence is only a part of the whole gene-regulatory element. Thus, s100a4 promoter activity is slightly different from the S100A4 protein level regulated by the whole gene-regulatory elements for S100A4.

Therefore, we studied the s100a4 promoter activity in wound healing tissue by examining the expression of CAS9 in FSP1-cre:ROSA26-LSL-CAS9 mice by immunohistochemistry. In this mouse model, CAS9 protein level was determined by s100a4 (FSP1)-promoter activity. Thus, we evaluated s100a4 promoter activity by examining the CAS9 protein level. As presented below, CAS9 was extensively expressed in fibroblasts and not in other cell types such as epithelial cells, endothelial cells, smooth muscle, and lymphocytes in wound healing tissues. Additionally, CAS9-expressing cells were negative for F4/80, macrophage marker. Therefore, in at least wound healing tissue, s100a4 promoter activity is likely to be highly fibroblast-specific.

FSP1 promoter activity was reported to be fibroblast-specific by a substantial number of studies including that by Chen *et al.* (Chen, Y., McAndrews, K.M. & Kalluri, R. Clinical and therapeutic relevance of cancer-associated fibroblasts. *Nat Rev Clin Oncol* **18**, 792–804, 2021).

② Induction of Prrx1 in Col1a2-rtTA:TetO7-Prrx1-Luc mice was confirmed by immunohistochemistry for Prrx1 in wound healing tissue. The data are demonstrated in Fig. 8F, as presented below.

③ “Control” mice in Fig. 2 are Col1a2rtTA:TetO7-Prrx1Luc mice that are not treated with doxycycline. We have mentioned this in the revised manuscript.

6) For figure 3B, since there is no evidence provided regarding aberration of Prrx1 in fibroblasts only (knockout or loss of function mutant), it is difficult to reconcile the RNA seq data.

Response:

Thanks for your detailed instructions. As you suggested, we have added evidence for Prrx1 deletion or mutant in fibroblasts (presented in Fig. below).

5) Figure 2E: there is no quantification of the data in the AOM/DSS models. Furthermore, no labeling of carcinoma cells was conducted in the imaging experiments. Once again, it is difficult to ascertain the expression of Prrx1 in the fibroblast compartment. As stated above, the resolution is low, and it is difficult to draw conclusions for the figure.

Response:

As you suggested, we have added the quantitative data of Fig. 2E in the revised Fig. 2D. Specifically, we quantified collagen deposition in connective highlighted by Masson's trichrome staining. We also quantified Prrx1 and SMA expression in cancer stroma as follows.

We confirmed that there were frequent aSMA/Prrx1-double-positive cells in the collagen-abundant regions in the cancer stroma whereas there were considerably few aSMA/Prrx1-double-positive cells in both collagen-poor stroma and sounding carcinoma cell regions. We performed staining for ki67 (below figure) to distinguish carcinoma as you pointed out.

Figure 3C- how many biological replicates were conducted for the FACS-based assays?

Response:

We conducted three biological replicates for FACS-based apoptosis assay as presented in Revised Fig. 3B and have added the sentence "Experiments were done in triplicates" in the legend of revised Fig. 3B.

What markers were used to validate the identity of stomach CAFs? What is their source?

Response:

The stomach CAFs were isolated from gastric cancer tissues of patients with gastric cancer resected in our hospital, and these cells were characterized in our previous study (Lee *et al.* Twist1 is a key regulator of cancer-associated fibroblasts. *Cancer Res* 75(1), 73–85, 2015). In that paper, we demonstrated the expression of representative fibroblast markers including FAP, SMA, and PDGFRa as presented below. Additionally, the pathologist confirmed that the histology of CAFs was completely different from that of epithelial cells under the microscope. We have cited this paper in the revised manuscript, from which readers can know the origin of the stomach CAFs.

For 3D, there is no quantification of gel contraction, and one cannot draw conclusions from the images provided. The labeling of the boundaries of the contracted gel makes it very difficult to assess this objectively – this labeling should be removed.

Response:

As you suggested, we have added quantification of gel contraction in the revised Fig. 3C and Supplementary Fig. 8. Additionally, we have deleted the label of the contraction gel boundary for objective interpretation.

Also, figure 3F lacks quantification.

Author response:

According to your suggestion, we have added quantification data in revised Fig. 3E (bottom) as follows.

7) For figure 4H, there is no quantification, and it is difficult to draw conclusions from the figure. As mentioned above, the authors did not describe the L.O.F mutant. 4I-no evidence of Prrx1 silencing is provided.

Response:

Thanks for your detailed instructions. Following your instructions, we have attached the quantitative data of Fig. 4H to revised Fig. 5B. Furthermore, we have added additional data according to comments from Reviewer #4 regarding the relationship between Prrx1 and smad3. These improved data are provided in the revised Fig. 5.

We confirmed Prrx1 silencing in L.O.F mutant by western blotting and RNA-seq as follows:

We have added this evidence for Prrx1 silencing in L.O.F mutant of MEF (mouse embryonic fibroblasts) in Supplementary Fig. 6A.

8-1) Figure 5 lacks normal fibroblast controls for the Chip-seq.

Response:

We additionally performed H3K27ac ChIP-seq of normal fibroblasts, which we established from non-tumor colon tissue of patients with CAF#4. We identified enhancers and super-enhancers of these normal fibroblasts. Unsupervised-clustering analysis of the super-enhancer profiles of these cells revealed three distinct clusters corresponding to 'normal fibroblast', 'Cluster-A CAF', and 'Cluster-B CAF' as presented below. These data strongly indicate that CAFs have a distinctly different epigenetic topography from normal fibroblasts. Additionally, these data were inserted in Supplementary Fig. 16A (below figure).

8-2) The RNA expression levels of Prrx1 in the "Cluster A" subset is not provided. For 5G, no evidence of Prrx1 knockdown is provided.

Response:

The RNA expression levels of Prrx1 in the "Cluster A" subset and evidence of Prrx1 knockdown are presented in Supplementary Fig. 18A.

9) For Figure 6, as above, no Prrx1 silencing specifically in fibroblasts is demonstrated, and no detail regarding the induction of CAS9 activity and how silencing was obtained is discussed by the authors.

Response:

A detailed answer to *Prrx1* silencing specifically in fibroblasts has been provided in our response to comment by reviewer comment #4 mentioned above. Details regarding the induction of CAS9 activity and how silencing was obtained is added in the Supplementary Information section as follows:

SgRNA formulation for delivery *in vivo* system

sgNS and sg*Prrx1* were produced by co-transfection of viral production plasmids in 10 cm culture plates containing HEK-293T cells per plate. DNA mixture for each plate was prepared as follows: 1 mL of plain Dulbecco's modified Eagle's medium (DMEM) with pLKO.1_Puro_NS or pLKO.1_Puro_sg*Prrx1* expression vector, 7.5 µg of packaging plasmid, PSPAX2, and 2.5 µg of enveloping plasmid. DNA mixture was added to 500 µL of plain DMEM containing 40 µL of lipofectamine (lipo; Invitrogen, catalog 11668027) and incubated at room temperature in the dark for 20 min. The DNA/lipo mixture for each sample was further added to each plate containing HEK-293T cells and incubated for 16 h at 37 °C. The medium was replaced with 10 mL of fresh complete DMEM, and the plates were incubated for 24 h. Viral supernatant was harvested and filtered through a 0.45 µm MF-Millipore™ Membrane Filter (Millipore; catalog HAWP04700) and concentrated either by ultracentrifugation at 10,000 *g* for 1–4 h or with Lenti-X Concentrator reagent for 30 min to a final concentration of 10× or 20× as per the instruction manual (CloneTech; catalog #PT4421-2). The supernatant was resuspended in phosphate-buffered saline (PBS). To test the feasibility of gene editing with lentiviral sgRNA delivery (Figure S20H), we designed seven unique sgRNAs targeting the gene *Prrx1*. After determining the most potent sgRNA in an *in vitro* assay in the mouse-derived wound healing fibroblast cells (Figure S20F), titering of viral particles was done with qPCR and western blot analysis.

Virus Delivery into the skin

The formulation was subsequently administered to Cas9-expressing mice. Two-month-old mice were anesthetized using isoflurane and set up in a biosafety cabinet. Mice were injected intradermally around the wound area with 100 µL concentrated sgRNA once every three days from the wounding start. For effective delivery, the intraperitoneal injection (IP) was performed simultaneously. In our study, we observed no significant change in the gross measure of health in mice.

In summary, fibroblast-specific cas9 expression mouse was generated by crossing Rosa26-LSL-Cas9 Knockin mice with FSP1-Cre-expressing mice. The Rosa26-LSL-Cas9 knock-in mice had a floxed-STOP cassette preventing the expression of the downstream bicistronic sequences (Cas9 and

EGFP). Without Cre recombinase, the expression of cas9 and EGFP was prevented by the STOP cassette. After exposure to Cre recombinase, the expression of cas9 and EGFP was allowed. Cas9 expression was tightly controlled in a Cre-dependent manner as previously reported Platt RJ *et al.* Fibroblast specific delivery of sgRNA was performed by injecting the mice around the wound area. SgRNA formulation and delivery to the wound area methods have been included as Supplementary Information.

Furthermore, we performed additional experiments to confirm the genome editing (Insertion-Deletion (indel) mutations) by Cre recombinase. The experimental procedure is illustrated as follows:
 Detection of nuclease-induced genomic mutations

Genomic DNA was extracted using DNeasy Blood and Tissue extraction kit (Qiagen) following the manufacturer's instructions. The target locus was amplified by PCR using High Fidelity DNA Polymerase (NEB). Oligos are listed in the Supplementary Information. To evaluate indels caused by cleavage of one gRNA, purified PCR products were Sanger sequenced and analyzed using the SYNTHIGO ICE software.

In conclusion, we confirmed a knockout score of 34% in sgPrx1-treated mice compared to control. This indicates that sgPrx1 is effectively delivered to the target tissue.

10) For Figure 7, conclusions regarding the dual effect of cisplatin and prrx1 deletion in fibroblasts cannot be drawn since the authors did not establish fibroblast-specific knockout of prrx1 in the CAS9 model.

Response:

The fibroblast-associated Prrx1 knockout in the CAS9 model has already been described in the response to reviewer comment No. 4 above.

Supplementary Figures:

11) The authors utilize co-injection of MMTV-derived CAFs with the 168FARN cell line to demonstrate the effect of PRRX1 on tumor progression (figure S4), however no data regarding the identity of the MMTV-CAFs is provided. Which markers did the authors use to validate these were indeed CAFs? The authors must also show that Prrx1 is knocked out in MMTV CAFs- no data are provided.

Response:

We confirmed the identity of the MMTV-CAFs in terms of histology, lack of expression for other cells, and high expression of activated fibroblast and CAF markers as follows:

We have already established and used MMTV-CAF in our previous paper (Lee *et al.* Drug repurposing screening identifies bortezomib and panobinostat as drugs targeting cancer-associated

fibroblasts by synergistic induction of apoptosis. Invest New Drugs 36(4), 545–560, 2018). We also excluded the possibility that MMTV-derived CAFs were contaminated by cancer cells through graft experiments. MMTV-CAFs alone could not form any tumor when grafted to mice whereas MMTV-cancer cells formed large tumors when grafted.

12) Figure S4C lacks critical controls. The authors claim that “MMTV-CAFs moved from the tail vein to the lung and induced massive metastasis of fat pad-residing 168 FARN cancer cells” however no markers for FARN cells or CAFs in the metastatic niche are provided. Given that no data are given regarding the identity of MMTV-CAFs, these could likely be cancer cells that underwent EMT and established metastases in the lung. At least 1 component-168 FARN or CAFs must be labeled with a reporter or stained with a marker to validate the authors’ claims.

Response:

As you recommended, we performed additional immunohistochemical staining to ascertain whether the atypical cells in the lung (revised Fig. S3B-C) were 168-FARN cells instead of MMTV-CAFs. Specifically, we performed staining for CK (cytokeratin, for cancer cell detection), a-SMA (for CAF detection). Tumor cells were mostly negative for a-SMA and positive for Cytokeratin. Combining histological features and marker profiles, most of the tumor cells in the lung were concluded to be 168-FARN cells. This additional result was added in Supplementary Fig. 3C.

It is also curious that all cancer cells injected alone induce no observable tumor after more than 40 days inoculation. HT-29 cells and HCT116, for example, are highly aggressive and develop rapidly in immunodeficient animals.

Response:

The delayed growth of cancer cells was owing to the implantation of small numbers of cancer cells. Generally, 2×10^6 – 2×10^7 cancer cells are grafted, however, only 5×10^5 cancer cells were grafted in this study as follows:

Comparison of the number of cancer cells injected into immunodeficient mice model.

Cell line	Cell number(in our study)	Cell number(other studies)
HT29	5×10^5 cells	$5 \times 10^6 \sim 2 \times 10^7$ cells
HCT116	5×10^5 cells	$2 \times 10^6 \sim 2 \times 10^7$ cells
MKN28	5×10^5 cells	$5 \times 10^6 \sim 8 \times 10^6$ cells

The authors conclude the section with the sentence:

“These results and additional supporting findings (Supplementary Fig. S5) suggest that Prrx1 is essential for the tumorigenicity and metastasis-inducing capacity of CAFs and that Prrx1 alone can reprogram tumor-suppressive normal fibroblasts into CAFs”.

Supplementary figure 5 contains a significant amount of data and should not be summarized with a single broad sentence. The authors must detail what was conducted in the figure, namely the murine experiments with co-injections of CAFs/MEFs, the fluorescent staining, and the conclusions that are drawn from these experiments. Also, the IHC images are also not quantified and since the resolution is low, it is difficult to ascertain what the authors conclusions are. There is also no quantification or discussion regarding the IF staining in Fig. S5, and the images are of poor resolution. Finally, for the coinjection experiments with Prrx1 knockdown, the authors should show that the knockdown of Prrx1 does not affect the viability or proliferation of the fibroblasts themselves.

Response:

We followed your recommendations and have improved the Supplementary Figs. 3, 4, 5, 7 by adding quantification data and improved images. The viable fibroblasts were identified significantly more than in the control group (tumor only) in Prrx1 knockdown as presented in Supplementary Fig. 4, although SMA and Prrx1 signals were decreased in those fibroblasts.

Additionally, we have added more description about Supplementary Fig. 4 in the main text as follows:

“Similar experiments were repeated by cografing human colon cancer cells (HT29 and HCT116) and stomach cancer cells (MKN28) with MEF (mouse embryofibroblast). Then, identical results, which are presented in Supplementary Fig. 4 were obtained.”

13) Fig. S6A-B: As stated for fig.2 above, there is no compelling evidence of induction of Prrx1 in fibroblasts uniquely. The IF staining is of low resolution and lacks quantification. Furthermore, S6A and B do not contain any staining for the carcinoma cells, making the specificity of staining for fibroblast difficult to resolve. S6D: no quantification of the data in the AOM/DSS models and no labeling of carcinoma cells was conducted in the imaging experiments.

Response:

We have already presented evidence of Prrx1 induction in fibroblasts uniquely in responses to comments #4 and #5 above. We have improved the resolution of the images by replacing the over-compressed IHC digital image with a less compressed file. Additionally, we did quantification of IHC staining using Image J Fiji software and have added the results in the revised Supplementary Fig. 7A as follows:

The carcinoma cells were reconfirmed by professional pathologists (C.O.S and S.H.K). They have very unique histologic features, such as irregularity in the nuclear membrane, enlarged nucleolus, abnormal distribution of chromatin, etc. Thus, a trained professional pathologist can distinguish cancer cells from normal cells with a very high degree of accuracy.

We also quantified Supplementary Fig. 6D according to your recommendation and have added the results in the revised Supplementary Fig. 7C as follows:

14) Fig. S7C: no quantification of gel contraction is provided, and one cannot draw conclusions from the images provided. Fig. S7D-S7G contain large amounts of data, and the authors do not detail the results in the text - these should be discussed.

Response:

Following your instructions 1, we have added the quantification data of Fig. S7C in the revised Fig. S8C and a description about the Fig. S7D-S7G (revised as Fig. S8 and S9) in the revised manuscript and supplementary figure legend.

Minor comments:

1) Figure 1E lacks quantification of PRRX1 levels in the stroma compared to the cancer compartment.

Response:

In this study, the frequency of cancer cells presenting Prrx1 protein expression by immunohistochemistry was very low, therefore, a comparison of the expression difference between fibroblasts and cancer cells was not conducted. When Prrx1 expression level was estimated in cancer cells, only 10 cases (5.4%) of 185 CRCs revealed Prrx1 expression in >5% of cancer cells. The signal intensity of Prrx1 was also weaker in cancer cells compared with that in fibroblast. Therefore, we concluded that Prrx1 was dominantly expressed in fibroblasts albeit not in cancer cells. This finding is well correlated with the result from independent single-cell sequencing data (Fig. 1D and Supplementary Fig. 1D).

2) There are some confusing inconsistencies with the table in figure S4A- under the "Figure" column, the title "supple 4A" is redundant since it is not describing a scheme- did the authors intend to call supple 4A supple 4B and supple 4B supple 4C? In addition, there is a typo in Supple 5A-it should read "Co-injection" not "Go-injection".

Response:

As you suggested, we have removed the table in Fig. S4A and corrected in Supplementary Fig. 5A.

3) Many of the figures are too crowded and explanations in the text would suffice: one example is in figure 2D "summary box" is unnecessary, and the explanation of experiments conducted in 2E can be moved to a separate supplementary table.

Response:

Thanks for your detailed instructions. Following the same, we have removed the "summary box" from Fig. 2D and rearranged the experimental description from Fig. 2E to Supplementary Fig. 7.

4) The migration, invasion, and sphere formation assays in S6C are not referenced in the text.

Response:

Owing to the limitation in the total number of words in the manuscript and not informative of this figure, we have deleted Fig. S6C in the revised manuscript.

5) It is not stated what is the mutation in Prrx1 in Fig S5A. How does this mutation prevent the function of Prrx1? Is it a canonical mutation?

Response:

Thanks for your detailed instructions. Genetic alteration information of Prrx1-mutated MEF cells is mentioned in the study by Lu *et al.* For a generation of the Prrx1-LacZ, a *LacZ* gene was introduced in a frame within the homeodomain in exon 2.

We backcrossed these mice into C57BL6 and isolated Prrx1 mutant MEF cells. Unlike WT, in this mouse, since the *LacZ* gene was inserted in a frame within the homeodomain of Prrx1 exon 2, expression of Prrx1 was impossible, and we confirmed this by western blot, RNA-seq, and ICC. Therefore, this is a canonical mutation secondary to the malfunction of Prrx1 expression.

We have added a description of the information of Prrx1 mutant MEF cells to the Supplementary methods section as follows:

"Co-injection model: Prrx1 mutant MEF cells were isolated from Prx-1lacZ mutant mice (Prrx1^{tm1Jfm/Mmmh}, MMRRC_000347-MU, backcrossed to C57BL6), in which the *LacZ* gene was inserted in a frame within the homeodomain of Prrx1 exon 2."

Reviewer #2

In Lee et al., the authors identify PRRX1, a transcription factor associated with fibroblast activation, mesenchymal identity and wound healing as a master transcription factor (mTF) of cancer associated fibroblasts (CAFs). PRRX1 is over expressed in CAFs, associated with tumor promoting fibroblast functions. In a series of elegant genetic experiments toggling PRRX1 specifically in fibroblasts, the authors show that PRRX1 is required for the pro-tumorigenic activity of CAFs. This is repeated in both immune compromised and immune competent mouse models and spans studies of recurrence, tumor growth/metastases, and tumor initiation. As expected PRRX1 also plays a critical role in wound healing suggesting that cancers co-opt and hyperactivate wound healing functions in fibroblasts to drive growth and metastases. Although this has long been suggested and implied, these data vigorously support this hypothesis.

These functional experiments are combined with rigorous mechanistic investigation into the role of PRRX1. The authors confirm that it is a master transcription factor by multiple approaches. Finally, the authors validate PRRX1-targeting in CAFs as a therapeutic target using fibroblast lineage-based depletion in concert with cisplatin treatment in a lewis lung carcinoma model.

Overall this is an extremely impressive study that combines extensive in vivo modeling of PRRX1 with a rigorous mechanistic examination of PRRX1 function and its molecular signatures in CAFs. I only have four minor comments for the authors.

1. It is possible I missed this, but the authors should confirm that loss of Prrx1 and its associated super enhancers translates into loss of gene expression for the myofibroblastic signatures.

Response:

Thanks for your kind comment. We already mentioned the raised issue in the revised Fig. 3A and revised Supplementary Fig. 13-14. In those experiments, we knock-downed Prrx1 expression in various fibroblasts and then examined the super-enhancer profile and RNA-expression pattern of these cells by H3K27ac ChIP-SEQ and RNA-SEQ. We observed that the super-enhancer landscape

significantly changed in fibroblast in response to Prrx1 Knock-down. The super-enhancers lost in response to Prrx1 knock-down were significantly associated with myofibroblastic signatures. Additionally, an identical pattern was observed in RNA-expression profiles. We have improved the readability of these data and described detail in Supplementary Figs. 13–14. We further experimented Chip-seq with Prrx1 antibody and confirmed that the target genes directly bound by Prrx1 were mostly associated with myofibroblastic functions (Revised Fig. 7F-H).

2. Are the cluster B CAFs that lack PRRX1 less tumorigenic?

Response:

As you recommended, we performed additional experiments to identify differences in tumor-promoting ability between CAFs of cluster A and cluster B. First, we evaluated the ability of CAFs to promote invasion, migration, and sphere formation of cancer cells. Results revealed that the cluster-B CAFs were inferior to cluster-A CAFs as follows:

Of these results, the CAF's abilities to promote tumor invasion and migration are related to promoting tumor metastasis whereas the ability to promote tumor spheroid formation is more closely related to promoting tumorigenicity. Therefore, it can be concluded that cluster-B CAFs were less tumorigenic compared with cluster-A CAFs. These results have been added in the revised Fig. 6G.

3. The authors cite the Van Groningen paper (citation 31) which shows that PRRX1 is associated with tumor-intrinsic mesenchymal cell state change and resistance to therapy in neuroblastoma, but don't really comment on what PRRX1 might be doing when expressed in cancers. Can the authors comment more on this in the discussion?

Response:

As you pointed out, the role of Prrx1 when expressed in cancers is important to be determined. Prrx1 is characteristically and most highly expressed in neural crest and embryo primitive mesenchyme. Since at least some subsets of neuroblastoma and CAFs are derived from the neural crest, Prrx1 may likely run common developmental programs in both cells. Interestingly, Prrx1 may be the main switch to turn on developmental programs in other cancer cells. Developmental programs, such as EMT, are highly likely to play a critical role in advanced cancers. However, transcription factors, including Prrx1, work invariably as a team with other transcription factors and cofactors. Therefore, the exact function of Prrx1 depends on its partner, and there is a limit in predicting the role of Prrx1 in cancer without the knowledge of its partner. We have inserted the following comments in the Discussion section:

"According to Van Groningen *et al.* study²⁴, Prrx1 is associated with tumor-intrinsic mesenchymal cell state change and resistance to therapy in neuroblastoma. It is plausible that Prrx1 may induce common mesenchymal programs in cancer cells, however, this requires further studies."

4. This is likely beyond the scope of this study, but to truly validate the therapeutic index and potential of PRRX1 as a CAF target, the authors should consider a systematic model to down regulate it such as a dox inducible shRNA or degradation system. In addition to impaired wound healing, what are the likely consequences of systematic PRRX1 inhibition (even transiently) and would this support a therapeutic index?

Response:

Thanks for your kind comment. We acknowledge that it is very important to build the systemic models for the validation of the therapeutic index and potential of Prrx1 as a CAF target. We have considered transgenic mice containing dox inducible shRNA against the Prrx1 system as you recommended. As an alternative, Rosa26-CreER^{T2} mice crossed by Prrx1^{fl/fl} mice can be considered.

Prrx1 has been known to be expressed transiently during embryonic tissue and not to be expressed in adult tissue except for a minor population of mesenchymal stem cells, wound healing tissue, and cancer stroma. We examined the Prrx1 expression pattern in mouse embryo tissue development as follows:

Expectedly, Prrx1 was transiently expressed at specific time points (days 9–12) during embryogenesis and then dramatically decreased as morphogenesis was completed.

Additionally, the analyses of single-cell RNA-seq data of multiple cancer types and normal tissues again confirmed that Prrx1 expression is limited to cancer stroma in adult tissues as follows:

Fig. 1D

Suppl Fig. 1D

These findings suggest that Prrx1 inhibition in an adult is unlikely to induce side effects. In line with this suggestion, in our fibroblast-specific Prrx1 knockdown mice, no serious side effects were

observed. In conclusion, we suspect that inhibition of Prrx1 may yield a good therapeutic index. However, we acknowledge that verification should be done using such a systemic mice model. We have addressed these issues in the Discussion section of the revised manuscript as follows:

"In the Discussion section-

"Notably, targeting Prrx1 is least likely to produce side effects since its expression is restricted to embryonic mesenchymal cells^{52,53}, which are rarely expressed in normal adult tissues, except for a small number of mesenchymal stem cells. We treated mice for more than 5 months with sgRNA against Prrx1 and observed no serious side effects. However, full evaluation using a systemic Prrx1 inhibition model would be needed to evaluate the true therapeutic index for Prrx1 targeting treatment."

Reviewer #4

In this manuscript by Lee, Yeo, Gong and colleagues, the authors investigate putative master transcription factors that regulate cancer-associated fibroblast fate. They identify PRRX1 as a key regulator of the classically activated, myofibroblastic CAF subtype, consistent with its established role in non-cancer-associated mesenchymal lineages. The authors perform extensive gain- and loss-of-function experiments in diverse mouse models of cancer and wound healing, as well as robust functional genomics studies, computational analyses of publicly available datasets from several human cancer types, and homotypic and heterotypic culture experiments. This suite of experimental systems includes several sources of murine fibroblasts as well as human CAFs from several cancer types. This expansive analysis convincingly implicates PRRX1 as a critical regulator of myofibroblastic CAF fate and function in several solid tumor types, and suggests a tumor-promoting role for stromal PRRX1. This gains significance in light of the recent flurry of studies suggesting transcriptional heterogeneity for CAFs across solid tumors, and begins to fill a crucial knowledge gap with respect to the regulators of these distinct CAF subtypes as well as molecular determinants of their tumor-supportive versus tumor-suppressive function. The present study is very impressive in breadth, and the data presented generally support the central claims of the paper. The manuscript is nicely written, experimental design is generally sound though exceptions are noted (detailed below), statistical analyses are appropriate throughout, and the study is likely to have a strong impact on the CAF field. A limited number of additional experiments, analyses, and/or edits will help to clarify the authors' findings.

Specific comments:

1. The authors have neglected to cite several key papers highly relevant to this study. These papers include recent reports of the key role of PRRX1 in fibroblasts in the context of wound healing and/or fate determination (Leavitt et al., *Cell Reports*, 2020, PMID: 33176144; Currie et al., *Biol Open*, 2019, PMID: 31278164) and another defining a tumor-promoting role for PRRX1 in pancreatic CAFs (Feldmann et al., *Gastroenterology*, 2021, PMID: 33007300). The lattermost does not meaningfully reduce the impact of the present study, but is important to acknowledge in the manuscript.

Response:

As you recommended, we have included these references and mentioned them in the Discussion section in the revised manuscript.

2. The relevance of intravenously injected CAFs for the results depicted in Fig S4C is unclear, and should be explained in the text.

Response:

As you suggested, we have explained the relevance of intravenously injected CAFs in the results depicted in Fig. S4C in the text as follows:

"~Then, the systemic effect of MMTV-CAFs on tumor cells was explored by injecting MMTV-CAFs into the tail vein of NOD/SCID mice. In contrast, 168FARN cancer cells were implanted into a mammary fat pad (Supplementary Fig. 3B). Massive metastasis of 168FARN cancer cells to the lungs was observed, causing the death of four of five mice. These findings suggested that MMTV-CAFs in lungs moved from the tail vein were likely to induce metastasis of 168FARN cancer cells present in the fat pad through systemic effect. This interesting phenomenon was not observed in mice injected with Prrx1-deleted MMTV-CAFs. Immunostaining for cytokeratin and α SMA confirmed the metastasis of 168FARN cells to the lung (Supplementary Fig. 3C)."

3. The authors use sgRNA against Prrx1 in mice as a basis for many loss-of-function experiments. Based on the results shown across diverse tumor models, it seems that the authors used this model to deplete Prrx1 in fibroblasts systemically. However, important experimental details are lacking to explain how the sgRNAs were formulated and delivered to enable this systemic depletion; this is only briefly mentioned on line 65 of the supplement.

Response:

As you recommended, we have included sgRNA delivery methods to enable this systemic depletion *in vivo* in the revised Supplementary method section as follows:

"SgRNA formulation for delivery *in vivo* system

sgNS and sgPrrx1 were produced by co-transfection of viral production plasmids in 10-cm culture plates containing HEK-293T cells per plate. DNA mixture for each plate was prepared as follows:

1 mL of plain Dulbecco's modified Eagle's medium (DMEM) with pLKO.1_Puro_NS or pLKO.1_Puro_sgPrrx1 expression vector, 7.5 μ g of packaging plasmid PSPAX2, and 2.5 μ g of enveloping plasmid. DNA mixture was added to 500 μ L of plain DMEM containing 40 μ L of lipofectamine (lipo; Invitrogen, catalog 11668027) and incubated at room temperature in the dark for 20 min. The DNA/lipo mixture for each sample was further added to each plate containing HEK-293T cells and incubated for 16 h at 37 °C. The medium was replaced with 10 mL of fresh complete DMEM, and the plates were incubated for 24 h. Viral supernatant was harvested and filtered through a 0.45 μ m MF-Millipore™ Membrane Filter (Millipore; catalog HAWP04700) and concentrated either by ultracentrifugation at 10,000 *g* for 1–4 h or with Lenti-X Concentrator reagent for 30 min to a final concentration of 10 \times or 20 \times as per the instruction manual (CloneTech; catalog #PT4421-2). The supernatant was resuspended in phosphate-buffered saline (PBS). To test the feasibility of gene editing with lentiviral sgRNA delivery (Fig. S20H), we designed seven unique sgRNAs targeting the gene *Prrx1*. After determining the most potent sgRNA in an *in vitro* assay in the mouse-derived wound healing fibroblast cells (Fig. S20F), titering of viral particles was done with qPCR and western blot analysis."

4. Validation should be provided for the fibroblast-specific knockout of Prrx1 in vivo as depicted in Fig 2A and elsewhere in the manuscript.

Response:

Reviewer #1 provided a similar comment (#1–4) and we have answered it above. Validation of the fibroblast-specific knockout of Prrx1 was performed in three different ways as follows: First, after delivery of sgRNA into the target organ of FSP1-cre:LSL-CAS9 mice, tissues were harvested and the expression of Prrx1 was further confirmed using western blotting. The results revealed that Prrx1 expression was significantly reduced in sgPrrx1 treated tissue.

Second, gDNA was extracted from the target organ of these mice, and a T7E1 (T7 endonuclease

1) assay was performed to detect genome-editing by CRISPR-cas9. T7E1 was supposed to recognize cleavage-associated DNA alteration and additionally cleave DNA fragments. We collected wound healing tissues of FSP1^{cre}-CAS9^{EGFP} mice and performed T7E1 assay as follows: In sgPrrx1 injected mouse samples, T7E1 mediated DNA cleavages were observed whereas in sgNS injected control mouse samples, no such cleavages were observed.

The validation protocol of CAS9-mediated cleavage using T7E1(T7 endonuclease I) assay

Third, immunostaining was performed to validate fibroblast-specific Prrx1 knockdown in protein level. The immunostaining revealed that CAS9-expressing stromal fibroblasts were negative for Prrx1.

Forth, we studied the s100a4 promoter's activity in wound healing tissue by examining the expression of CAS9 in FSP1-cre:ROSA26-LSL-CAS9 mice by immunohistochemistry. In this mouse model, CAS9 protein level is determined by s100a4 (FSP1)-promoter activity. Thus, we evaluated s100a4 promoter activity by examining CAS9 protein levels. As presented below, CAS9 was extensively expressed in fibroblasts and not in other cell types such as epithelial cells, endothelial

cells, smooth muscle, and lymphocytes in wound healing tissues. Additionally, CAS9-expressing cells were negative for F4/80, macrophage marker. Therefore, in at least wound healing tissue, s100a4 promoter activity is likely to be highly fibroblast-specific.

The results are presented in Supplementary Fig. 21 of the revised manuscript.

5. Clarification is needed to explain why cancer cell outgrowth after surgical resection is only observed at the mediastinum around the heart in Fig 2D, S6C with systemic induction of Prrx1 in fibroblasts. Based on the authors' central claims, it seems that Prrx1 induction in fibroblasts systemically, as is the case in their model, should generate tumor-supportive niches across many anatomical sites.

Response:

We observed that fibroblast and mesothelial cells on the surface of pleura in pericardial space and mid-mediastinum were activated by the systemic effect of the wound in Col1a2-rtTA transgenic mice. We suspect that this phenomenon is related to the intrinsic properties of Col1a2-rtTA transgenic mice. The transgenic mice may express their target genes in aberrant tissue depending on the insertion site of genes in chromosomes. Although the detailed mechanism of this phenomenon is not identified, this is certainly a highly reliable and reproducible phenomenon in this transgenic mouse. Therefore, from a practical point of view, this mouse can be used as a good experimental model for the study of tumor recurrence and metastasis.

Detailed microscopic examination of metastatic tumor sites has yielded many interesting results. The metastatic tumors were multiple rather than single and grew on the pleural surface of pericardium and mid-mediastinum. Interestingly, in the identical sites, a considerable number of

inflammatory lesions were also observed. These inflammatory lesions can be defined as inflamed granulation tissue in pathological terms. Inflamed granulation tissue is composed of fibroblast proliferation and blood vessels, which are commonly observed in wound healing tissues and chronic inflammation. Most interestingly, most fibroblasts in these inflamed granulation tissues expressed Prrx1 in their nucleus as presented below.

Summary of histologic findings.

1. Multiple metastatic cancers on the surface of pericardium and pleura in mid-mediastinum.
2. Multiple abnormal inflammatory lesions on the surface of pleura and pericardium
3. These inflammatory lesions were inflamed granulation tissue.
4. In these inflammatory lesions, Prrx1-positive stromal cells and fibroblasts were frequently found.

This phenomenon did not occur in the skin area where the tumor was excised. We speculate that the inflammatory response was suppressed in this skin area since the skin was sutured with thread after surgical resection of the tumor, finally causing repression of tumor recurrence. Suturing the wound with thread reduces the tension on the skin and prevents foreign substances from entering the skin, thereby, suppressing the inflammatory reaction. Microscopic examination of the tumor excision site revealed that there were no inflammatory reactions and Prrx1-positive fibroblasts in these sutured wound sites.

In conclusion, we strongly suggest that Prrx1 induction in fibroblasts plays a critical role in tumor recurrence and metastasis.

6. Based on the results in Fig 3C and S7A, it seems that Prrx1 knockdown or knockout is killing

the CAFs. Is this the basis for the *in vivo* results involving Prrx1 loss? Or is this due to CAF reprogramming? These conclusions have very different implications, especially because broad CAF depletion strategies seem undesirable as a therapeutic strategy based on the current literature. This also impacts interpretation of loss-of-function assays like those depicted in Fig 3D and S7C. Is the reduced contractility due to reprogramming and loss of transcriptional activity at genomic loci critical for myofibroblastic function? Or are the CAF/fibroblasts all dying?

Response:

As you pointed out, Prrx1 knockdown has effects of both enhancing apoptosis and reducing functions inherent to myofibroblasts. However, at least for the co-transplant assay, since only CAFs that had persistently survived after Prrx1 depletion were selected and injected into mice, it is very unlikely that the *in vivo* results involving Prrx1 loss would be induced by the death of CAFs by Prrx1 knockdown. In this study, we observed that Prrx1 depletion initially had a significant impact on the survival of CAFs, however, a considerable portion of CAFs survived. Interestingly, these surviving CAFs lost their identity as myofibroblast-like cells. The loss of myofibroblast-inherent functions was more likely to be owing to CAF reprogramming rather than a transient response to stimulus since Prrx1 is a master transcription factor and loss of Prrx1 induced the extensive change in the epigenetic landscape including super-enhancers as presented in Supplementary Figs. 13-14. In conclusion, we presume that Prrx1 knockdown may have therapeutic effects through both CAF reprogramming and CAF depletion.

7. The results in Fig 4H are interesting, but perhaps overstated in the text. Is there molecular evidence to suggest a functional interaction between TGFb and PRRX1? To this end, it would be helpful to treat control and PRRX1 knockdown/knockout CAFs with TGFb and do qPCR for fibrogenic genes, and/or analyze SMAD translocation under the same conditions.

Response:

Thanks for your informative and valuable comment. As you recommended, we performed further studies to provide molecular evidence to suggest a functional interaction between TGFb and Prrx1, as follows:

[1] We quantified the change in the nuclear orientation angle and the number of fibronectin fibers induced by TGFb1 treatment. In wild-type MEFs, the nuclear orientation angle became significantly

anisotropic and the amount of fibronectin increased drastically in response to TGFb1 treatment. However, these TGFb1-induced changes were not observed in the Prrx1 mutant MEFs. These results have been added in the revised Fig. 5B.

[2] As you recommended, we treated control WT and Prrx1 knockout or mutant fibroblasts with TGFb1 and examined the expression changes of fibrogenic genes as follows:

We confirmed that TGFb-induced enhancement of these representative fibrogenic genes was not observed in Prrx1 mutant MEF cells, indicating that Prrx1 is indispensable for TGFb-induced fibroblast activation.

[3] We studied physical and functional interaction between Prrx1 and Smad3 protein. We further investigated the Prrx1 binding regions revealed by Prrx1 ChIP-seq. Interestingly, the Smad3 binding motif was frequently observed 75 bp away from the binding site of the Prrx1 binding site, indicating that Prrx1 may form a DNA-binding complex with Smad3 through physical binding. We observed that Prrx1 was bound to Smad3 through co-Immunoprecipitation experiments. Furthermore, we observed the TGFb-induced colocalization of Prrx1 and Smad3 in the nucleus of MEFs. This phenomenon was not observed in Prrx1 mutant MEFs. These findings strongly suggest that Prrx1 is an important part of TGFb signaling in fibroblast activation. The above data are presented in the

revised Fig. 5 as follows:

8. The results in Figure 7 do not necessarily support a role for PRRX1 in cisplatin resistance. As genetic inhibition of Prrx1 alone profoundly reduces tumor growth in this model, the authors cannot conclude that the CRs observed in the combination-treated group are due at all to cisplatin treatment (similar responses here +/- chemo). This role might be supported by histological analyses to show differences between Prrx1-depleted and Prrx1-depleted plus cisplatin, or by survival analysis. Otherwise, the conclusions should be tempered accordingly.

Response:

As you commented, we have deleted all sentences suggesting the role of Prrx1 in cisplatin resistance.

Reviewer #5 (Remarks to the Author): with expertise in single-cell transcriptomics, tumor microenvironment

The manuscript presented by Keun-Woo Lee, So-Young Yeo, Jeong-Ryeol Gong and colleagues characterises the role of the transcription factor Prrx1 in the specification of myofibroblasts and its impact on wound healing and tumorigenesis.

This manuscript presents a wealth of information including in silico analysis of public datasets, molecular analysis, in vitro and in vivo models for loss and gain of Prrx1 function, functional assays and experiments with patient-derived material.

Most of the conclusions claimed by the authors are supported by experimentation and experiments are generally well designed and executed. There are however key some concerns that need attention. Of particular importance, given the title of the manuscript, are the concerns associated to the epigenetic characterisation presented in Figures 4 and 5. As it stands, the title of the manuscript should be carefully reviewed.

Response:

We appreciate your constructive and critical comments to improve our manuscript. We performed additional analyses and extensive additional experiments, including CHIP-seqs and ATAC-seqs, to demonstrate the robustness of the epigenetic characterization. Detail responses have been described under each comment below. The title of the manuscript has also been revised.

Here is a summary of the concerns:

1. The manuscript includes extensive information and in some instances, this could be streamlined to focus on the main claims of the manuscript. I suggest evaluate redundancies and only keep the necessary information to support the claims and the main message of the manuscript.

Response:

We acknowledge your concerns. We have re-organized the manuscript with the necessary information supporting the results based on the main message that Prrx1 is a master transcription

factor for a subgroup of fibroblasts with myofibroblastic phenotype and the experimental and analytical evidence supporting it. We believe that the figure panels in a figure were excessively complicated to read, therefore, we have sub-divided and simplified the figure panels according to the main message and ensured that the number of figures is within the limit allowed by the journal policy (up to 10 items) to improve readability.

2. Perhaps associated to the point above, there are some inconsistencies between the text and figures, to mention a few, in Figure 1 the text mentions 7 tumour types but in the figure, there are only 6 (head & neck seems to be missing). In Figure 2a, the text says 10/10 but the figure says 6/6. Figure 2E states 72.73% (2/13). S9 panels D and G have the same legend...etc. The manuscript should be carefully reviewed.

Response:

We apologize for such a confusing description. We used a variety of datasets in this study. A total of seven cancer types (colorectal cancer from two different studies) from eight datasets of scRNAseq were included (Supplementary Fig. S1). Among those eight datasets, six included scRNA-seq data from both tumor tissue and normal tissue, which are presented as the main Fig. 1A. We acknowledge that these were confusing. Thus, we have mentioned the exact number of each dataset used in each figure. For Fig. 2, we have corrected the mismatch number between the Fig. and main body (6/6 in Fig. is correct). The text in the S9 panel has also been corrected appropriately. The main text and figures in the revised manuscript have been reviewed carefully.

3. In Figure 1, what is the origin of the normal fibroblasts? What is their activation score as assessed in panel 1A? Are these comparable between cancer types?

Response:

In Figs. 1A-D, we analyzed public single-cell sequencing data (Fig. 1A), and the fibroblasts were extracted from the single-cell sequencing data using known fibroblast marker genes (Supplementary method section with heading "Probabilistic single-cell annotation for 11 cell types in eight datasets"). The normal fibroblasts were indicated in the extracted fibroblasts in normal tissue whereas CAFs were indicated as fibroblasts in cancer tissue from the single-cell sequencing data. We have clearly defined the normal fibroblast used in Fig. 1.

The activation score (activated CAF score) was calculated using the fibroblast activation gene set

based on the study by Mahmoudi *et al.* (Nature 574, 553–558, 2019). The activation score for every single fibroblast from single-cell sequencing expression data was estimated using the AddModuleScore function of Seurat to calculate the score for feature expression of the fibroblast activation gene set. Therefore, this score represents the status of activated fibroblast (Nature 574, 553–558, 2019). We have added this description to the revised manuscript with refer bottomnce.

Concerning the question “Are these comparable between cancer types?” raised by you, we acknowledge that there may be a concern for batch effect in the single-cell sequencing data between cancer types and the different roles of fibroblasts between cancer types. However, the comparison in Fig. 1D between the fibroblast activation score and TF expression was caused by every single organ, but not the comparison between cancer types. Therefore, Fig. 1A indicates that overexpression of Prrx1 is correlated with activated fibroblast across multiple cancer types.

Additionally, when we need to combine two different datasets, such as our analysis of single-cell sequencing data from two colorectal cancer cohorts (revised Fig. 7I), the batch effect was removed using the canonical correlation analysis (CCA) algorithm in the Seurat R package (version 3.1.5).

4-1. What is the level of expression of Prrx1 after Cas9 deletion and upon Dox-mediated overexpression, and what is the baseline level of expression of Prrx1 in WT fibroblasts? Wound healing data

Response:

In this study, to confirm the impact of Prrx1 expression and tumor growth *in vivo*, we established the transgenic mouse model of CRISPR/CAS9 system and doxycycline-induced model in immunocompetent mice. As presented in Fig. 2A, we injected the tumor cells into mice and introduced sgRNA-guided deletion of Prrx1. To identify the deletion of Prrx1 in the expected target tissues, we performed immunohistochemistry on paraffin sections obtained from mice tissues using Prrx1 antibody and then analyzed the expression of Prrx1 (below figure).

Wound healing in vivo figure

As presented in Supplementary Fig. 7, we added quantification of Prrx1 in targeted tissues. Similarly, quantification of Prrx1 was conducted from the Doxycycline-induced mice model, which has also been added in the same section of Supplementary Fig. 7 and Fig. 2D. In addition to the tumor grafts model, we quantified the wound healing model *in vivo* to exhibit the expression of Prrx1. The baseline level of Prrx1 was not detected within WT fibroblasts mice tissue.

4-2. How do these levels compare to the different species of fibroblasts described in Fig 1C.

Response:

Human CAF transcriptome data presented in Fig. 1C is single-cell RNA seq data. However, the transcriptome data of these fibroblasts (MEF, wound healing fibroblast, MMTV CAF) produced in this study are bulk RNA-seq data of *in vitro* cultured mouse fibroblasts. Since mouse CAF bulk RNA-seq and human scRNA-seq data were generated using completely different sequencing platforms, it is very difficult to correct the batch effect of these datasets. Therefore, mRNA levels of Prrx1 cannot be directly compared between two types of datasets.

As an alternative approach to handle this issue, we assessed the level of Prrx1 protein instead of RNA in these cells. Specifically, we performed western blotting and IHC staining to examine the protein expression of Prrx1 between human and mouse fibroblasts (below figures). For a more accurate assessment, we used the same Prrx1 antibody on both human and mouse fibroblasts (Prrx1 Mouse Monoclonal Antibody [Clone ID: OTI1E10], origin. CAT#: TA803116). Consequently, the quantified protein levels of Prrx1 of human colon CAFs were comparable to those of mouse CAF (MMTV-CAFs). Additionally, there was no significant difference in the protein levels of Prrx1 in stromal fibroblasts between human and mouse tissues. In conclusion, Prrx1 is highly expressed in both human and mouse activated fibroblasts without any considerable difference in its levels.

4-3. There is abundant literature that reports transcriptional signatures to define at least 2 different types of CAFs in human and mice (PMID: 29988129 (Ref#23), PMID: 29198524, PMID: 30514914, PMID: 33852842, PMID: 32790115) what signatures overlap with high and low Prrx1 populations.

Response:

As per your comments, we downloaded the scRNA-seq datasets of CAFs of various human and mice cancers, such as PMID:29988129 (Ref#23), PMID:29198524, PMID:30514914, PMID:33852842, and PMID:32790115, and assessed the Prrx1 levels in those CAFs datasets. The results are summarized as follows:

Overall, Prrx1 tends to be more highly and frequently expressed in myofibroblast-like subtypes of human CAFs across the multiple human cancer types, such as head & neck squamous cell carcinoma (PMID:29198524), human triple-negative breast cancer (PMID:32790115), and human lung cancer (PMID:29988129). In contrast, such preference of Prrx1 expression for myofibroblast-like subtype was not observed in transgenic mice (MMTV-pyMT)-based dataset. In these transgenic mice, there was no significant difference in Prrx1 expression based on the subtype of CAFs. In the case of these transgenic mice, the oncogenesis process and the tumor microenvironment differ greatly from those of human cancer, which may be partly responsible for these differences.

A

5. Results in Figure 2c are intriguing. The authors sought to explain the distant activation of Prrx1 after a wound-healing effect. It seems there is a distant hotspot of Prrx1 activation but it seems widespread activation to some extent in the whole animal. Can authors elaborate more on this point? From this part of the manuscript, it would be interesting to add an explicit narrative in the text highlighting similarities/differences of Prrx1 activation in the context of wound healing and cancer. This is important because the title specifies cancer context but rather the results seem to point to a general function of Prrx1 in myofibroblasts. In which case

the title should be reworded accordingly as this is not CAF specific. Some of the wound-healing effects of Prrx1 enhancer has been already reported PMID: 31278164.

Response:

We thank you for this suggestion. To elaborate on the distant hotspot of Prrx1 activation, we performed additional experiments of the wound-healing assay *in vivo* as presented below. The distant hotspot was located in the pericardial space and mid mediastinum as observed by bioluminescence imaging.

Was this hotspot always localised in the same area? What is the Luc expression of not wounded mice after Dox exposure? (it seems here "Control" is not really a Col1a2 reporter mouse).

Response:

We repeated these experiments more than thrice, and the hotspot was always localized in the same area. We microscopically examined this hotspot exhibiting tumor metastasis.

One of the interesting findings was the multiple inflammatory lesions located along the pleural surface of the lung, pericardium, and mid-mediastinum as presented below. We suspected that these inflammatory lesions appeared as the hotspots in bioluminescence imaging. A strong Prrx1 expression was observed in these inflammatory lesions. A closer examination of these lesions revealed that these were composed of proliferating fibroblasts and blood vessels. These findings were perfectly compatible with the pathologic diagnosis of inflamed granulation tissue, which is usually observed in aberrant wound healing reactions, chronic inflammation, and cancer stroma.

The other interesting finding was that the tumor recurred or metastasized on these inflammatory lesions as presented below. This finding strongly suggests that Prrx1-expressing fibroblasts and their associated niche are strongly related to tumor recurrence and metastasis.

We also checked the Luc expression of unwounded mice after Dox exposure as presented below. Expectedly, no signal was detected in the entire body.

What is the basis for this specific site for hotspot activation? How do the authors rule out there is a higher concentration of Dox in the heart mediastinum? Did the tumour relapse at the incision site?

These questions are important to evaluate if there is a specific metastatic tropism in Prrx1 overexpressed mice, or this is dependent on how much Prrx1 expression is achieved in each tissue. If metastasis correlates with high Prrx1 expression it might also appear in form of relapse in the primary hotspot around the wound produced by the tumour resection.

Response:

We suspect that this phenomenon is related to the intrinsic properties of Col1a2-rtTA transgenic mice. The transgenic mice may express their target genes in aberrant tissue depending on the insertion site of genes in chromosomes. Such aberrant gene expressions in transgenic mice have been reported in multiple studies as follows:

- Position-independent, aberrant expression of the human ornithine decarboxylase gene in transgenic mice (Biochem Biophys Res Commun 180:1; 262–7, 1991).
- Aberrant tissue specific expression of the transgene in transgenic mice that carry the hepatitis B virus genome defective in the X gene (Arch Virol 132:3–4; 381–97, 1993).
- Transgenic mice aberrantly expressing human ornithine decarboxylase gene (The journal of Biological Chemistry 266:29;19746–19751, 1991).

Although the detailed mechanism of this phenomenon has not been identified, this is certainly a highly reliable and reproducible phenomenon in this transgenic mouse. Therefore, from a practical point of view, this mouse can be used as a good experimental model for the study of tumor recurrence and metastasis.

Microscopic examination suggested that the inflammatory lesions on the pleural surface of the lung, pericardium, and mid-mediastinum were the hotspots. Since the pleural surface is an anatomically exposed open structure, it is very less likely for Dox to concentrate in this space unless pleural effusion occurs. Additionally, pleural effusion was not observed in any mice.

We observed that the tumor did not relapse at the incision site. Notably, we sutured this incision site with threads after surgical resection of the tumor. We suspect that the suture of the incision site may be the main reason for the absence of tumor relapse in the incision site since the suture suppresses the inflammatory reaction by reducing tissue tension and accelerating epithelial closure. In summary, we speculate that the initial surgical incision might have provoked hotspot reaction around the mid-mediastinum, however, suture of the incision site inhibited the prolonged inflammation in incision sites.

What is the luciferase signal after tumour implantation, does the hotspot dynamic (e.g. time span) change from sham and tumour implanted mice?

Response:

Although your suggestion is very interesting, it was practically difficult to monitor fibroblast-specific luciferase signal after tumor implantation since luciferase was inserted into both grafted LLC1 tumor cells and fibroblasts of recipient transgenic mice. Additionally, luciferase signal hotspot around the mid-mediastinum was detected after surgical resection of primary tumor mass in tumor implanted mice and wound excision in sham mice.

6. The desmoplasia phenotype is not convincing. How is fibroblast proliferation without tumour? What is the desmoplastic phenotype during wound healing instead AOM. Following my previous comment, is this a common phenotype driven by Prrx1 or is it cancer-specific?

Response:

In this study, we intended to describe "desmoplasia" as increased fibrosis based on its meaning provided by the dictionary. We further included Masson's trichrome staining to reveal the fibrosis, and quantitative analysis of the staining area using the Image J program revealed increased fibrosis in Dox (+) (figure provided below; Fig. 2D, left), which indicates that Prrx1 is required for stromal fibrosis. Additionally, SMA+ and Prrx1+ stromal fibroblasts significantly increased in Dox (+) group

(figure provided below; Fig. 2D, right). We have defined “desmoplasia” as increased fibrosis more clearly in the revised manuscript.

It is well-known that this concept is also observed in wound healing tissue and is sometimes described as a scar if it remains a permanent defect. In our previous study (Yeo *et al.* Nat comms 3016, 2018), Prrx1 was transiently expressed during the wound healing process (figure provided below), and increased expression of Prrx1 in wound healing fibroblasts may suggest that they are unlikely to be cancer-specific. At the end of the normal wound healing process, Prrx1-positive fibroblasts disappeared, however, we believe that desmoplasia will occur if Prrx1-positive fibroblasts continue to exist, like CAF. This is also supported by Froidure *et al.*'s study demonstrating that Prrx1 is the key regulator of idiopathic pulmonary fibrosis (IPF) fibroblast.

Therefore, we have modified the title of this manuscript to fibroblast rather than CAF to represent

this concept that the myofibroblastic feature of Prrx1+ fibroblast is not cancer-specific.

7. What is the basis of the gating for figure 3C. A non-stained control should be shown. Representative images are shown in Figures 3C and D but no statistical analysis nor extent of replicates performed is shown.

Response:

We attempted a triplicate of the FACS-based apoptosis assay as presented below to obtain more accurate results. Additionally, we added unstained control (Fig. 3B provided below) and quantitative graphs of the gel contraction assays performed in triplicate (Fig. 3C provided below) in the revised manuscript.

8. The results shown in Figures 4 and 5 are intriguing and open a number of key questions.

8.1. The authors use a super-enhancer mapper to define super-enhancers based on H3K27ac and H3K4me3 profiles. The typical definition (and peak profile) of super-enhancers are large >12.5Kb clusters of H3K27ac, consistent with some of the results shown by the authors, for example in figure S9, where the mean length of SE range from ~19 to 46Kb in size, however, the results shown in figure 4 and particular 4C does not seem to be consistent with this definition, rather is consistent with a typical promoter definition. The authors should clearly define super-enhancers and apply appropriate visualisation in the figures to evaluate their presence and enable comparisons.

This SE definition is again repeated in figure 5, particularly concerning the SE definition in ADAMTSL.

Response:

We acknowledge your concern. We also recognize that super-enhancer could be discovered by enhancers with large lengths (Proc. Natl. Acad. Sci. U S A. 110, 17921–17926, 2013).

However, in this study, we adopted a more widely accepted definition of super-enhancer proposed by Pott, Sebastian, and Jason D. Lieb. ("What are super-enhancers?" Nature genetics 47.1, 8–12, 2015) (figures provided below). According to this definition, super-enhancer is determined by signal intensity (read count) instead of the length of enhancer.

The detailed super-enhancer discovery process is presented above. The whole process is composed of three steps:

[Step 1] First, typical enhancers have to be identified.

[Step 2] Enhancers closer than 12.5 kb to other enhancers are merged into one new enhancer, called a stitched enhancer. (Step 2, figure provided above). Therefore, the enhancers identified in step 1 were reclassified into stitched enhancers or remaining individual enhancers.

[Step 3] Further, these redefined enhancers were compared in terms of ChIP-seq intensity (normalized read count) by plotting the distribution of ChIP-seq intensity values. Generally, a few enhancers have an extremely high ChIP-seq signal intensity whereas most enhancers have very low

signal intensity. In our study, enhancers with exceptionally high signal intensity are defined as super-enhancers. The criterion to divide the group with high signal intensity from the group with low signal intensity was the point where the slope becomes one in the hockey-stick plot presented above.

When Med1 Chip-seq was not available, other Chip-seq, including H3K27ac, was also used for the peak intensity (normalized read count within each enhancer genomic region) as Y-axis. Therefore, when using this definition, SE could theoretically be less than 12.5 kb in size. Indeed, according to Hnisz *et al.*'s study (Hnisz *et al.* "Super-enhancers in the control of cell identity and disease." Cell 155, 934–947, 2013), the average length of super-enhancer is 8.6 kb, whereas the mean length of a typical enhancer is 0.7 kb as presented below. We have mentioned this definition with citations of the relevant references in the revised Supplementary method section.

In this study, based on this principle, we identified the SE of fibroblasts and CAFs. Specifically, first, we identified enhancer peaks from H3K27ac ChIP-seq using the MACS2 algorithm. Second, we identified clusters of peaks within 12.5 kb (stitched enhancers) and figured out the peak intensity of the clusters. Third, according to the definition of super-enhancer, enhancers whose slope was greater than one in the hockey-stick plot of signal intensity were selected (main Fig. 4 for mouse fibroblast and Supplementary Fig. 15 presented below).

As per your comment, we double-checked the signal intensity of ADAMTSL, which had a short length; however, the peak intensity ranks (slope of the plot >1) were sufficient to be SEs as per the criteria used in this study. Nevertheless, the original Fig. 5D including ADAMTSL was less informative, therefore, we have modified it in the revised manuscript.

8.2. It is also concerning the results shown for Prrx1 ChIP. What is the peak calling analysis for this ChIP, the data seems quite noisy, how many replicates have been performed of each ChIP? As a transcription factor-based ChIP it might need more replicates to ascertain the noise level. Additionally, a comparison of an IgG control is necessary for these ChIPs. What is the scale for the visualisation of 4C? Can authors specifically define what do they interpret as a SE in 4C?

Response:

We acknowledge the issue raised by you. The peak calling analysis was performed using the MACS2 peak caller tool, which is one of the most widely used programs. During the revision, we conducted repeated experiments for Prrx1 ChIP-seq. Consequently, we performed Prrx1 ChIP-seq and IgG ChIP-seq twice for three types of murine fibroblasts. Further, we conducted a comparison

with an IgG control as per your recommendation. In the repeated experiment, the results were produced in the same way as in the original experiment (figures presented below). The peaks of Prrx1 were repeated in these additional experiments (repeated Prrx1 ChIP-seq and comparison with an IgG control), and we also reconfirmed that there was no difference in the phenotype through GO analysis, such as ECM stiffness, TGF-beta signaling, angiogenesis, and EMT. The updated results of the repeated Prrx1 ChIP-seq with IgG control have been summarized in revised Fig. 5A. We have also mentioned that Prrx1 Chip-seq was duplicated and IgG was used as a normal control in the Supplementary Method section.

Additionally, we have added a scale bar to H3K27ac peak plots of Fig. 4C (revised fig. 4F). These super-enhancers in revised Fig. 4F as presented below were identified according to Richard. A. Young's criteria (following the procedure described in Comment #8.1).

8.3. Additionally, the visualisation of super-enhancers is not appropriate (for example in 4I or Fig S9 B-G and subsequent figures) should be performed with a wider scale (not just +/-1kb or 2kb of the TSS) as these regulatory regions are large in nature. The visualisation presented is just not informative for SE.

Figure S9 B -G the graphs should be shown in the same graph WT vs Prrx1 deletion to allow appropriate comparison. Attention should be given to the scale of these graphs.

Response:

We agree that the visualization presented in the original figures was not informative. Therefore, we have modified the figures to be more informative (Fig. 4) in the revised manuscript. Additionally, Figs. S9B-G have also been modified to have the same range of scale in the same graph (Fig. S13).

8.4. Conceptually the results also open some key questions that need to be addressed. The data presented shows that Prrx1 KD results in a wide reprogramming of SE, however, the super-enhancer reorganisation is very different in the 3 different fibroblasts tested with only 6 SE in common. If Prrx1 is a master regulator of the myofibroblast fate, a more consistent reprogramming should be expected. Subsequently and given the low overlap, the functions presented although consistent with a myofibroblast functional fate are not consistent between the models, rather each model presents certain aspects of each of these functions. What are the specific functional modules underpinned by each of these models? This is also applicable to 4L. What is the overlap between the Prrx1 bound genes in the 3 fibroblast models? As a

keeper of specific cell identity, these should extensively overlap.

Thus this contradicts one of the main claims of the manuscript which is assigning a master role to Prrx1.

Response:

Although the overlap of individual SEs of the three fibroblasts affected by Prrx1 knockdown was low, the overall functional pathways of SEs affected by Prrx1 were highly overlapped between cells as presented below. Therefore, the specific functional modules supported by Prrx1 were considerably similar for each cell, most of which are related to the function of myofibroblasts, as presented in the figure below.

One of the reasons for this discrepancy in the Prrx1 KD results maybe that cell identity is determined not by one master Transcription Factor (mTF) alone but by a combination of multiple mTFs. For instance, the identity of embryonic stem cells is determined not by a single mTF but by a set of mTFs, such as Oct4, Sox2, Nanog, and Klf4. Similarly, the identity of fibroblasts is highly likely to be determined by a set of mTFs instead of a single mTF alone.

In this study, we also identified other mTF candidates of these fibroblasts besides Prrx1. By looking at the list of mTF candidates of each fibroblast, we can receive profound insight into the key cell-identity determining transcriptional circuit called "core regulatory circuit (CRC)" of these fibroblasts. As presented below, these three fibroblasts have eight mTF candidates in common and additional different mTFs in their CRC. Since a significant fraction of the CRC member (mTF) is shared by all three fibroblasts, these fibroblasts exhibit similar properties. In contrast, there is also

a slight difference in the composition of the CRC members, which may explain the response to the Prrx1 knockdown.

Ultimately, it is mandatory to discover and validate all the mTF candidates of each fibroblast. In this study, we rigorously confirmed that Prrx1 satisfied all the criteria proposed by Young *et al.* for being an mTF^{28,29}. Additionally, we also confirmed that Prrx1 has a causal relationship with the myofibroblastic phenotype through rigorous *in vitro* and *in vivo* experiments, including KD and O/E experiments (figures provided below: Supplementary Fig. 13C and figures for the response of the comment 8.2). These figures indicate that Prrx1-specific super-enhancer-associated genes have a similar function in terms of gene ontology. We further revealed that Prrx1 builds a complex with SMAD in three types of fibroblast for TGF- β signaling, which was associated with the myofibroblastic phenotype (figure provided below: revised Fig. 5C-H).

As per your comment, we reanalyzed Prrx1 binding sites and Prrx1-associated genes. Common Prrx1 peaks compared to IgG control were identified from three types of murine fibroblasts. We inferred the genes which were the nearest within +/- 1 Kb from these common Prrx1 peaks. From these results, we identified 515 overlapped genes in three types of fibroblasts, and GO analysis

determined these genes enriched in activated fibroblast, such as ECM stiffness, TGF-b signaling, angiogenesis, and chromatin changes.

8.5. Another question that might explain the disparate reprogramming is perhaps the initial state of these fibroblasts as they come from very different origins. What is the level of Prrx1 expression in all these lines before manipulation? How does this compare with the Prrx1 high/low definition?

Response:

As you pointed out, these fibroblasts came from considerably different origins and the initial state of these fibroblasts was likely to be different. However, the level of Prrx1 expression in all these lines did not differ significantly between cells as presented below.

For the Prrx1 high/low definition, the criterion for the Prrx1 high/low expression is determined empirically by considering the overall pattern of Prrx1 expression in cells. Generally, since the expression of Prrx1 in each cell is divided into a high and low group without an intermediate group, it is not difficult to distinguish Prrx1-high from Prrx1-low groups. Practically, the Prrx1-low group is closer to the state in which Prrx1 expression is completely absent.

We speculate that the contrasting reprogramming may be secondary to the difference in the composition of core regulatory circuit (CRC). CRC is composed of a set of master transcription factors (mTFs) and determines the cellular identity. These three mouse fibroblasts have Prrx1 as a common mTF albeit are most likely to have additionally different mTFs for each fibroblast. Furthermore, transcription factors, including Prrx1, require cofactors that may be different among these fibroblasts. These things may account for the detailed difference in the myofibroblastic program executed by Prrx1. However, there is no change in the fact that Prrx1 executed essentially the same myofibroblastic program in all these fibroblasts

This is further shown in the heterogeneity found in the 9 CRC CAF cultures shown in figure 5. where authors find two very distinct SE organisations. How is the level of Prrx1 in these lines? Given the results in Figure 1, a more direct classification of the 9 CRC CAF explanted fibroblast could just rely on Prrx1 expression. Are these explanted cell lines pure clones in nature or present heterogeneity of expression of Prrx1?

Response:

For the nine CRC CAFs, the measured expression levels of Prrx1 and CAFs presented relatively high expression of Prrx1. Since the levels of Prrx1 expression were similar to each other, it seems that there were some limitations in classifying the CAFs only with Prrx1. Additionally, these data indirectly suggest that the super-enhancer action through protein complex formation can have a greater effect on actual cell function than the expression level of a single gene since the CAF-B group (CAF5 ~ CAF9) had Prrx1 expression but not super-enhancer for Prrx1. They also did not show a myofibroblastic phenotype. We acknowledge that confirming the heterogeneity of the CAFs is an effective suggestion. One of the best ways to confirm the intracellular heterogeneity in single cell sequencing, which was unfortunately not convenient secondary to the cost and cell preparation. However, in the additional experiments of Prrx1 knockdown in nine CAFs, the number of cells decreased and apoptosis increased (figure presented below: Revised Supplementary Fig. 18), which suggests that most cells in each CAF group express Prrx1.

9. In figure 5 authors find two main groups of fibroblasts in CRC patients. Clinically what is the survival data associated with these patients, is this consistent with the KM graphs shown in the in silico analysis?

Response:

We appreciate your interesting suggestion. The groups were identified from cultured CAFs from nine patients with CRC, and the groups were classified based on the super-enhancer pattern. Therefore, it was difficult to compare the survival characteristics of the groups in these small numbers of patients with short follow-up survival data owing to no available genomic data for all patients. However, we investigated TCGA colorectal cancer set (n=623) with RNA-seq gene expression data and survival data. We focused on Cluster A since this group was associated with Prrx1 and patients with high Prrx1 in CAF revealed worse prognosis (Fig. 1 E-F). Using the 463-target gene set for Cluster-A, the enrichment score using the GSVA program was calculated to represent the signature of the 463-target gene set in each patient. Further, we identified a worse prognosis of patients with high Cluster-A signature compared with patients with low signature, specifically in disease-free survival (revised Supplementary Fig. 16D presented below). Although this approach was indirect, the poor survival tendency by Cluster-A signature was consistent with the KM graphs depicted in Fig. 1F, and this signature represents super-enhancer signature than only Prrx1 expression level.

Secondly, authors characterise cluster A extensively and basically keep cluster B as “not A”;

apart from the obvious value of this analysis the results from cluster B can also validate the functional definition of cluster A. What are the genes associated with cluster B?

What are the differential SE overrepresented in cluster B over A, what are the functional modules associated with cluster B?

Response:

We acknowledge your concern. We further investigated the characteristics of cluster B. Differential SE overrepresented in cluster B included *FOXF1*, *NKX2-3*, and *PITX1*. We have revised the relevant figures and results to include these findings (blue arrowhead in the figure presented below: Revised Fig. 7B).

Additionally, we identified that cluster B was characterized to represent senescence fibroblast with DNA damage and senescence-associated genes in aging cells by super-enhancer-associated genes and highly expressed genes from cluster B. Therefore, we believe that cluster B is compatible with the inactivated fibroblast group to be senescence (revised Figs. 6D-6E provided below).

Fig. 6

What is the distribution of K27Me3Ac in cluster B and how does that correlate with the distribution of peaks in cluster A, from figure 5B it seems most of the identified peaks are flanking the TSS.

Response:

Overall distribution (pattern) of K27M3AC of cluster B was different from that of cluster A; cluster A and cluster B were separately grouped by PCA analysis using H3K27ac; this finding was repeated using independent ATAC-seq, which was performed during revision (revised Figs. 6A-C). Therefore, we aimed to demonstrate the difference between cluster A and cluster B. The results were summarized in original Fig. 5B. However, the original Fig. 5B had low resolution and was unclear. Therefore, we have removed this figure and replaced it with a PCA plot (revised Fig. 6B-C) and reconstructed the figure by summarizing the different super-enhancers between two groups (revised Figs. 7A, 7B, 7D).

The authors' interpretation of the SE distribution of the 9 CRC CAF lines is that they belong to two different groups based on Prrx1 biology, however, there is an alternative interpretation of this data irrespective of biology. Cluster B presents a lower number of loci, there are fewer SE identified and these are smaller (Fig S12), thus these data can be also interpreted as technical bias, due to potentially lower quality ChIPs for some of the cell lines analysed. In this case cell lines with high-quality ChIPs would be represented by cluster A and cell lines with low-quality ChIPs would form cluster B. Replicates will be necessary to address whether these 2 clusters are due to technical variation rather than biology. This hypothesis is further supported by the results in Figures 5E and 5F. As mentioned before, authors focus on Cluster A (high-quality ChIPs) but do the same analysis stands for cluster B, can authors identify direct correlations with the SE defined in cluster B and the gene expression profile? What is the level of transcriptional activity of Cluster A vs B, are these of the same magnitude? Authors should clearly control for quality biases and show this cluster A/B definition is not driven by quality. Additionally, there is a third possibility that is not addressed in the experiments and it is the potential of senescence associated with cluster B, this is further supported by the results in panel 5I, where cluster B-like terminal shows overall less activation, and thus this could explain the overall reduced output of the K27me3ac ChIPs. Do Prrx1high fibroblast present canonical markers of CAF activation? For example, as defined in PMID: 20839032 To rule out technical biases and conclusively compare the K27me3Ac within these lines a spike in ChIP seq

normalisation might be required.

Response:

As per your comment, cluster B presented a lower number of loci and fewer SE, which were small (Fig. S12E-H). We investigated whether these data can be interpreted as technical bias, owing to potentially lower-quality ChIPs for some of the analyzed cell lines. We calculated the fraction of reads in peak (FRiP) and read depth- the most widely accepted QC measure using the ChIPQC R package. The result was that initially submitted ChIP-seq data passed most of the criteria (read depth >10 M and FRiP >0.2 (optimal) specified by the Encode consortium QC standard guideline).

However, to further eliminate the possibility that these 2 clusters were secondary to technical variation rather than biology, we further repeated H3K27ac ChIP-seq for 'Cluster A' CAFs twice and 'Cluster B' CAFs thrice. Consequently, all the ChIP-seq data passed QC criteria set by the Encode consortium (read depth >10 M and FRiP >0.2) (Fig. presented below).

Additionally, using this improved data, the same results were reproduced as those of analyzing the initially submitted ChIP-seq data. Particularly, principal component analysis (PCA) of the SE profile obtained from the newly improved ChIP-seq data revealed that these nine CAFs were clustered into two clusters, similar to the results obtained from the initially submitted data (Fig. presented below). This result of the repeated ChIP-seq indicates that there is no significant difference between initial and improved SE profiles. Overall, it indicates that data quality did not affect CAF clustering and these two clusters were secondary to CAF biology rather than technical issues.

Furthermore, we validated the characterization of these clusters by applying another independent analytical technique, ATAC sequencing, to all nine CAFs. Based on the ATAC-peak pattern obtained by this ATAC sequencing, nine CAFs were studied through PCA analysis. Based on the results obtained through ChIP-seq, these nine CAFs are redivided into two identical groups: Cluster A and cluster B (revised Fig. 6C presented below).

To further validate the SEs identified using H3K27ac ChIP-seq, we investigated the overlap between ATAC peaks and SEs and that between ATAC peaks and typical enhancers (TEs), respectively.

SE was previously reported to overlap with the ATAC peaks more than TE. Most of the SEs (>99%) overlapped with the ATAC peak whereas a significantly lower fraction of TE overlapped with the ATAC peaks (30–85%) (Figs. provided below). It indicates that there is no technical bias in our H3K27ac data. Our results are consistent with those of Lee *et al.* study (Lee, Z., Raabe, M., & Hu W.S. Epigenomic features revealed by ATAC-seq impact transgene expression in CHO cells. *Biotechnol. Bioeng.* 118.5, 1851–1861, 2021).

Additionally, as you suggested, we examined the direct correlation of the SE in clusters A and B with RNA expression profiles. As presented below, SE-associated genes demonstrated a remarkably higher gene expression than TE-associated genes. SE has been known to confer stronger enhancer activity than TEs. This is consistent with our ChIP-seq results. We also examined the level of transcriptional activity of clusters A and B. RNA-seq analysis of these CAFs revealed that the total read counts of these CAFs were similar. This indicates that the levels of transcriptional activity in clusters A and B are of approximately the same magnitude.

Additionally, using the cluster B-specific SE, as mentioned above, we identified that cluster B is

associated with senescence which you suggested as a third possibility. This may explain the reduced output of the K27me3AC ChIPs in cluster B. We have revised the relevant figures and results to include these updated findings in the revised manuscript (revised Figs. 6 and 7, Supplementary Data3).

We have demonstrated that various activated CAF signatures are enriched in 'cluster A' CAFs in revised Figs. 6, 7. Especially, we observed the super-enhancers of Col1A1, which is a representative marker of activated CAF in 'Cluster A' CAF. The super-enhancer profiles of CAFs and gene expression of activated CAF signatures are higher in 'Cluster A' CAF than in 'Cluster B' CAF.

In conclusion, we used diverse epigenetic and transcriptomic data to validate that SEs of CAFs were not technically biased. Using ATAC-seq, RNA-seq from nine CAFs, and H3K27ac ChIP-seq from Encode consortium database, we confirmed that there were two clusters of colon cancer CAFs and 'Cluster B' was not technically biased. However, CAFs of these clusters exhibited senescence phenotypes compared to 'Cluster A' CAFs.

Performing H3K27ac ChIP with spike had practical difficulties. We repeated H3K27ac ChIP-seq for 'cluster A' CAFs twice and 'cluster B' CAFs thrice to perform sufficient replicate experiments. From these results, we ensured that the same results were obtained in all cases.

10. The experiments shown in Figure 5G are not convincing, first apoptosis cannot be assessed by crystal violet, as cell numbers are mostly dependent on cell proliferation and only balanced by apoptosis levels; second, what is the level of expression of Prrx1 in the cells from cluster A and B? And what is the Prrx1 level after knockdown. Controls are missing for this experiment, therefore "transcriptional addiction to Prrx1 is not conclusively demonstrated".

Response:

As you suggested, we assessed the apoptosis of these nine CAFs by examining Annexin V and Propidium iodide (PI) using flow cytometry. The results were summarized as presented below (Supplementary Fig. 18D). In 'Cluster A' CAFs, the number of cells presenting apoptosis remarkably increased in response to shRNA-mediated Prrx1-knockdown. In contrast, such a drastic increase of apoptotic cells was not observed in 'Cluster B' CAFs.

We have added the controls for this experiment as you suggested. The details including controls are presented as follows (Supplementary Fig. 18B-18C):

The expression levels of Prrx1 in the cells from clusters A and B were determined using

quantitative PCR, and the results are presented below.

The expression level of Prrx1 after a knockdown is presented below (Supplementary Fig. 18A).

These data have been included in the revised Supplementary Fig. S18. All the CAFs were immortalized before experiments. This immortalization may affect the Prrx1 level in CAFs, however, the enhancer and super-enhancer do not seem to be affected. This highlights the importance of investigating the epigenetic landscape of cells which reflects the basic identity of cells better than mRNA profile.

11. The pseudotime analysis shows terminal #1 and Terminal #2, again assigning a myofibroblast identity to cluster A (terminal 1-like) but what is the identity of Cluster B terminal #2 remains unknown. Importantly what is the origin of these two terminals, what is the nature of the fibroblast source? As presented in Figures 5H and 5I there are 3 classes of Fibroblasts in the CAFs analysed, namely Source, Terminal 1 and Terminal2. As such the violin plots in this analysis should contain the three cellular states/fates.

Response:

We acknowledge your concern. This analysis was performed using fibroblasts in cancer tissues

from the single-cell RNA seq data. Annotation of fibroblast was described in the Supplementary method section. The fibroblasts in "Source" had shorter pseudotime compared with other fibroblasts such as terminal #1 and terminal #2. Therefore, we believe that the fibroblasts in "Source" represent fibroblasts before further differentiation in terms of pseudotime. CAFs are considerably heterogeneous. Although many studies have been conducted to elucidate the progenitor cells (source) of activated CAF, it is still controversial and difficult to characterize. One of the suggested origins of CAF is bone-marrow-derived mesenchymal stem cells. We identified that the stemness decreased from the "Source" to the "Terminal" using various stem cell signature gene sets (below figure and Fig. 7I). Therefore, we believe that the fibroblasts in "Source" are fibroblasts with stem-cell properties based on the comparison of three cases of the fibroblasts, which also suggests that the fibroblasts in "Source" are in a less differentiated status. These findings are well-correlated with pseudotime lines between "Source" and "Terminal". We believe that the term "Source" is confusing, therefore, we have modified it to "FB before differentiation" in the revised manuscript. Terminal #1 and Terminal #2 were similar with cluster A (CAF-A group) and cluster B (CAF-B group), respectively. Terminal #1 revealed features of activated CAF with myofibroblastic properties whereas Terminal #2 revealed inactivated fibroblast properties.

As per your comment, we have further revised violin plots containing three cellular fates (revised Fig. 7I). In the revised plots, we observed that during the differentiation of CAFs, CAF on the trajectory of CAF A obtained myofibroblastic phenotypes, such as ECM assembly, supramolecule signature, and TGF-beta signatures.

How does the module SE score for cluster A and B correlate with the murine fibroblasts of figure 4?

Response:

We identified eight common candidate master transcription factors (Prrx1, klf4, tead1, fosl2, cebpb, junb, klf13, and nfix) based on the SE from the three types of murine fibroblasts. Among them, three master transcript factors (Prrx1, klf4, tead1) were detected in Cluster-A (CAF-A) group-specific. Prrx1 has been reported to be a common super-enhancer not only in various types of murine fibroblasts but also in human CAF. These points suggest that Prrx1 is commonly involved in the myofibroblastic phenotype, regardless of the type of fibroblast. However, the less overlap of SE between the fibroblast types is believed to be a more complex underlying signal regulated by Prrx1 to induce myofibroblastic phenotype.

12. The results presented in figure 7 are astonishing. What is the mechanism that induces cell death in cancer cells after Prrx1 deletion? How does this compare with the effects of SNF32? Did the cells migrate to metastatic sites? Understanding the effects in cancer cells is essential to propose Prrx1 depletion as a therapeutic avenue.

Are these experiments restricted to resistant cancer cells?

Additionally, as a control, the levels of Prrx1 in the CAFs should be assessed. All 4 arms should be presented in the same graph (panel B) as they form part of the same experiment.

The word “therapeutic” should be removed from “therapeutic deletion” or “therapeutic depletion”, the result and the hypothesis can’t be together in the same definition.

Response:

Since there was no residual tumor in tumor tissues secondary to complete remission after fibroblast-specific Prrx1 deletion, it was practically difficult to study a mechanism to induce cell death in cancer cells of those tissues. Therefore, we devised an alternative based on a spheroid co-culture system that mimics the above-mentioned *in vivo* model, as follows:

To mimic the *in vivo* mouse model used in Fig. 7 (revised Fig. 9), we used the same LLC1 cancer cells and wound healing fibroblasts used in this *in vivo* model. Additionally, we injected sgPrrx1 in this co-culture system after tumor formation, similar to what we did in the *in vivo* model. Interestingly, fibroblast-specific Prrx1 deletion induced by sgPrrx1 caused a massive tumor cell death in this *in vitro* co-culture system similar to that in the *in vivo* model. We further proceeded this experiment by treatment with cisplatin as follows:

Consequently, the combined treatment of cisplatin and sgPrrx1 yielded better results, suggesting that there is some synergistic effect between them. Specifically, apoptosis was observed much more

frequently in the sgPrrx1-treated group than in the sgNS-treated group, and the effect was more pronounced when cisplatin was co-administered. These results have been added in the revised Supplementary Fig. 22.

Additionally, we demonstrated related mechanisms of this effect in Fig. 3G (revised Fig. 3F) as presented below. This figure presents the effect of fibroblast-specific Prrx1 deletion on cancer cells (LLC1) using RNA-seq.

In Fig. 1 and Supplementary Fig. S5, we demonstrated that SNF32, a normal stomach fibroblast, acquired a strong tumor-promoting ability by the expression of Prrx1. Although fibroblast-specific Prrx1 deletion study in Fig. 7 (revised Fig. 9) revealed the significance of Prrx1 through loss-of-function studies, the MKN28/SNF32-Prrx1 study (Fig. 1/Suppl Fig. S5) confirmed the significance of Prrx1 through gain-of-function studies. One of the interesting findings was that SNF32-Prrx1 induced lung metastasis of MKN28. In those metastatic lesions, a large portion of fibroblast close to the cancer cells was positive for aSMA, Prrx1, and FLAG-tag as presented below, suggesting Prrx1-expressing SNF32 in the primary tumor site may migrate to metastatic lesions.

Lung metastasis images from MKN28+SNF32 PRRX1 O/E co-injection

Finally, as you pointed out, we have added the results of the four groups on the same graph as presented below (Revised Fig. 9B) and removed the term "therapeutic" from the revised manuscript.

Reviewers' Comments:

Reviewer #1:

Remarks to the Author:

I am pleased to say that the author have nicely addressed all our comments and I believe the paper is now ready for publication.

Reviewer #2:

Remarks to the Author:

Overall the authors have done a thorough job responding to my and the other reviewer questions. This required an impressive amount of work given 4 lengthy reviews and a 70 page rebuttal. As a result, the revised manuscript is significantly improved and should be published without further revision.

Reviewer #4:

Remarks to the Author:

The authors have meaningfully addressed my concerns from the original submission.

Reviewer #6:

Remarks to the Author:

The authors have provided a comprehensive revision of the original manuscript, which already contained a wealth of information on the role of Prrx1 on fibroblast fate specification. The authors have made a considerable effort clarifying some of the points raised in my original review with a better-tuned language, new analysis, and inclusion of new experiments to shed light on areas where the manuscript wasn't clear or where further investigation was necessary to justify the conclusions made. These include new visualisation of the data, the addition of more replicates to the existing datasets (eg ChIP), new controls, mouse and organoid experiments and an array of new data to better back up the conclusion drawn.

Authors have also provided justified responses when the practicality for generating new data was not possible and attempted to address every single concern raised.

Overall, new key information has been satisfactorily incorporated in the manuscript and as a result, this reviewed version has increased in quality. This manuscript is well executed and it will clearly be of interest to the readership of nature communications.

I have no further comments.

REVIEWERS' COMMENTS

Reviewer #1 (Remarks to the Author):

I am pleased to say that the author have nicely addressed all our comments and I believe the paper is now ready for publication.

Author response:

We appreciate full evaluation and very constructive critical comments on our manuscript.

Reviewer #2 (Remarks to the Author):

Overall the authors have done a thorough job responding to my and the other reviewer questions. This required an impressive amount of work given 4 lengthy reviews and a 70 page rebuttal. As a result, the revised manuscript is significantly improved and should be published without further revision.

Author response:

We appreciate full evaluation and very constructive critical comments on our manuscript.

Reviewer #4 (Remarks to the Author):

The authors have meaningfully addressed my concerns from the original submission.

Author response:

We appreciate full evaluation and very constructive critical comments on our manuscript.

Reviewer #6 (Remarks to the Author):

The authors have provided a comprehensive revision of the original manuscript, which already contained a wealth of information on the role of Prrx1 on fibroblast fate specification. The authors have made a considerable effort clarifying some of the points raised in my original review with a better-tuned language, new analysis, and inclusion of new experiments to shed light on areas where the manuscript wasn't clear or where further investigation was necessary to justify the conclusions made. These include new visualisation of the data, the addition of more replicates to the existing datasets (eg ChIP), new controls, mouse and organoid experiments and an array of new data to better back up the conclusion drawn. Authors have also provided justified responses when the practicality for generating new data was not possible and attempted to address every single concern raised.

Overall, new key information has been satisfactorily incorporated in the manuscript and as a result, this reviewed version has increased in quality. This manuscript is well executed and it will clearly be of interest to the readership of nature communications.

I have no further comments.

Author response:

We appreciate full evaluation and very constructive critical comments on our manuscript.